

# Potential of European ¹⁴CO₂ observation network to estimate the fossil fuel CO₂ emissions via atmospheric inversions

Yilong Wang[1,*], Grégoire Broquet[1], Philippe Ciais[1], Frédéric Chevallier[1], Felix Vogel[1], Lin Wu[1], Yi Yin[1], Rong Wang[1], Shu Tao[2]

[1]Laboratoire des Sciences du Climat et de l'Environnement, CEA-CNRS-UVSQ- Université Paris Saclay, 91191, Gif-sur-Yvette CEDEX, France
[2]Laboratory for Earth Surface Processes, College of Urban and Environmental Sciences, Peking University, Beijing 100871, China

*Correspondence to*: Yilong Wang (yilong.wang@lsce.ipsl.fr)

**Abstract.** Combining measurements of atmospheric $CO_2$ and its radiocarbon ($^{14}CO_2$) fraction and transport modeling in atmospheric inversions offers a way to derive improved estimates of $CO_2$ emitted from fossil fuel (FFCO$_2$). In this study, we solve for the monthly FFCO$_2$ emission budgets at regional scale (i.e. the size of a medium-sized country in Europe) and investigate the performance of different observation networks and sampling strategies across Europe. The inversion system is built on the LMDZv4 global transport model at 3.75 °×2.5 °resolution. We conduct Observing System Simulation Experiments (OSSE) and use two types of diagnostics to assess the potential of the observation and inverse modeling frameworks. The first one relies on the theoretical computation of the uncertainty in the estimate of emissions from the inversion, known as "posterior uncertainty", and on the uncertainty reduction compared to the uncertainty in the inventories of these emissions which are used as a prior knowledge by the inversion (called "prior uncertainty"). The second one is based on comparisons of prior and posterior estimates of the emission to synthetic "true" emissions when these true emissions are used beforehand to generate the synthetic fossil fuel $CO_2$ mixing ratio measurements that are assimilated in the inversion. With 17 stations currently measuring $^{14}CO_2$ across Europe using 2-week integrated sampling, the uncertainty reduction for monthly FFCO$_2$ emissions in a country where the network is rather dense like Germany, is larger than 30%. With the 43 $^{14}CO_2$ measurement stations planned in Europe, the uncertainty reduction for monthly FFCO$_2$ emissions is increased for UK, France, Italy, Eastern Europe and the Balkans, depending on the configuration of prior uncertainty. Further increasing the number of stations or the sampling frequency improve the uncertainty reduction (up to 40% to 70%) in high emitting regions, but the performance of the inversion remains limited over low-emitting regions, even assuming a dense observation network covering the whole of Europe. This study also shows that both the theoretical uncertainty reduction (and resulting posterior uncertainty) from the inversion and the posterior estimate of emissions itself, for a given prior and "true" estimate of the emissions, are highly sensitive to the choice between two configurations of the prior uncertainty derived from the general estimate by inventory compilers or computations on existing inventories. In particular, when the configuration of the prior uncertainty statistics in the inversion system does not match the difference between these prior and true estimates, the posterior estimate of emissions deviate significantly from the truth. This highlights the difficulty to filter the targeted signal in the model-data misfit for this specific



inversion framework, the need to strongly rely on the prior uncertainty characterization for this, and, consequently the need for improved estimates of the uncertainties in current emission inventories for real applications with actual data. We apply the posterior uncertainty in annual emissions to the problem of detecting a trend of FFCO$_2$, showing that increasing the monitoring period (e.g. more than 20 years) is more efficient than reducing uncertainty in annual emissions by adding stations. The coarse spatial resolution of the atmospheric transport model used in this OSSE (typical of models used for global inversions of natural CO$_2$ fluxes) leads to large representation errors (related to the inability of the transport model to capture the spatial variability of the actual fluxes and mixing ratios at sub-grid scales), which is a key limitation of our OSSE setup to improve the accuracy of the monitoring of FFCO$_2$ emissions in European regions. Using a high-resolution transport model should improve the potential to retrieve FFCO$_2$ emissions, and this needs to be investigated.

# 1 Introduction

CO$_2$ emitted from fossil fuels is the major contributor to the increase of atmospheric CO$_2$ (Ballantyne et al., 2015). Knowledge of FFCO$_2$ emissions and their trends is essential to understand the drivers of their variations and assess the effectiveness of agreed upon emission reduction policies over time (Pacala et al., 2010). At national scale, FFCO$_2$ emission inventories are derived based on energy and fuel use statistics, combustion efficiencies and emission factors. These inventories have low uncertainties in OECD countries, and large uncertainties in developing countries due to uncertain energy data and fuel-specific emission factors (Liu et al., 2015; Ballantyne et al., 2015; Andres et al., 2014; Ciais et al., 2010). At sub-national and intra-annual scales, the uncertainties in the estimates of FFCO$_2$ emissions are higher than at national and annual scale (Ciais et al., 2010; Wang et al., 2013) because subnational intra-annual estimates require either the top-down disaggregation of national annual emissions relying on uncertain socio-economic proxies (Wang et al., 2013; Pregger et al., 2007; Oda and Maksyutov, 2011; Andres et al., 2012), or a detailed knowledge of local activity data for a bottom up-scaling of emissions (Gurney et al., 2009). The comparison of different emission maps of that kind also suggests large uncertainties due to treatment of administrative or land/water borders, the use of different proxies, and different spatial resolutions of the maps (Andres et al., 2016), etc. In consequence, national budgets obtained by aggregation of emission maps may have larger uncertainties than those based on national energy use and fuel accounting systems.

Atmospheric inversions exploit the observed variability in atmospheric mixing ratios of CO$_2$ to quantify CO$_2$ fluxes. Inversions have been applied for natural CO$_2$ sources and sinks based on CO$_2$ observations (Broquet et al., 2011; Chevallier et al., 2010; Peylin et al., 2013). Recent attempts to quantify FFCO$_2$ emissions with inversions based on atmospheric CO$_2$ measurements have stressed the importance to measure mixing ratio gradients very close to the emitting source, such as a city (Staufer et al., 2016; Cambaliza et al., 2014; Lindenmaier et al., 2014) or a power plant (Turnbull et al., 2016). Away from the emitting source, the atmospheric signals of FFCO$_2$ emissions mixes with those of natural fluxes, so that FFCO$_2$ emissions can hardly be monitored by atmospheric CO$_2$ measurements only (Shiga et al., 2013). Because of this, monitoring FFCO$_2$ emissions



at national scales, using continental networks of stations located outside the vicinity of the largest sources, is only possible
when measuring an additional tracer specially sensitive to the signal of $FFCO_2$ emissions (Miller and Michalak, 2017; Basu et
al., 2016). Radiocarbon in $CO_2$ is arguably the best tracer (Levin et al., 2003; Turnbull et al., 2006). Pacala et al. (2010)
proposed to estimate national fossil fuel emissions of the US with an inversion based on measurements of radiocarbon in $CO_2$.
Assuming 10,000 atmospheric $^{14}CO_2$ observations at 84 sites per year and a transport model of $5°\times5°$ horizontal resolution,
they suggested that the inversion could reduce the relative uncertainty in monthly emissions of the US from 100% (prior) to
less than 10% (posterior). Ray et al. (2014) assumed virtual $FFCO_2$ observations are sampled every 3 h from a network of 35
measurement towers, and their inversion at $1°\times1°$ resolution could reduce errors on 8 days country-level fossil-fuel emissions
from about 15% (prior) down to 7% (posterior). Basu et al. (2016) developed an inversion system at $1°\times1°$ resolution to account
for the fact that $^{14}CO_2$ is not a perfectly accurate tracer of $FFCO_2$ alone and that its mixing ratio is also affected by natural
fluxes. They showed that given the coverage of $^{14}CO_2$ measurements available in 2010 over North America (969 measurements
per year), the US national total fossil fuel emissions can be constrained with a relative precision of 1% for the annual mean,
and less than 5% for most months.

In all these pioneer studies, the actual spatial scale of the areas emitting $FFCO_2$ is smaller than the grid sizes of the transport
models (from 100 to 500 km). The misfits between the spatial scales controlled or modeled within the inversion system and
those of actual emissions, or those of the $FFCO_2$ patterns in the atmosphere generate errors known as *aggregation* and
*representation* errors (see Sect. 2.2.2), which strongly affect the inversion of $FFCO_2$ emissions (Wang et al., 2017). Those
errors were not formally accounted for in previous $FFCO_2$ inversion studies.

In recent years, as part of the ICOS project, a rather dense network of standardized, long-term and high precision
atmospheric measurements of $CO_2$ has been set up in Europe. Some of the ICOS sites also measure $^{14}CO_2$ and this type of
measurement will be extended in the near term with the aim of determining gradients of $FFCO_2$ mixing ratios across the
European continent. The ICOS atmospheric network is expected to sample 2-week integrated $^{14}CO_2$ at about 40 stations (1,000
analyses    per    year;    ICOS    Stakeholder    handbook    2013    at    http://www.icos-uk.org/uk-icos/sites/uk-
icos/files/documents/Stakeholders%20Handbook%202013.pdf). In this context, network assessment studies are needed to
understand how much this $^{14}CO_2$ network will improve the knowledge on $FFCO_2$ emissions.

In this study, we study the potential of an atmospheric inversion system to quantify $FFCO_2$ emissions at regional scales
(i.e. the size of a medium-sized country in Europe like France or Germany) over the European continent based on continental-
scale networks of atmospheric $CO_2$ and $^{14}CO_2$ measurements. Special attention is paid to the representation and aggregation
errors induced by the use of a coarse grid transport model. Wang et al. (2017) already evaluated the statistics of these errors
for the inversion system that we apply here, which is based on the LMDZv4 global transport model (Hourdin et al., 2006).
Their results highlighted that both the representation and aggregation errors have large magnitudes, and could thus strongly
reduce the ability of the inversion to filter the information on the uncertainties in regional $FFCO_2$ emissions. They also stressed
the fact that the spatial scales of the correlations in the representation and aggregation errors are smaller than that of the





projection in the atmospheric observation space of the typical uncertainties in the prior estimates of regional emissions (called "prior FFCO$_2$ errors" hereafter). Therefore, if the observation networks are dense enough to provide information at finer spatial scale, the impact of aggregation and representation errors on the inversion of the regional budgets of FFCO$_2$ emissions could be small (Wang et al. 2017). In this study, we use the results from Wang et al. (2017) to take into account representation and aggregation errors and we check whether using dense networks could overcome the limitations brought by coarse resolution transport models and by the uncertainties in the distribution of the emissions at high resolution when retrieving regional emission budgets.

Our inversion system solves for monthly FFCO$_2$ emissions in different regions of Europe over a period of one year by assimilating synthetic observations of atmospheric gradients of FFCO$_2$ mixing ratios obtained from co-located CO$_2$ and $^{14}$CO$_2$ measurements at ICOS-like stations. We assume here that $^{14}$CO$_2$ is a perfect tracer of FFCO$_2$ so that the only uncertainties in the FFCO$_2$ mixing ratios data are related to the instrumental precision of CO$_2$ and $^{14}$CO$_2$ measurements. In particular, we ignore the signals of $^{14}$CO$_2$ fluxes (natural fluxes, nuclear power plants) other than those of FFCO$_2$ emissions. Although the results are presented only over Europe, our inversion system is global. We assess the potential of this inversion to improve the estimates of regional fossil fuel emissions based 1) on the statistics of the theoretical prior and posterior uncertainties provided by a Bayesian statistical framework, and 2) on the statistics of the misfits between the prior and posterior estimates of emissions against the assumed "truth" generated by the choice of another emission inventory independent from the one used as prior (see Sect. 2.3). The second type of assessment is used to test the impact of error structures that can hardly be accounted for by the representation of the prior and model uncertainties in the theoretical framework of the atmospheric inversion.

The presentation of the results first focuses on regional FFCO$_2$ emission budgets over one year. It also explores the monitoring of the decadal changes of FFCO$_2$ emissions, compared to a baseline year, which is also of importance since it corresponds to climate mitigation targets set for the Kyoto Protocol and the Intended Nationally Determined Contribution. The trends of FFCO$_2$ emissions over multiple years can be computed using simple regression of series of annual emissions estimates from inventories or atmospheric inversions. The relative uncertainties in decadal trends (e.g. the relative uncertainties in regression slopes) tend to be lower than that in the emission budget of a given year (Pacala et al., 2010), implying that changes can be monitored more accurately than annual budgets. Here, we provide a quantitative analysis of how accurate the trends of national annual FFCO$_2$ emission can be monitored using measurements of FFCO$_2$ mixing ratios.

The paper is organized as follows. Section 2 gives a full description of the inversion and OSSE framework. Section 3 analyzes the statistics of the posterior uncertainties and misfits from inversions using different observation networks. Section 4 evaluates the potential of atmospheric inversion for the monitoring of decadal changes and discusses the relevance of using a coarse-resolution transport model in the inversion system to quantify regional FFCO$_2$ emissions. Conclusions are drawn in Sect. 5.





## 2. Methodology

### 2.1 The configurations of the observation network

We consider three different observation networks, in which the number of the stations ranges from 17 to 233. The minimum network (NE17) includes 17 sites, based on existing European ICOS $^{14}CO_2$ stations in 2016. Using these sites and possible future additional $^{14}CO_2$ stations listed in the 2013 ICOS Stakeholder handbook (available at http://www.icos-uk.org/uk-icos/sites/uk-icos/files/documents/Stakeholders%20Handbook%202013.pdf), we also consider an intermediate $^{14}CO_2$ network of 43 sites (NET43). The NET17 and NET43 networks have high densities in France, Germany, UK and Switzerland, but remain sparse in Eastern Europe (Fig. 1). The corresponding site locations are given in Table S2. We also test a very dense network of 233 sites (NET233), in which two sites are placed in each European land pixel of the LMDZv4 transport model (Fig. 1c).

The high-altitude station Jungfraujoch (JFJ) at 3450 meter above sea level (masl) in Switzerland samples free tropospheric air over Europe, assumed to be representative of the "background" concentration. In all the three configurations of the observation network, JFJ is chosen as the reference station. In this study, we assimilate gradients of FFCO$_2$ between other sites and JFJ in the inversion. Measurements at other sites than JFJ are all assumed to be made at 100 meter above the ground level (magl), the typical height of ICOS tall towers (Kadygrov et al., 2015; Marquis and Tans, 2008).

Wang et al. (2017) have already made a detailed characterization of the distributions of representation errors at the sites considered here and characterized two types of stations based on the population density of the grid cells within which a station is located and on the locations of large point sources (e.g. large power plants). All the sites in different networks are thus categorized as "urban" or "rural" sites according to their results. In the NET233 network, the two sites in each land pixel of the transport model are assumed to be one urban and one rural.

### 2.2 Configuration of the inversion system

The assessment of the potential of different networks to constrain fossil fuel emissions is based on the inversion framework presented by Wang et al. (2017). In this section we summarize the main elements of this framework for which the details can be found in Wang et al. (2017).

#### 2.2.1 Theoretical framework of the Bayesian inversion and diagnostics of the inversion performance in OSSEs

The inversion relies on a Bayesian statistical framework. The estimate of the fossil fuel emission budgets at monthly and regional scales over one year, called hereafter the control variables **x**, is corrected from a prior knowledge of these variables **x**$^b$ (that from a gridded inventory covering the globe). This correction is based on (i) a set of gradients of FFCO$_2$ mixing ratios sampled during the afternoon (see Sect. 2.2.2) across Europe, called hereafter the "observations" **y**$_o$, (ii) the observation operator **H** linking **y** with **x** based on the spatial and temporal distribution of the emissions within a control region and within





a month, on a linear $CO_2$ atmospheric transport model, and on the sampling of the gradients between the corresponding sites and (iii and iv) a modeling of the covariances **B** and **R** of the distributions of the uncertainties in the prior estimate and of the observation errors. The observation error is a combination of the measurement error, the errors from the model transport, representation and aggregation errors. In this study, we ignore the impact on the $FFCO_2$ gradients from the transport model initial conditions that are not controlled by the inversion since it is assumed to be negligible (Wang et al., 2017). Assuming that the prior uncertainties and observation errors are uncorrelated with each other and have unbiased and Gaussian statistical distributions, the statistical distribution of the estimate of **x**, given $\mathbf{x}^b$ and $\mathbf{y}_o$, is also unbiased and Gaussian, and its corresponding mean $\mathbf{x}^a$ and covariance matrix **A** are given by:

$$\mathbf{A} = (\mathbf{B}^{-1}+\mathbf{H}^T\mathbf{R}^{-1}\mathbf{H})^{-1} \tag{1}$$

$$\mathbf{x}^a = \mathbf{x}^b + \mathbf{A}\mathbf{H}^T\mathbf{R}^{-1}(\mathbf{y}_o-\mathbf{H}\mathbf{x}^b) \tag{2}$$

where $^T$ and $^{-1}$ denote the transpose and inverse of a matrix respectively.

Equation (1) shows that **A** depend on neither the value of the observations $\mathbf{y}_o$ nor the prior emission budgets $\mathbf{x}^b$ themselves, but rather on the prior and observation error covariance matrices, on the observation times and locations (through the definition of **H** corresponding to the **y**-space) and on the observation operator. Equation (2) shows that the actual value of $\mathbf{x}^a$ also depends on the observations $\mathbf{y}_o$ and on the prior emission budgets $\mathbf{x}^b$.

A common performance indicator is the theoretical uncertainty reduction (UR) for specific budgets of the fossil fuel emissions (at control or larger space and time scales), defined by:

$$\mathrm{UR} = 1 - \frac{\sigma^a}{\sigma^b} \tag{3}$$

where $\sigma^a$ and $\sigma^b$ are the standard deviations of the posterior and prior uncertainties in the corresponding budget of emissions. Such an indicator can directly be derived from the modeling of **B** and from the theoretical computation of **A** by Eq. (1). Of note is that the scores of uncertainty and of UR given in this study will refer to the standard deviation of the theoretical uncertainty in a specific emission budget.

However, if the modeling of **B** and **R** does not match the actual statistics of the prior and observation uncertainties, or if the theoretical framework of the inversion (assuming that all sources of uncertainty have unbiased and Gaussian distributions, that prior and observation errors are uncorrelated and that the observation operator is linear) is not well satisfied, such a theoretical computation of UR may not reflect the actual performance of the inversion. Wang et al. (2017) derived the statistics of the different components of the observation errors for the same inversion framework as used here. Their statistics of the representation and aggregation errors were based on the comparison of transport model simulations made at high and low spatial resolutions. They highlighted the fact that the distribution of these errors depart from purely Gaussian distributions, and that their covariances can hardly be characterized by the relatively simple models traditionally used in atmospheric inversion systems. In this study, we thus test the inversion system with OSSEs using synthetic truth and errors to build $\mathbf{x}^b$ and $\mathbf{y}_o$ that better reflect the type of observation errors found by Wang et al. (2017). We use Eq. (2) to derive the estimates of $\mathbf{x}^a$ and we



analyze the misfits between $\mathbf{x}^b$ and $\mathbf{x}^a$ against the synthetic true emission budgets $\mathbf{x}^t$. This leads us to define an alternative

indicator of the inversion performance, called misfit reduction (MR) hereafter. While this indicator does not provide an

exhaustive statistical view of the uncertainty in the inverted emissions, it is used to evaluate the confidence in the more

complete (with a full covariance estimate rather than just a realization of the distribution) but more theoretical computation of

the posterior uncertainties and of the UR based on Eq. (1). We write the MR for specific budgets of the fossil fuel emissions

(at control or larger space and time scales) as follows:

$$\mathrm{MR} = 1 - \frac{\varepsilon^a}{\varepsilon^b} \qquad\qquad\qquad (4)$$

where $\varepsilon^a$ and $\varepsilon^b$ are the posterior and prior misfits between the inverted and prior emission budgets against true values for the

corresponding emission budgets.

We focus on uncertainties and misfits at both monthly and annual scales. However, we only have one practical realization

for $\mathbf{x}^b$, $\mathbf{y}_o$ and $\mathbf{x}^a$ (see Sect. 2.3). Consequently, at monthly scale, in order to strengthen the evaluation of the theoretical

uncertainties based on these single realizations of misfits, we compare, for a given region, the quadratic mean of the twelve

monthly misfits (called "monthly misfits" without mention of a specific month in Sect. 3) to the quadratic mean of the standard

deviations of the twelve monthly uncertainties (called "monthly uncertainties" without mention of a specific month in Sect. 3),

which characterizes the average monthly uncertainties over the year. In the result section, UR and MR scores derived at the

"monthly" scale without mention to a specific month will correspond to the relative difference between the prior and posterior

values of these average monthly uncertainties and misfits (following equations similar to Eq. (3) and (4)). At the annual scale,

the diagnostics of UR will have to be compared to MR values for single realizations of the annual misfits. In addition, we

discuss the scores of the relative uncertainty and misfit, defined as the ratios of the absolute uncertainties and misfits to the

absolute prior emission budgets.

### 2.2.2 Practical setup

**Control vector**

The inversion system has a global coverage and controls monthly budgets of $FFCO_2$ emissions for a set of regions during

the year 2007. The map of these regions is given in Fig. 2a. The space discretization of regions is higher where emissions are

the largest in Europe (area of interest, Fig. 2b) and also in the US and China. In other areas with lower emissions or where

observational data to further constrain the prior emissions are lacking (Fig. 2a and Table S1), the size of the control regions is

much larger, and can reach that of a continent. The spatial resolution of the control vector (a region) in Central and Eastern

Europe corresponds to the typical size of a medium-sized European country, but in western Europe apart from Spain, Portugal

and Ireland, where emissions are the highest, the control variables correspond to sub-national regions (e.g. southern and

northern UK, southern and northern Italy, western and eastern Germany, western and eastern France in Fig. 2b). Monthly

emissions over the ocean are included in the control vector, but the ocean is considered as one large region. In total, the world

is divided into 54 land regions and 1 ocean region (Table S1). The inversion solves for the 12 monthly budgets of emissions





for these regions, but not for the spatio-temporal distributions within each region and month. In our framework, choosing year 2007 for the inversion only impacts the meteorological conditions and thus the atmospheric transport conditions. We assume that the atmospheric transport conditions in 2007 are representative of average conditions. We also ignore the impact of inter-annual variations of FFCO$_2$ emissions, which is usually less than 4% (Levin and Rödenbeck, 2008), and of their prior uncertainty (see below the configuration of the prior uncertainty matrix, which is a function of the emissions).

**Time selection of data to be assimilated**

State-of-art inversion systems generally make use of data during the afternoon only, due to limitations of transport models in simulating night-time mixing ratios near the ground. Given the ability to have an intermittent filling of air samples for $^{14}$C analysis (Turnbull et al., 2016; Levin et al., 2008), we thus define the observations to be selectively sampled only during the afternoon (12:00-18:00 local time). Since the cost of the $^{14}CO_2$ analysis of one sample is presently high, monitoring of $^{14}CO_2$ (and thus FFCO$_2$) during a whole year favors the choice of integrated samples at the weekly to 2-week scale (Levin et al., 1980; Turnbull et al., 2009; Vogel et al., 2013). In this study, we first consider 2-week integrated afternoon data. In addition, we present tests with daily afternoon data, for which such a sampling scheme would be more costly. Sampling FFCO$_2$ observations at high temporal resolution should decrease the weight of the random errors on longer time scales, which should improve the potential of the inversions of monthly to annual emission budgets. While inversions are conducted with 2-week samplings for the three networks, daily sampling is tested for NET43 only, which is sufficient to evaluate the usefulness of high frequency sampling.

**Observation operator**

The atmospheric FFCO$_2$ mixing ratios are influenced by the 3-D initial FFCO$_2$ distribution, and by surface emissions during the year. In this study, the inversion rescales all emissions during one year (here 2007) and we ignored initial conditions on January 1$^{st}$ which are rapidly transported out of Europe and do not cause subsequent FFCO$_2$ gradients between European sites (Wang et al., 2017). The observation operator is restricted to a matrix **H** which consists of a chain of three sub-operators, **H**=**H**$_{samp}$**H**$_{transp}$**H**$_{distr}$, where **H**$_{distr}$ distributes regional monthly emission budgets into a gridded emission map at the resolution of the transport model, **H**$_{transp}$ is the atmospheric transport model, and **H**$_{samp}$ samples the FFCO$_2$ gradients corresponding to the observation vector from the transport model outputs (Wang et al. 2017).

We use the high-resolution (0.1°) annual FFCO$_2$ emission map from the PKU-CO$_2$ inventory in the year 2007 (Wang et al., 2013) to distribute the emissions in space within each region. PKU-CO$_2$ is an annual emission map with no temporal profile, so that the modelled temporal distribution in **H**$_{distr}$ is flat between months. This implementation of **H**$_{distr}$ is denoted **H**$_{distr}$$^{PKU}$.

The off-line version of the general circulation model of Laboratoire de Météorologie Dynamique LMDZv4 (Hourdin et al., 2006) forms **H**$_{transp}$. Atmospheric transport simulations was nudged to analyzed wind fields from the European Centre for Medium-Range Weather Forecasts (ECMWF) Interim Reanalysis (ERA-Interim, Dee et al., 2011) for the year 2007. We denote this implementation of **H**$_{transp}$ by **H**$_{transp}$$^{LMDZ}$.

The sampling of FFCO$_2$ gradients relies on the extraction of individual simulated mixing ratio data at the measurement



locations and chosen temporal sampling frequency, followed by the computation of differences (gradients) between time series of $FFCO_2$ mixing ratios at each site and that at the JFJ reference site. The mixing ratio data for a given site is sampled at the chosen sampling height in the transport model grid cell containing this site. We recall that the sampling height is 100 m above ground level (magl), the $1^{st}$ level of LMDZv4, except for JFJ being at 3450 m above sea level (masl), the $6^{th}$ level. The resulting implementation of $\mathbf{H}_{samp}$ is denoted $\mathbf{H}_{samp}^{coloc}$.

In sum, the practical observation operator used for inversions is defined by $\mathbf{H}^{prac}= \mathbf{H}_{samp}^{coloc}\ \mathbf{H}_{transp}^{LMDZ}\ \mathbf{H}_{distr}^{PKU}$.

### Prior error covariance matrix

Emission estimates from inventories are limited to annual and national scales and rarely provide systematic assessments of uncertainties. There are a limited number of datasets providing emission maps at higher spatial/temporal resolutions. Although there have been some efforts to compare such $FFCO_2$ emission maps (Macknick et al., 2009; Ciais et al., 2010; Andres et al., 2012; Andres et al., 2016), the ability to characterize the uncertainties of an emission inventory is limited, especially for sub-national and sub-annual scales. In this study, we use different streams of information to model the prior emission uncertainty covariance matrix $\mathbf{B}$ and we use two different configurations of this matrix in the inversions.

The first configuration of the $\mathbf{B}$ matrix, called here notional or $\mathbf{B}^{notion}$, is related to the notional estimates of (1-sigma) uncertainties for national emissions claimed by inventory compilers to range from 1-2.5% for the USA (US EPA, 2015), 2%-7% for European countries (Andres et al., 2014; Ballantyne et al., 2015) to 7.5-10% for China (Gregg et al., 2008; Liu et al., 2015). However, Ciais et al. (2010) found that the ratios between geographically distributed emission maps, even after correction for inconsistencies and aggregated at national scale, ranged from 0.86 to 1.5, which is larger than the uncertainties claimed by inventory compilers. In this study, the prior uncertainty covariance $\mathbf{B}^{notion}$ of monthly emissions is set up based on three constraints: 1) the relative uncertainty in annual emission equals 10% for US and European national budgets, 15% for China, and 10% for individual control regions outside US, Europe and China; 2) uncertainties in monthly emissions have a 2-month exponentially decaying temporal auto-correlation, and 3) spatial correlations between uncertainties in monthly emissions across adjacent regions within the same country are fixed to -0.2, a negative value to account for the fact that sub-national emissions are usually disaggregated from national inventories, so that a positive bias in part of a country must be compensated by a negative one in another. All other spatial correlations in $\mathbf{B}^{notion}$ are assumed to be null, and the overall correlation matrix in $\mathbf{B}^{notion}$ is derived from the Kronecker product of temporal and spatial correlation matrices (assuming that the correlation between two control variables are given by the product of the spatial and temporal correlations between the two corresponding control regions and the two corresponding time window respectively). The full computation of $\mathbf{B}^{notion}$ is detailed in Appendix A. With this setting, prior uncertainties in monthly emissions can exceed 10% and be as large as 30% for some sub-national control regions.

The second configuration of the $\mathbf{B}$ matrix, known as empirical or $\mathbf{B}^{empiric}$, is based on the empirical derivation of the statistics of the differences between two spatially gridded emission maps (which will be used to define the prior and true estimate of emissions in the OSSEs, see Sect. 2.3). The two maps are PKU-$CO_2$ (Wang et al., 2013, http://inventory.pku.edu.cn/)




and IER-EDG (available at http://carbones.ier.uni-stuttgart.de/wms/index.html), both corresponding to the year 2007. The IER-EDG map combined EDGAR annual map with country specific temporal profiles (monthly, daily and hourly) from IER. In general, the differences in annual emissions from the control regions in Europe between these two emission maps range

from 3% to 20%, except for the Balkans where they reach up to 44%. Statistics of the difference between the two maps aggregated at the control region scale are fitted by a covariance model that combines four different covariance matrices, with exponentially decaying temporal correlations at time scales of 1 month, 3 months, and 6 months for the first three ones respectively, and a full temporal correlation over the year for the fourth one (representing the annual bias on the prior emissions). The mathematical formulation for this computation and the full derivation of $\mathbf{B}^{empiric}$ is detailed in the Appendix B. In addition,

we assume that there is no spatial correlation of the prior uncertainty between different control regions. Again, the overall correlation matrix in $\mathbf{B}^{empiric}$ is derived from the Kronecker product between the temporal and spatial correlation matrices.

$\mathbf{B}^{empiric}$ is built using an error covariance model which cannot perfectly characterize the structure of the differences between the PKU-CO$_2$ and IER-EDG budgets at the control resolution, which will be used to derive realistic $\mathbf{x}^b$ and $\mathbf{x}^t$ respectively and thus the "actual prior errors" in the OSSEs with synthetic data (see Sect. 2.3). However, by construction,

$\mathbf{B}^{empiric}$ better fits these errors in our OSSEs than the $\mathbf{B}^{notion}$ matrix in terms of both the standard deviation of the uncertainty at the 1 month / regional scale and the temporal correlations. The differences between the results of the inversions using either $\mathbf{B}^{empiric}$ or $\mathbf{B}^{notion}$ will be used to give an estimate of the range of the inversion skills as a function of different assumptions regarding the prior uncertainty in emission budgets.

**Observation error covariance matrix**

Wang et al. (2017) derived estimates of the observation errors in FFCO$_2$ gradients across Europe when using the same inverse modeling framework as in this study. They analyzed four sources of observation errors (i.e. sources of misfits when comparing the modeled to the measured FFCO$_2$ gradients other than the uncertainties in the estimates of the emission budgets at the 1-month and regional scale), one related to the FFCO$_2$ data, three to the observation operator:

1) The measurement error $\varepsilon_i$ on FFCO$_2$ gradients is simply assumed to be 1 ppm with no temporal and spatial correlations,

which corresponds to the typical precision of the analysis of air samples by accelerator mass spectrometry (AMS) for $^{14}CO_2$ (2‰-3‰) (Hammer et al., 2016; Turnbull et al., 2014).

2) The representation error $\varepsilon_r$ arises from the mismatch between the coarse resolution of modelled emissions and concentrations in the observation operator (here the transport model) and the spatial variability of the actual emissions and concentrations.

3) The transport errors $\varepsilon_t$ is due to discretized and simplified equations for modeling transport, using a given meteorological

forcing in practice.

4) The aggregation error $\varepsilon_a$ arises from the mismatch between the control resolution (budgets of regions in each month) and the resolution of the emission modeled in the observation operator (here the transport model). It reflects uncertainties in $\mathbf{H}_{distr}$.

In this study, we use the estimates of the standard deviations and of the correlation functions for these different types of observation errors from Wang et al. (2017) to set up the $\mathbf{R}$ matrix. Wang et al. (2017) sampled representation and aggregation




errors by using simulations with a mesoscale (with higher resolution than LMDZv4) regional transport model and by degrading the spatial and temporal resolution of the emission maps in the input of this model and in the output FFCO$_2$. Based on these samples, the standard deviation of $\varepsilon_r$ was characterized by a function of season and on whether a station is "urban" or "rural" (see Sect. 2.1). For $\varepsilon_a$, the standard deviation for spring/summer and autumn/winter were derived. The temporal and spatial correlations in the representation and aggregation errors were characterized by the sum of a long-term component and a short-term component: $r(\Delta t)=a \times e^{-\Delta t/b}+(1-a) \times e^{-\Delta t/c}$ where $\Delta t$ is the timelag (in days) and $a$, $b$, $c$ are parameters optimized by regressions against the samples of the errors. Following Wang et al. (2017), we do not account for the spatial correlation in the **R** matrix, as the scale of the spatial correlation is smaller than the size of the grid cells of the global transport model $\mathbf{H}_{transp}^{LMDZ}$ used for the inversion. The standard deviation of the transport error at a given site is assumed proportional to the temporal standard deviation of the 1-year long time-series of the high-frequency variability of the detrended and deseasonalized simulated daily mean afternoon mixing ratios in the grid cell of the transport model, at which the sites are located. The corresponding values of the standard deviation and the modelling of temporal autocorrelation of the observation errors for 2-week/daily mean afternoon FFCO$_2$ gradients are listed in Table S3 and Table S4.

Assuming that all these sources of errors are independent from each other and have Gaussian and unbiased distributions: i.e. $\varepsilon_i \sim N(\mathbf{0}, \mathbf{R}_i)$, $\varepsilon_r \sim N(\mathbf{0}, \mathbf{R}_r)$, $\varepsilon_t \sim N(\mathbf{0}, \mathbf{R}_t)$, $\varepsilon_a \sim N(\mathbf{0}, \mathbf{R}_{adistr})$, **R** is given by the sum of the covariance matrices corresponding to each of them: $\mathbf{R} = \mathbf{R}_i + \mathbf{R}_r + \mathbf{R}_t + \mathbf{R}_a$.

### 2.3 Configurations of the OSSEs

In this study, we consider two types of OSSEs corresponding to the two configurations of prior error covariance matrix $\mathbf{B}^{notion}$ and $\mathbf{B}^{empiric}$. The first OSSEs use $\mathbf{B}^{notion}$ (called here INV-N), while the second type of OSSEs use $\mathbf{B}^{empiric}$ (called here INV-E). As discussed in Sect. 2.2.1, in both types of OSSEs, the theoretical computation of the posterior uncertainty and UR is based on Eq. (1). These diagnostics would perfectly characterize the performance of the system if the prior uncertainty and the observation errors have Gaussian and unbiased distributions that are perfectly characterized by the set-up of the prior uncertainty covariance matrix **B** and observation error **R** in the inversion system. In both types of OSSEs, these diagnostics are evaluated based on a practical application of Eq. (2), and on the analysis of posterior misfits and MR, with a synthetic truth (true emissions and true observation operator) and observations that are generated in a similar way as in Wang et al. (2017). Here, the "actual" prior and observation errors have a complex origin and structure which is not perfectly adapted to the unbiased and Gaussian assumptions and are not perfectly reflected by the set-up of the prior uncertainty covariance matrix **B** and observation error covariance matrix **R** in the inversion system, even in INV-E where $\mathbf{B}=\mathbf{B}^{empiric}$ and **R** are fitted to the "actual" prior and observation errors. Of note is that in INV-N, $\mathbf{B}^{notion}$ has significant inconsistencies with the actual differences between $x^b$ and $x^t$, so that, in this experiment, the analysis of the posterior misfits and MR will be used to evaluate the performance of the inversion when using a poor configuration of the prior uncertainty covariance matrix in the inversion system in addition to accounting for errors which hardly fit with the assumption that their distribution is Gaussian and unbiased.



This corresponds to situations for which there is little knowledge about the uncertainties in the inventories used for inversions with real data. The analysis of misfits and MR in INV-N is thus more pessimistic than that in INV-E.

In the OSSEs, the synthetic prior estimate of the regional/monthly emissions $\mathbf{x}^b$ is built based on the emissions from PKU-$CO_2$ ($\mathbf{x}^{PKU}$ hereafter). The synthetic true emission budgets and synthetic observations are modelled using a realistic representation the "actual" emission budgets $\mathbf{x}^t$ and of the "actual" $\mathbf{H}_{distr}$ operator based on the relatively independent IER-EDG inventory. The synthetic true regional/monthly emissions and the synthetic true $\mathbf{H}_{distr}$ operator are thus referred to as $\mathbf{x}^{IER\text{-}EDG}$ and $\mathbf{H}_{distr}^{IER\text{-}EDG}$ hereafter. The synthetic observations are generated using $\mathbf{x}^{IER\text{-}EDG}$ and the operator $\mathbf{H}^{OSSE}=\mathbf{H}_{samp}^{coloc}\mathbf{H}_{transp}^{LMDZ}\mathbf{H}_{distr}^{IER\text{-}EDG}$, which relies on the same $\mathbf{H}_{samp}^{coloc}$ and $\mathbf{H}_{transp}$ operators as the $\mathbf{H}^{prac}$ observation operator

used in the inversion system. Consequently, the difference between $\mathbf{H}^{OSSE}$ and $\mathbf{H}^{prac}$ underlies aggregation errors only. Therefore, in order to account for the transport, representation and measurement errors, the data $\mathbf{H}^{OSSE}\mathbf{x}^{IER\text{-}EDG}$ are perturbed following the statistics of the corresponding errors as detailed in Sect. 2.2.2.

The parameters of the two inversion configurations are summarized in Table 1 and Fig. 3. All the combinations of networks and data temporal sampling described in Sect. 2.1 and 2.2.2 are tested with the two configurations of OSSEs. The

resulting eight OSSEs are listed in Table 2.

## 3. Results

### 3.1 Assessment of the performance of inversions when using the NET17/NET43 and 2-week integrated sampling

### 3.1.1 Analysis of the results at the regional and monthly scale

Figure 4 shows the URs of monthly emissions using the NET17 and NET43 networks and 2-week sampling (N-17W, E-

17W, N-43W and E-43W in Table 2). With NET17, INV-N and INV-E inversions show similar spatial patterns of UR scores. The largest UR occurs in region western Germany, being 34% for inversion N-17W and 38% for E-17W. The URs are also significant in eastern Germany for both inversions. This stems from the fact that several stations are located around and within these regions and that the emission in these regions are higher than those in other regions. Moderate UR values are found for Benelux (12%) and Eastern France (15%) in inversion E-17W and the UR values elsewhere are marginal. Going from NET17

to NET43 adds a significant increase (improvement) of the UR for southern UK (from 3% to 23%), northern Italy (from 3% to 18%) and Eastern Europe (from 2% to 15%) in INV-N (Fig. 4d). The increase of UR in E-43W, compared with the UR in E-17W, mainly occurs in eastern France (from 16% to 33%) and the Balkans (from 3% to 13%). Because the added stations in NET43, compared to NET17, are mostly located outside Germany, the URs over western and eastern Germany are not significantly improved (Fig. 4d and 4e). Despite their different URs for specific regions, both types of inversions highlight the

overall increase in the UR for western European regions by increasing the number of sites from NET17 to NET43.

The differences in the spatial patterns of UR between INV-N and INV-E inversions shown in Fig. 4 reveal the high



sensitivity of UR to the configuration of the prior uncertainties. Figures 5a and 5b show the prior uncertainties associated with the two configurations of $\mathbf{B}^{notion}$ and $\mathbf{B}^{empiric}$. The regions where these uncertainties and thus the potential for reducing these uncertainties from the inversion are the highest are very different between $\mathbf{B}^{notion}$ and $\mathbf{B}^{empiric}$. For example, $\mathbf{B}^{empiric}$ defines a

much larger uncertainty than $\mathbf{B}^{notion}$ over eastern France (43% vs 16%) while the opposite is true for southern UK (4% vs 14%). As a result, the UR of eastern France is 33% in E-43W and 8% in N-43W, and the UR of southern UK is 2% in E-43W and 23% in N-43W.

Complementing the uncertainty reduction, Fig. 5 shows the prior and posterior uncertainties and provides insight into the precision of the estimates of monthly $FFCO_2$ emissions after inversion with NET17 and NET43 and 2-week sampling. For

example, using NET17, uncertainties in monthly $FFCO_2$ emissions are reduced from 29% (or 17%) in the prior estimates to 17% (or 9%) in the posterior estimates for western Germany in INV-N (or INV-E). Using additional sites in NET43 reduces the uncertainties in monthly $FFCO_2$ emissions in southern UK from 25% in the prior estimates to 19% in the posterior estimates in INV-N, and reduces the uncertainties in monthly $FFCO_2$ emissions in eastern France from 44% in the prior estimates to 29% in the posterior estimates in INV-E. Like the UR, posterior uncertainties and their spatial variations are different between INV-

N and INV-E inversions, and demonstrate a strong dependence on the choice of $\mathbf{B}=\mathbf{B}^{notion}$ or $\mathbf{B}=\mathbf{B}^{empiric}$.

The scores of the MR and misfits of monthly emissions in both inversions using NET17 and NET43 and 2-week sampling are shown in Fig. 4 (b, d, f, h) and Fig. 5 (b, d, f, h, j, l). In INV-E, there are slight differences between posterior misfits and uncertainties, and between MR and UR. For example, for E-43W, the MR (21%) for Iberian Peninsula is larger than the UR (5%), while the MR (40%) for western Germany is slightly smaller than the UR (47%). Despite such differences, the spatial

patterns of the MRs in Fig. 4 and posterior misfits in Fig. 5 are close to those of the URs and posterior uncertainties. On the contrary, there are large differences between the statistics of posterior misfits and posterior uncertainties, and between MRs and URs in INV-N. In some regions, such as southern UK (MR= -0.9 in N-17W and MR= -1.4 in N-43W) and northern Italy (MR= -0.4 in N-17W and MR= -1.5 in N-43W), the MRs are negative and far below zero. This means that the posterior misfits are even larger than the prior misfits (comparing Fig. 5f and 5j with Fig. 5b), and thus a degradation of the emission estimates

from the inversion is seen in these regions when assimilating $FFCO_2$ data. This suggests that the theoretical computation of posterior uncertainty poorly characterizes the actual performance of the inversion in practice when the configuration of the prior uncertainty covariance matrix and the actual prior errors are not consistent.

Figure 6 shows the correlations in the prior and posterior uncertainties in monthly emissions from different regions, and their differences in inversions N-43W and E-43W. After assimilating the observations, the change of correlations mainly occurs

among regions that have large URs. In both inversions, there are negative correlations between the posterior uncertainties in monthly emissions from some neighboring regions, in particular between western Germany and eastern Germany (from -0.27 to -0.18 depending on the months). The negative correlations between the posterior uncertainties in monthly emissions of different regions indicate that NET43 brings a strong constraint on the budgets over a large area but does not separate individual regions so well. At the same time, the temporal correlations in the posterior uncertainties between different months for a given



region also change after the inversion. For example, in INV-N, temporal correlations between posterior uncertainties in monthly emissions for a specific region are smaller than those between prior uncertainties for that region when the time lag is smaller than 3 months, while they are larger than the ones in prior uncertainties when the time lag exceeds 3 months (Fig. 6e). Because our setup of $\mathbf{B}^{\text{notion}}$ only considers an exponentially decaying temporal correlation with a correlation length of 2 months (Sect. 2.2.2), these longer term correlations in monthly posterior uncertainties must hence be driven by the temporal

correlations in observation error, which contains a long-term component (see Sect. 2.2.2). On the contrary, in INV-E where $\mathbf{B}^{\text{empiric}}$ includes a component with annual-scale temporal correlations, the temporal correlations between posterior uncertainties in the monthly emissions are smaller than those between prior uncertainties. The analysis of the correlations in the prior or posterior uncertainties from N-17W and E-17W leads to very similar conclusions, but is not shown here.

### 3.1.2 Analysis for annual emissions

We compare the performance of different inversions to constrain annual mean FFCO$_2$ emissions. Corresponding UR and MR values are shown in Fig. 7. The patterns and values of UR for annual emissions are very similar to those at monthly scale (Fig. 4). High URs and MRs occur mostly in regions where the observation networks are dense and the emissions are high. For example, up to 47% UR is achieved for annual emissions in western Germany when using network NET43 and 2-week sampling. As a result, the posterior uncertainties of annual fossil fuel emissions, when using NET43 with 2-week sampling,

are 10% (or 4%) for southern UK, 8% (or 8%) for western Germany and 15% (or 28%) for eastern France in INV-N (or INV-E).

Both the spatial spread and the magnitude of the MR of annual emissions in INV-E (Fig. 7d and 7h) are larger than those of the UR. The differences between MR and UR are much larger at annual than at monthly scale (when comparing Fig. 4 and 7). The cause of the discrepancy between UR and MR was presented in Sect. 2.2.1, and it may have a larger impact at the

annual scale than at the monthly scale due to the evaluation of annual UR scores to annual MR values corresponding to single realizations of the misfits. In INV-N, the spatial spread and the magnitude of the MR are still significantly different from those of the UR and the MRs for some regions are still negative and far below zero.

### 3.2 Impact of using daily measurements and using a dense observation network

Figure 8 shows the URs and MRs of monthly emissions from inversions using NET43 and daily sampling, and from

inversions using NET233 network and 2-week sampling (N-43D, E-43D, N-233W and E-233W in Table 2). When using NET43 and daily sampling, the URs of monthly emissions are generally larger (improved) than when using 2-week sampling for all regions. The differences between the UR values of monthly emissions with daily and with 2-week sampling are larger (meaning more improvement with daily sampling) over the regions where the network is dense and the emissions are high. For instance, the UR of monthly emissions for western Germany are as high as 62% (or 67%) in INV-N (or INV-E). When

using the much denser NET233 network yet with a lower 2-week sampling (Fig. 8 d-f), we found that UR of monthly emissions





in some regions that were poorly sampled by networks NET17 and NET43 are largely improved. For instance, UR value in Eastern Europe is 36% in N-233W (compared with 15% in N-43W) and is 73% in the Balkans in E-233W (compared with 13% in E-43W). In principle, large regions tend to encompass more sites and to be surrounded by more sites than small regions, and thus may have more observations to improve their estimates of emissions. However, in both N-233W and E-233W, the URs for regions with a large area like northern Europe are still limited to below 5%. Large URs are identified over the regions whose absolute uncertainties are high, revealing the important roles of the absolute prior uncertainties when using the coarse-resolution transport model in the inversion of $FFCO_2$ emissions over Europe. The scores of MR match relatively well those of UR only in E-43D and E-233W (INV-E inversions) but not in N-43D and N-233W (INV-N inversions) (comparing Fig. 8d versus Fig. 8c, and Fig. 8h versus Fig. 8g). Even though the temporal frequency or spatial coverage of the sampling of the $FFCO_2$ mixing ratios are largely improved using NET43 and daily sampling, or NET233 and 2 week sampling, the MRs are still negative and below zero for a large number of regions in Europe.

## 4. Discussion

### 4.1 Implication for long-term trend detection of fossil fuel emissions

In the Copenhagen conference of parties, the European Union (EU) set up the goal to decrease its emissions (in $CO_2$ equivalents) by 80%−95% below 1990 by 2050 (European commission, 2010). In 2015, the EU Intended Nationally Determined Contribution (INDC) submitted to the UNFCCC set a target of 40% domestic greenhouse gas emissions reduction below 1990 levels by 2030. These targets translate into annual reductions compared to 1990 of roughly 1% per year in the 2020s, 1.5% in the decade from 2020 until 2030, and 2 % in the two decades until 2050 (European commission, 2010). Levin and Rödenbeck (2008) showed that, taking into account the inter-annual variations of the atmospheric transport, changes of 7-26% between two consecutive 5-year averages of $FFCO_2$ emissions in south-western Germany could be detected at the 95% confidence level with monthly mean gradients of $^{14}CO_2$ observations between two stations (Schauinsland and Heidelberg) and the reference site JFJ. Such a detectability skill is clearly insufficient to support the "verification" of 1-2% annual change of emissions per year (meaning 5-10% changes between two consecutive 5-year averages) corresponding to the EU targets. Here, we evaluate the skill to detect trends when using the much larger $^{14}CO_2$ networks and the atmospheric inversion framework detailed in this study.

The uncertainty in the trend of $FFCO_2$ emissions calculated from the linear regression of a series of annual estimates, is independent of this trend itself (see Appendix C). This allows us to extrapolate posterior uncertainties in annual emissions from this study to investigate the detectability of emissions trends. Assuming that the uncertainties in annual emissions of different years are fully independent and ignoring the changes in transport on decadal scales (Aulagnier et al. 2009; Ramonet et al., 2010), we calculate the uncertainty in relative trends for different time lengths as a function of the posterior uncertainty in annual emissions (Table 3). Here, the relative trend is defined as the ratio of the linear regression slope of emissions to the



emission in the base year. Using NET17 or NET43 and 2-week sampling, the posterior uncertainty in annual emissions of some well-sampled regions, e.g. Germany, is largely below 10% (Sect. 3.1.2). In this case, given Table 3, the uncertainty in the relative trends over 20 years is in the range of 0.27% $yr^{-1}$ to 0.43% $yr^{-1}$. However, the uncertainty in trend estimation over

10 years would be 1% $yr^{-1}$. The EU target of 1-2% annual reduction, could thus be verified using NET17 or NET43 in these well-sampled regions over a period of 20 years but not over a period of 10 years. For other regions with sparser coverage of stations, either the posterior uncertainty in annual emissions are much larger than 10% (e.g. in Ireland and Balkans in INV-E) or the URs (or MRs) of annual emissions are marginal (meaning no improvement in the estimate of annual emissions from the inversion), so that the verification of the trend in these regions based on the inversion framework of our study is thus

challenging.

## 4.2 Adequacy of large-scale atmospheric inversion for the monitoring of fossil fuel emissions and potential improvements of the inversion skills

In this study, we showed that given the NET17 $^{14}CO_2$ measurement station network, the potential of our atmospheric inversion of fossil fuel emissions at large scale using a coarse-resolution model is limited (Fig. 4 and Fig. 5). When using the

denser NET43 network and 2-week sampling and assimilating ~1000 measurements per year, the potential of the inversion system is improved, yet mainly over high emitting regions.

We paid attention (as compared to previous OSSEs published for the USA) to account for aggregation and representation errors, which is the reason why our inversions do not provide as impressive error reductions as those of Ray et al. (2014) and Basu et al. (2016). However, we still did not account for all sources of uncertainty. Indeed, we assumed that atmospheric

$FFCO_2$ gradients can be derived from the $^{14}CO_2$ measurements with a precision of 1 ppm. This 1-ppm standard deviation approximately corresponds to the errors in the atmospheric measurements and ignores uncertainties in the conversion of $^{14}CO_2$ and $CO_2$ measurements into $FFCO_2$. Various fluxes that influence atmospheric $^{14}CO_2$, such as those from cosmogenic production, ocean, biosphere and nuclear facilities, bring systematic errors to the conversion of $^{14}C$ measurements into $FFCO_2$ (Lehman et al., 2013; Vogel et al., 2013). For example, over land regions, heterotrophic respiration is expected to be one of the

main contributors to the large-scale signals of atmospheric $^{14}CO_2$ (Turnbull et al., 2009). Over regions like Europe, $^{14}C$ emissions from nuclear facilities may have even larger influences than plant and heterotrophic respiration in some areas (Graven and Gruber, 2011). The influences from these fluxes will introduce additional errors in the $FFCO_2$ gradients.

In Sect. 3.3, we explored the concept of having more observations assimilated in the inversion system by increasing the sampling frequency and expanding the observational network. Wang et al. (2017) showed that because the representation error,

aggregation error and the prior $FFCO_2$ errors have very similar error structures in time, it is difficult to use daily sampling to filter uncertainties in the prior estimate of the emissions. However, we showed that when using NET43 and daily sampling, the UR of monthly emissions is still much larger than using 2-week sampling. This stems from the fact that having daily sampling decreases the weight of the measurement errors at the 2-week to annual scales, which are assumed not to have



temporal autocorrelations. We also tested the concept of extending the observation network to a very dense configuration, NET233, with a wide coverage across Europe. It exhibits a significant increase in the UR of monthly emissions across Europe, especially over Eastern Europe. Emissions in Northern Europe, however, remains poorly constrained. This illustrates the limitation of using a coarse resolution transport model to quantify fossil fuel emissions. Such a limitation is attributed to the following facts: 1) the observation error in the inversions are larger than the prior $FFCO_2$ error (typically 0.21 ppm for 2-week mean afternoon $FFCO_2$ gradients and 0.49 ppm for daily mean afternoon $FFCO_2$ gradients, Wang et al., 2017); and 2) the observation errors bear complex temporal and spatial correlations which are close to the prior $FFCO_2$ errors (Wang et al., 2017).

Wang et al. (2017) showed that the representation error contributes the most to the observation errors, followed by the transport and measurement errors. The representation and the transport error are highly dependent on the transport model resolution. Increasing the transport model resolution will reduce the representation errors and (potentially) reduce the transport error, if topography effects and synoptic variations are better simulated by finer-resolution models. We thus assume that using a regional mesoscale transport model with higher resolution than LMDZv4 (like for the regional scale natural flux inversions in Kadygrov et al., 2015; Broquet et al., 2013; Gourdji et al., 2012; Lauvaux et al., 2008) should be the most efficient way to improve the results from atmospheric inversion of $FFCO_2$ emissions at regional scale.

However, unlike such regional transport models, a global transport model can propagate uncertainties in emissions in other continents to Europe and thus allow to account for them when estimating the European emissions. To quantify the impact of the uncertainties in emissions from other continents, we conducted additional inversions that only solve for emissions in European regions ignoring those of other continents. The results show that emissions from other continents have negligible impacts on UR, MR and posterior emission budgets of European regions (the relative differences between these estimates being smaller than 1%; not shown). This indicates that the inversion system mainly exploits the signals of the gradients between the European sites to constrain the European emissions, and the incoming $FFCO_2$ over the European air-shed from emissions outside the European continent, results in very small $FFCO_2$ gradients between JFJ and other stations in Europe. As a result, it highlights the possibility of using a mesoscale regional transport model and a regional inversion framework to derive monthly and national scale emission budgets from $^{14}CO2$ networks in Europe. In such a framework, the uncertainties in the signals of emissions from remote emissions outside Europe could be neglected or coarsely accounted for by controlling the regional transport model boundary conditions.

## 4.3 The need for good estimates of the uncertainties in the prior estimate of the emissions from inventories

The inconsistencies between the posterior misfits and the theoretical computation of posterior uncertainties, and between the scores of MR and UR in INV-N inversions indicate that the theoretical computation of posterior uncertainty is not sufficient to characterize the actual performance of the inversion, especially when the prior uncertainty covariance matrix does not capture the actual error statistics of the prior estimate of the emissions. Moreover, in INV-N, there is a degradation of the emission estimates for many regions, characterized by negative and far-below-zero MRs in Sect. 3. This degradation occurs





even when using daily measurements or the network NET233. All these analysis reveal the difficulty to capture the signatures of uncertainties in the prior emission estimate and thus to derive good corrections when the prior uncertainty covariance matrix is not configured properly. This is likely due to the difficulty to filter such signatures from the assimilated prior-model data misfits in our specific inverse modeling problem. This difficulty is due to the fact that the standard deviation of the observation

error is generally larger than that of the prior $FFCO_2$ error and that the observation error has a similar structure of temporal autocorrelations to that of the prior $FFCO_2$ error, as highlighted by Wang et al. (2017). In such a situation, only a precise configuration of the prior uncertainty covariance matrix can support the filtering of the prior errors. Consequently, even though both $\mathbf{B}^{empiric}$ and $\mathbf{B}^{notion}$ are derived from realistic assumptions on the uncertainties in the inventories, and to some different extent, from the analysis of inventory maps, the inconsistencies between these two matrices lead, in general, to positive MR

when using the former and negative ones when using the latter.

In real applications, having such a good fit between the configuration of the prior uncertainty covariance matrix in the inversion system as between $\mathbf{B}^{empiric}$ and the synthetic prior errors in our OSSEs could appear to be unlikely, especially since the difference between $\mathbf{B}^{empiric}$ and $\mathbf{B}^{notion}$ illustrates the range of assumption we could have on the uncertainties in the existing inventories. Consequently, in order to improve the estimate of $FFCO_2$ emissions, on the one hand, more detailed and systematic

evaluations of the uncertainty in the $FFCO_2$ emission inventories and of their potential temporal/spatial correlations (Andres et al., 2014; 2016b) would be required. On the other hand, as mentioned in Sect. 4.2, using a regional mesoscale transport model with higher resolution would reduce the representation error and (potentially) the transport error, and thus the observation error. Such a model would be needed to decrease the ratio of the observation error to the prior $FFCO_2$ error and thus increase the ability to filter the prior errors from the prior-model data misfits.

**5. Conclusion**

In this study, we present the application of a global atmospheric inversion method to quantify $FFCO_2$ emissions over Europe at regional scale using three continental networks of $^{14}CO_2$ measurement sites. Its framework has been introduced by Wang et al. (2017). This method combines a prior emission estimate from an inventory with the information from atmospheric observations of $FFCO_2$ gradients to provide improved emission estimates with reduced uncertainties. A set of inversions are

performed to test the potential of such a global atmospheric inversion system and the relevance of the large-scale inverse modeling (using coarse resolution transport model and controlling the emissions at regional scale) to monitor $FFCO_2$ emissions. The results show that given the 17 $^{14}CO_2$ measurement stations that are available in 2016 and the typical 2-week sampling frequency, the inversion reduces the uncertainties in monthly emission estimates for western Germany by 34% to 38%, depending on the setup of the prior uncertainty. By using a plausible network containing 43 measurement stations which is

planned for the future and using 2-week sampling, one could expect higher URs of the emissions over the high emitters in Europe, e.g. eastern France (16% to 33%), southern UK (3% to 23%). In addition, given the posterior uncertainty in the





emissions that could be achieved in such an inversion system, the uncertainties in the regressed trends can be significantly reduced below 1% $yr^{-1}$ by monitoring the $FFCO_2$ emissions for more than 20 years.

Increasing the number of observations assimilated in the inversion system by using daily sampling or a very dense observational network could potentially increase the UR over European regions. However, even though the inverse modeling framework used here can be assumed to be optimistic, e.g. regarding the assumption of the $FFCO_2$ data precision (see Sect. 2.2.2), its potential to improve the estimate of $FFCO_2$ emissions is often limited. The concept of using a coarse-resolution transport model in a global inversion system to solve for fossil fuel emissions of the regions whose emissions are not as high as those of Germany/France is challenged by the fact that coarse-resolution transport model can hardly filter the signature of

the uncertainties in the emission budget from other signals and sources of errors within their coarse grid cells. Thus, regional high-resolution transport models could thus be required for the monitoring of $FFCO_2$. At the same time, the posterior estimate of the emissions are much degraded when the configuration of prior uncertainty in the inversion system is improper, implying that systematic evaluations of the uncertainties and temporal and spatial correlations in $FFCO_2$ emission inventories are also needed to improve the estimate of $FFCO_2$ emissions when applying such an inversion system to actual data.

**Appendix A. Setup of $\mathbf{B}^{notion}$**

The $\mathbf{B}^{notion}$ is a block diagonal matrix. The $i$th main diagonal block $\mathbf{B}_i$ represents the prior uncertainty covariance of the emissions for 12 months for a given region $i$. Assuming the relative error $\delta_i$ for $\mathbf{x}^b_i$ are the same for 12 months and $\mathbf{x}^b_{i,m}$ is the emission for region $i$ and month $m$ ($m=1$ means January, $m=12$ means December), so that the diagonal entries of the $\mathbf{B}^{notion}$ are:

$$\mathbf{B}_{i,(m,m)} = (\delta_i \mathbf{x}^b_{i,m})^2 \tag{A-1}$$

The assumed 2-month temporal autocorrelations (Sect. 2.2.2), expressed by an exponential decaying function, leads to the non-diagonal entries in $\mathbf{B}_i$. Accordingly, the covariance between the uncertainties in the emissions of 2 months (month $m$ and $n$, for instance) to be:

$$\mathbf{B}_{i,(m,n)} = e^{-\frac{|n-m|}{2}} \times (\delta_i \mathbf{x}^b_{i,m}) \times (\delta_i \mathbf{x}^b_{i,n}) \tag{A-2}$$

If region $i$ and region $j$ are within the same country, the off-diagonal block $\mathbf{B}_{i,j}$ is built to account for the spatial correlation

between these two regions. We assume that $\delta_i = \delta_j = \delta_{ij}$ and the spatial correlation between this two regions for a given month $m$ is -0.2 to account for fact that present emission estimates at such scales are generally disaggregated from national inventories, that is:

$$\mathbf{B}_{i,j,(m,m)} = -0.2 \times (\delta_{ij} \mathbf{x}^b_{i,m}) \times (\delta_{ij} \mathbf{x}^b_{j,m}) \tag{A-3}$$

We assume that the correlation between two control variables are given by the product of the spatial and temporal

correlations between the two corresponding control regions and the two months respectively. At last, the $\delta$ for each region are determined so that the prior annual emission uncertainty is satisfied, i.e. 10% for US, 10% for European countries, 15% for





China and 10% for other large regions.

## Appendix B. Setup of $\mathbf{B}^{empiric}$

The $\mathbf{B}^{empiric}$ is also a block diagonal matrix. For a given region $i$ and a specific month $m$, assuming the prior control parameter corresponding to PKU-$CO_2$ emission is $\mathbf{x}^b_{i,m}$, the "true" value of $\mathbf{x}$, corresponding to IER-EDG writes $\mathbf{x}^t_{i,m}$, so that the errors of the prior monthly emissions are:

$$\Delta\mathbf{x}_{i,m} = \mathbf{x}^t_{i,m} - \mathbf{x}^b_{i,m} \tag{B-1}$$

The long-term error component at annual scale $\varepsilon_{ann}$ equals:

$$\varepsilon_{ann} = \frac{1}{12}\sum_{m=1}^{12}\Delta\mathbf{x}_{i,m} \tag{B-2}$$

The residues are:

$$\mathbf{r}^{ann}_{i,m} = \Delta\mathbf{x}_{i,m} - \varepsilon_{ann} \tag{B-3}$$

Then the 6-month variation $\varepsilon_{6m}$ equals the standard deviation of the 6-month mean residues:

$$\varepsilon_{6m} = SD \text{ of } (\frac{1}{6}\sum_{m=1}^{6}\mathbf{r}^{ann}_{i,m}, \frac{1}{6}\sum_{m=7}^{12}\mathbf{r}^{ann}_{i,m}) \tag{B-4}$$

Again, the residues become:

$$\mathbf{r}^{6m}_{i,m} = \Delta\mathbf{x}_{i,m} - \varepsilon_{ann} - \frac{1}{6}\sum_{m=1}^{6}\mathbf{r}^{ann}_{i,m} \qquad (\text{if } m <= 6)$$

$$\mathbf{r}^{6m}_{i,m} = \Delta\mathbf{x}_{i,m} - \varepsilon_{ann} - \frac{1}{6}\sum_{m=7}^{12}\mathbf{r}^{ann}_{i,m} \qquad (\text{if } m >= 7) \tag{B-5}$$

In the same way, the 3-month variation $\varepsilon_{3m}$ equals the standard deviation of the 3-month mean residues:

$$\varepsilon_{3m} = SD \text{ of } (\frac{1}{3}\sum_{m=1}^{3}\mathbf{r}^{6m}_{i,m}, \frac{1}{3}\sum_{m=4}^{6}\mathbf{r}^{6m}_{i,m}, \frac{1}{3}\sum_{m=7}^{9}\mathbf{r}^{6m}_{i,m}, \frac{1}{3}\sum_{m=10}^{12}\mathbf{r}^{6m}_{i,m}) \tag{B-6}$$

And the corresponding residues:

$$\mathbf{r}^{3m}_{i,m} = \Delta\mathbf{x}_{i,m} - \varepsilon_{ann} - \frac{1}{6}\sum_{m=1}^{6}\mathbf{r}^{ann}_{i,m} - \frac{1}{3}\sum_{m=1}^{3}\mathbf{r}^{6m}_{i,m} \qquad (\text{if } m \leq 3)$$

$$\mathbf{r}^{3m}_{i,m} = \Delta\mathbf{x}_{i,m} - \varepsilon_{ann} - \frac{1}{6}\sum_{m=1}^{6}\mathbf{r}^{ann}_{i,m} - \frac{1}{3}\sum_{m=4}^{6}\mathbf{r}^{6m}_{i,m} \qquad (\text{if } 4 \leq m \leq 6)$$

$$\mathbf{r}^{3m}_{i,m} = \Delta\mathbf{x}_{i,m} - \varepsilon_{ann} - \frac{1}{6}\sum_{m=7}^{12}\mathbf{r}^{ann}_{i,m} - \frac{1}{3}\sum_{m=7}^{9}\mathbf{r}^{6m}_{i,m} \qquad (\text{if } 7 \leq m \leq 9)$$

$$\mathbf{r}^{3m}_{i,m} = \Delta\mathbf{x}_{i,m} - \varepsilon_{ann} - \frac{1}{6}\sum_{m=7}^{12}\mathbf{r}^{ann}_{i,m} - \frac{1}{3}\sum_{m=10}^{12}\mathbf{r}^{6m}_{i,m} \qquad (\text{if } 10 \leq m \leq 12) \tag{B-7}$$

The 1-month variation $\varepsilon_{1m}$ equals the standard deviation of these residues:




$$\varepsilon_{1m} = SD(r_{i,m}^{3m})$$
(B-8)

Using such a decomposition, the root mean square of the errors (RMSE) between the prior and the "true" values $\Delta x_{i,j}$ satisfy the following equation:

$$RMSE_i = \frac{1}{12}\sum_{m=1}^{12}\Delta x_{i,m}^2 = \varepsilon_{ann}^2 + \varepsilon_{6m}^2 + \varepsilon_{3m}^2 + \varepsilon_{1m}^2$$
(B-9)

Finally, for the diagonal entries of the **B** matrix corresponding to the monthly emissions of region *i*, they are equal to the RMSE_i, for the non-diagonal entries, the covariance between month *j* and month *k* for a given region is expressed as the sum of the products of the different variations multiplied by corresponding correlations (expressed by exponential decay functions) at different time scales:

$$\mathbf{B}_{i,(m,n)} = \varepsilon_{ann}^2 + \varepsilon_{6m}^2 + \varepsilon_{3m}^2 + \varepsilon_{1m}^2 \quad (if\ m=n)$$

$$\mathbf{B}_{i,(m,n)} = \varepsilon_{ann}^2 + e^{-\frac{|n-m|}{6}}\varepsilon_{6m}^2 + e^{-\frac{|n-m|}{3}}\varepsilon_{3m}^2 + e^{-\frac{|n-m|}{1}}\varepsilon_{1m}^2 \quad (if\ m\neq n)$$
(B-10)

**Appendix C. Calculation of trends and corresponding uncertainties**

Assuming the linear trend of the FFCO$_2$ emissions in an *n*-year period is to be calculated, which satisfies the function:

$$\mathbf{y} \approx \tilde{\mathbf{y}} = a\mathbf{x}+b$$
(C-1)

where *y* is the vector of annual emissions for the *n* years, *ỹ* is the predicted value by the regression and *x* is the vector of corresponding years, the slope *a* is the linear trend we are going to calculate by linear regression. We rewrite Eq. (C-1) as
follows:

$$\underbrace{\begin{bmatrix} y_1 \\ \vdots \\ y_{10} \end{bmatrix}}_{y} \approx \underbrace{\begin{bmatrix} y_1 \\ \vdots \\ y_{10} \end{bmatrix}}_{y} = \underbrace{\begin{bmatrix} x_1 & 1 \\ \vdots & \vdots \\ x_{10} & 1 \end{bmatrix}}_{\mathbf{X}} \underbrace{\begin{bmatrix} a \\ b \end{bmatrix}}_{p}$$
(C-2)

Thus the linear trend *a* and the interception *b* can be solved using linear algebra. With the notations used in Eq. (C-2), the result of the linear regression is:

$$\mathbf{p}=(\mathbf{X}^T\mathbf{X})^{-1}\mathbf{X}^T\mathbf{y}$$
(C-3)

the associated uncertainties in the regression parameters in vector *p* is thus given by the following covariance matrix:

$$cov(\mathbf{p})= (\mathbf{X}^T\mathbf{X})^{-1}\mathbf{X}^Tcov(\mathbf{Y})\mathbf{X}(\mathbf{X}^T\mathbf{X})^{-1}$$
(C-4)

where cov(.) is the covariance matrix for a set of variables.

Since **X** is a fixed matrix filled by the numbers of years and 1's, the uncertainties in the linear trend (first item in main diagonal of cov(*p*)), is independent of the annual emissions themselves but only depends on the uncertainties and associated



correlations of annual emissions. As sketched in Fig. C1, this error covariance of $y$ should include two independent parts: 1) the uncertainties associated with the estimation of the emissions for each year in $y$ and 2) the inter-annual variability (IAV) in the detrended $y$.

In this study, based on the time series of national annual emissions from IER-EDG, we assume a 5% IAV in the annual fossil fuel emissions for European countries. In general, this 5% IAV is the upper limit of the typical values for European
countries (Levin and Rödenbeck, 2007). Ballantyne et al. (2015) assumed that in the self-reported fossil fuel emission inventories, the emission error in one year could be highly correlated with the error from the previous year by an autoregressive coefficient of 0.95, due to potential errors that are not corrected retroactively after about 20 years. However, we do not conduct a multi-year inversion to get a typical estimate of the correlations in the posterior uncertainties in annual emissions, and assume that there is no correlations between the posterior uncertainties in annual emissions. This assumption is fairly conservative,
since Eq. (C-4) implies that the larger (either positive or negative) the correlations between the estimation of fossil fuel emissions from different years, the smaller the uncertainties in the regressed trends.

*Acknowledgement*. The authors acknowledge the support of the French Commissariat à l'énergie atomique et aux énergies alternatives (CEA). This study is co-funded by the European Commission under the EU Seventh Research Framework
Programme (grant agreement no. 283080, geocarbon project). G. Broquet and F. Vogel acknowledge funding from the industrial chair BridGES (supported by the Université de Versailles Saint-Quentin-en-Yvelines, the Commissariat à l'Energie Atomique et aux Energies Renouvelables, the Centre National de la Recherche Scientifique, Thales Alenia Space and Veolia). We are also grateful to Ingeborg Levin for the useful discussions on this topic. We also would like to thank the partners of the ICOS infrastructure for details of radiocarbon samplings and $FFCO_2$ monitoring.

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



**Table 1** Setup and performance indicators of the two types of inversions

| Input of inversions | INV-N | INV-E | Performance indicators |
|---|---|---|---|
| $\mathbf{B}$ | $\mathbf{B}^{\text{notion}}$ | $\mathbf{B}^{\text{empiric}}$ | $\mathbf{A}$ (Eq. (1)) |
| $\mathbf{R}$ | $\mathbf{R}_i + \mathbf{R}_r + \mathbf{R}_t + \mathbf{R}_a$ | $\mathbf{R}_i + \mathbf{R}_r + \mathbf{R}_t + \mathbf{R}_a$ | UR (Eq. (3)) |
| $\mathbf{H}$ | $\mathbf{H}^{\text{prac}}$ | $\mathbf{H}^{\text{prac}}$ | |
| $\mathbf{x}^t$ | $\mathbf{x}^{\text{IER-EDG}}$ | $\mathbf{x}^{\text{IER-EDG}}$ | $\mathbf{x}^a - \mathbf{x}^t$ (Eq. (2)) |
| $\mathbf{x}^b$ | $\mathbf{x}^{\text{PKU}}$ | $\mathbf{x}^{\text{PKU}}$ | MR (Eq. (4)) |
| $\mathbf{y}_o$ | $\mathbf{H}^{\text{OSSE}}\mathbf{x}^t + \varepsilon_i + \varepsilon_r + \varepsilon_t$ | $\mathbf{H}^{\text{OSSE}}\mathbf{x}^t + \varepsilon_i + \varepsilon_r + \varepsilon_t$ | |






**Table 2** Notations for the eight OSSEs.

|  | Number of synthetic data | INV-N | INV-E |
|---|---|---|---|
| NET17, 2-week sampling | 416 | N-17W | E-17W |
| NET43, 2-week sampling | 1092 | N-43W | E-43W |
| NET233, 2-week sampling | 6032 | N-43D | E-43D |
| NET43, 1-day sampling | 15288 | N-233W | E-233W |



**Table 3** Uncertainties in the regressed linear trends as a function of the posterior uncertainty in annual emissions. The uncertainties in the trends are defined as the ratio between the uncertainties in the linear regression slope of absolute annual emissions and the annual emission budget in the base year.

| Relative posterior uncertainty in annual emissions | 10-year trend | 20-year trend |
|---|---|---|
| 10% | 1.2% $yr^{-1}$ | 0.43% $yr^{-1}$ |
| 5% | 0.78% $yr^{-1}$ | 0.27% $yr^{-1}$ |
| 1% | 0.56% $yr^{-1}$ | 0.20% $yr^{-1}$ |



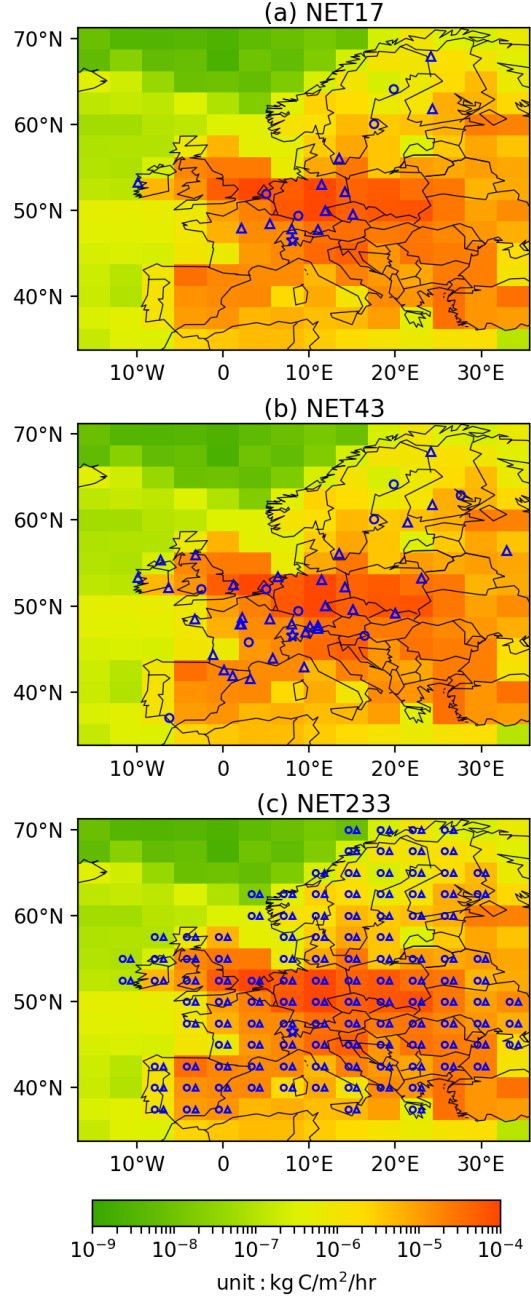

**Figure 1:** Site locations for the three continental network configurations used in this study: (a) NET17, (b) NET38, and (c) NET232. Circles correspond to "urban" sites and upper triangles are "rural" sites. Urban and rural sites are categorized according to the population density of the grid cells within which the stations are located and according to the locations of large point sources. The background color map is the annual FFCO$_2$ emissions in 2007 at the resolution of LMDZv4 from the PKU-CO$_2$ inventory (Wang et al., 2013).



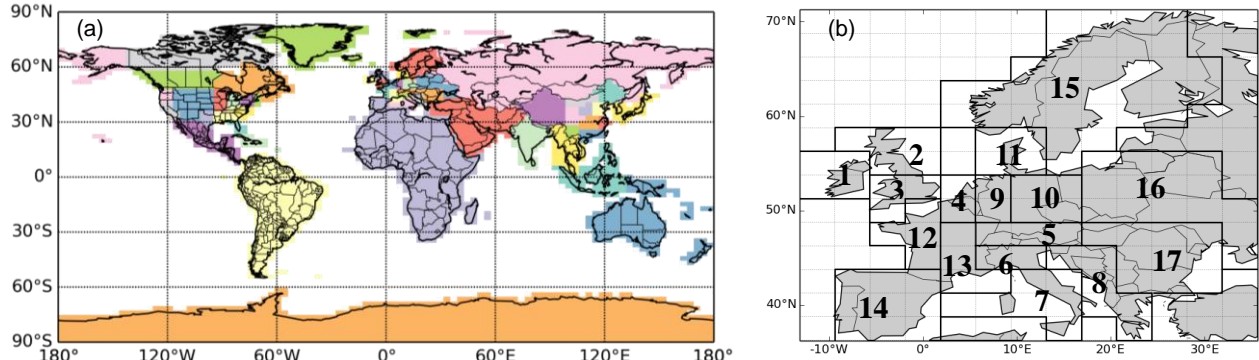

**Figure 2:** (a) Map of the 56 regions whose monthly emission budgets are controlled by the inversion; (b) zoom over the 17 control regions in Europe






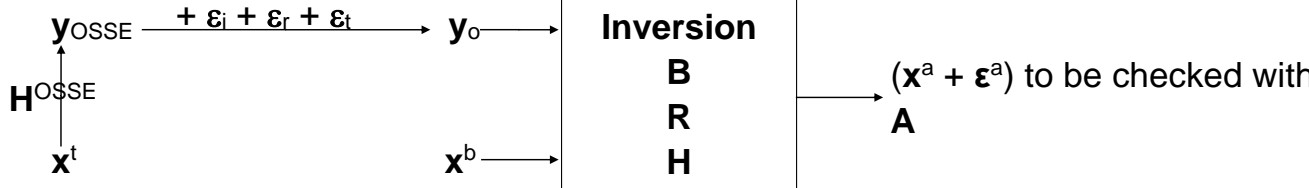

**Figure 3:** Schematic of the OSSEs



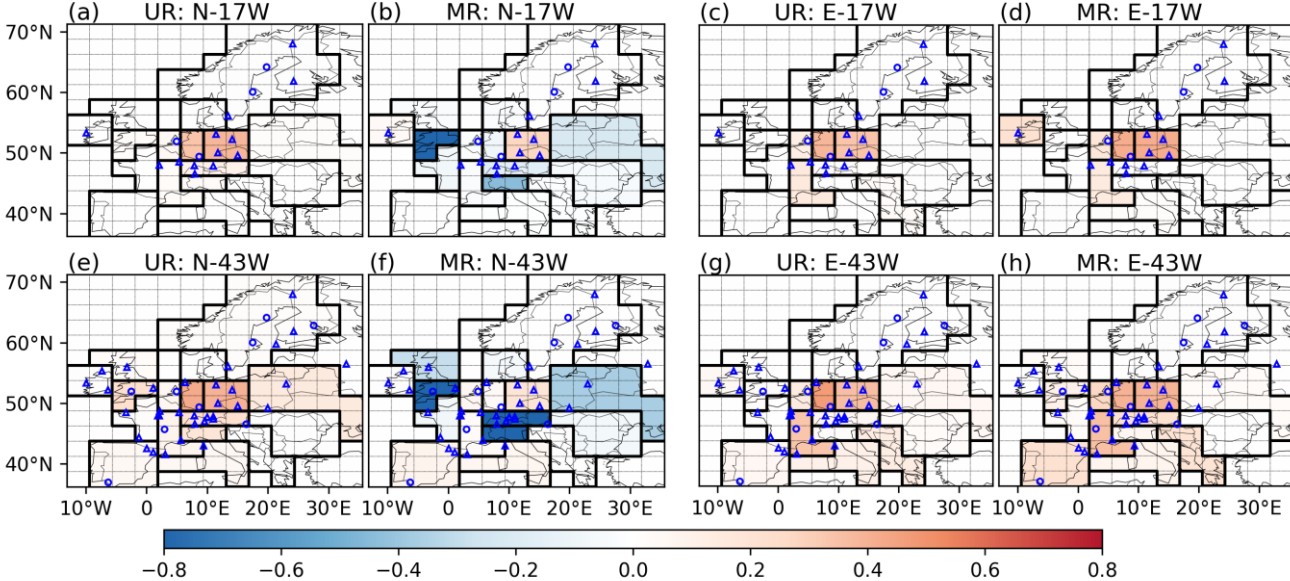

**Figure 4:** Average monthly uncertainty reductions and misfit reductions in FFCO$_2$ emissions over regions delineated by solid black lines, using the NET17 and NET43 networks and 2-week sampling for the inversions. The first and second columns are the results of INV-N inversions. The third and fourth columns are the results of INV-E inversions. The dashed lines show the grid cells of the transport model LMDZv4. The dots and triangles are the locations of the observation sites where the gradients are extracted with respect to the JFJ reference site. Dots (triangles) correspond to "urban" (or "rural") stations defined in Sect. 2.1. A value of UR and MR closer to unity means a better performance of an inversion to constrain FFCO$_2$ emissions in a region.





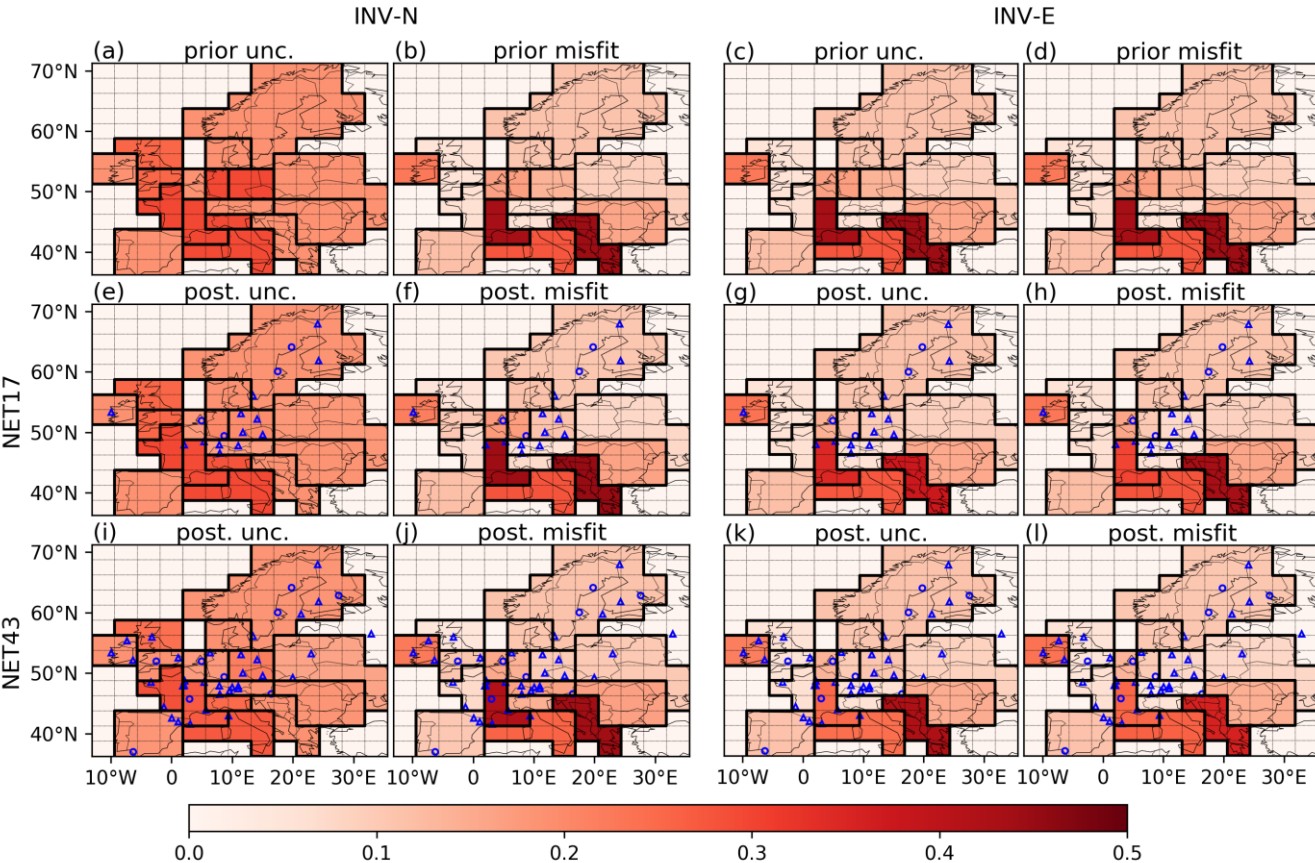

**Figure 5:** Average monthly relative prior and posterior uncertainties and misfits of FFCO$_2$ emissions over regions delineated
by black lines, using the NET17 and NET43 networks and 2-week sampling for INV-N (first and second columns) and INV-E
(third and fourth columns) inversions. First row shows the relative prior uncertainties and misfits. The second row shows the
posterior uncertainties and misfits after assimilating 2-week mean afternoon observations from network NET17. The third row
shows the posterior uncertainties and misfits after assimilating 2-week mean afternoon observations from network NET43.
The dashed lines show the grid cells of the transport model LMDZv4. The dots and triangles are the locations of the observation
sites where the gradients are extracted with respect to the JFJ reference site. Dots (triangles) correspond to "urban" (or "rural")
stations defined in Sect. 2.1.





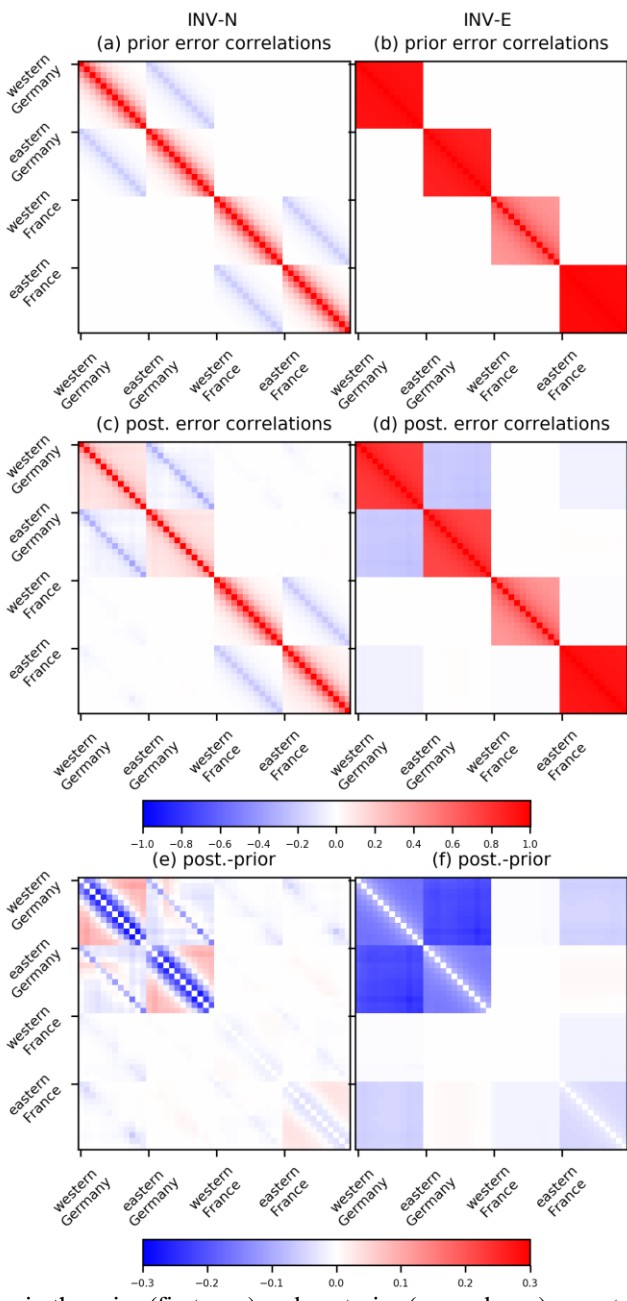

**Figure 6:** The correlation structure in the prior (first row) and posterior (second row) uncertainties in monthly regional FFCO$_2$ emissions for the four Germany and France regions using the NET43 network and 2-week sampling for INV-N (first column) and INV-E (second column) inversions, as well as their differences (third row). The $x$ and $y$ axes cover all the control region-months iterating through region first and months second (the blocks of pixels in each matrix). For clarity, we group these correlations into four regions and organize them for each region according to month indices.



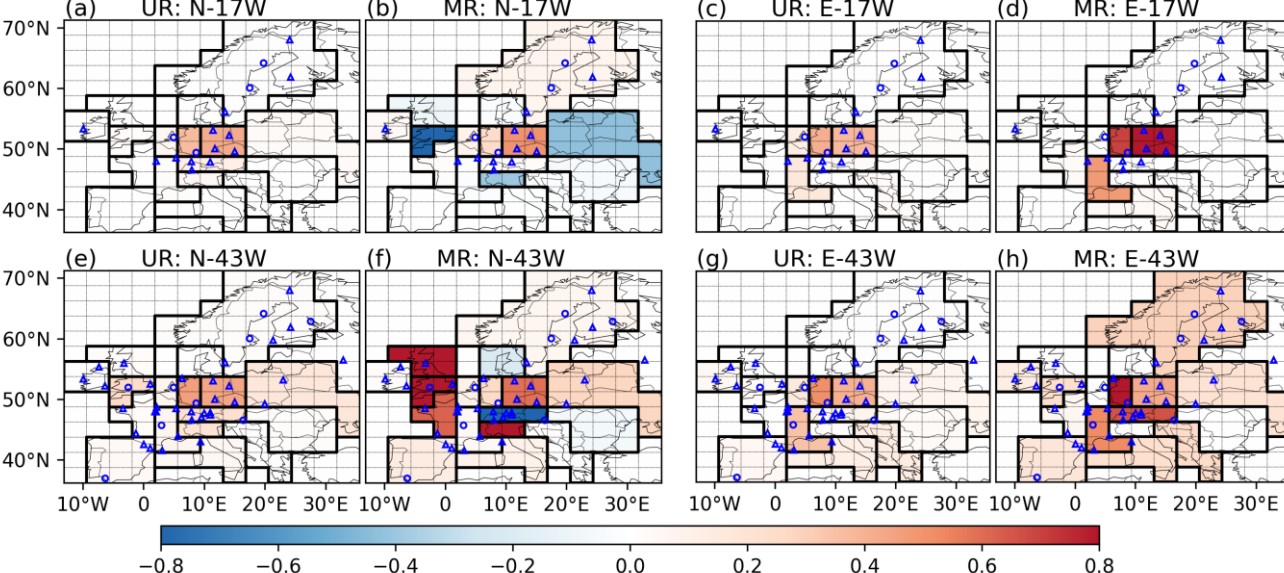

**Figure 7:** Uncertainty reduction (UR) and misfit reduction (MR) of annual $FFCO_2$ emissions over regions delineated by black lines using the NET17 and NET43 networks and 2-week sampling. The first and second columns show the results of INV-N inversions. The third and fourth columns show the results of INV-E inversions. The dashed lines show the grid cells of the transport model LMDZv4. The dots and triangles denote the locations of the observation sites where the gradients are extracted

with respect to the JFJ reference site. Dots (triangles) correspond to "urban" (or "rural") stations defined in Sect. 2.1. A value of UR and MR closer to unity means a better performance of an inversion to constrain $FFCO_2$ emissions in a region.





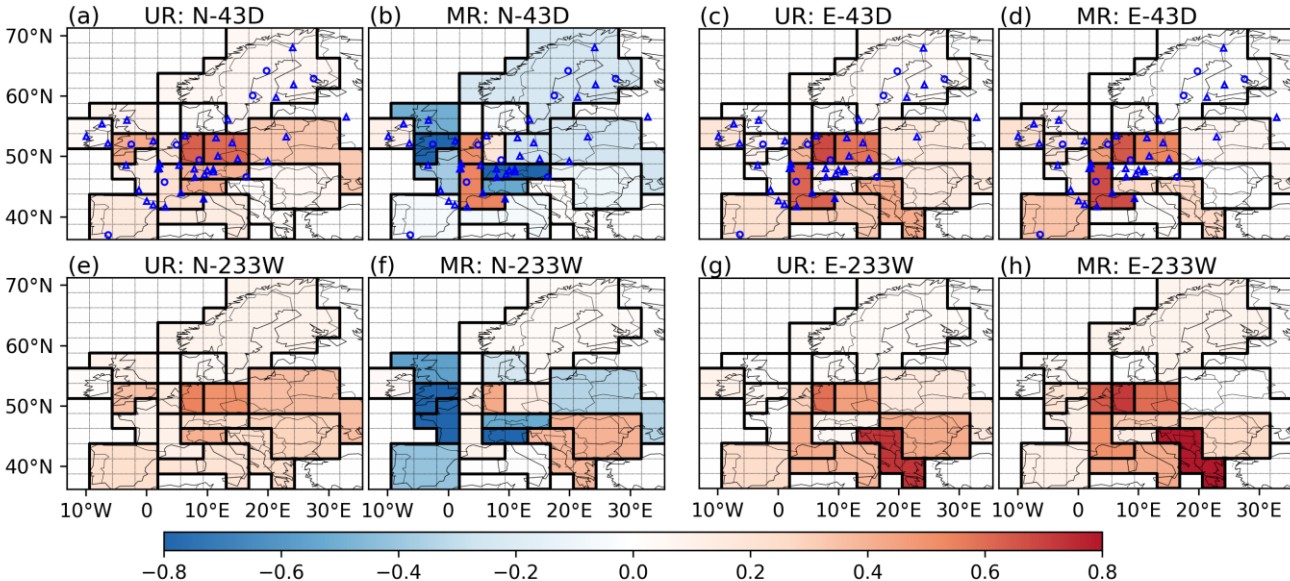

**Figure 8:** Average uncertainty and misfit reductions in the monthly FFCO$_2$ emissions over regions delineated by black lines using the NET43 network with daily sampling and NET233 network with 2-week sampling. The first and second columns are the results of INV-N inversions. The third and fourth columns are the results of INV-E inversions. The dashed lines show the grid cells of the transport model LMDZv4. The dots and triangles are the locations of the observation sites where the gradients are extracted with respect to the JFJ reference site. Dots (triangles) correspond to "urban" (or "rural") stations defined in Sect. 2.1. The locations of the sites in the OSSEs N-233W and E-233W are not plotted to avoid blurring the maps.




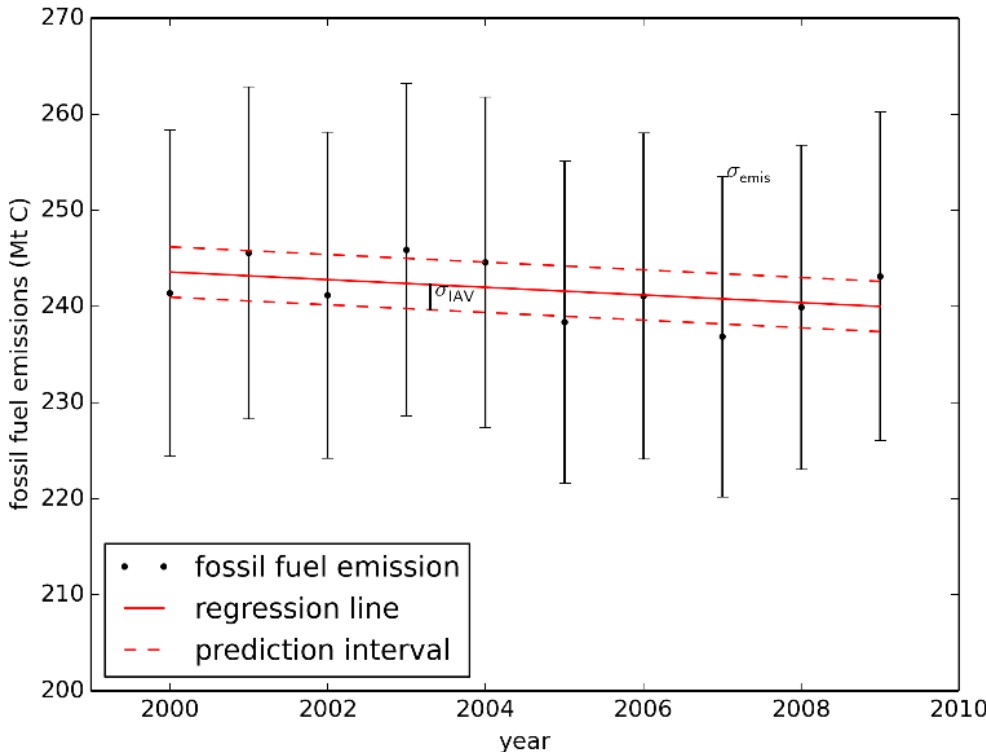

**Figure C1:** Annual FFCO$_2$ emissions from Germany in the period 2000-2009 calculated from IER-EDG.