# Peer review of "Potential of European 14CO2 observation network to estimate the fossil fuel CO2 emissions via atmospheric inversions"

_Atmospheric Chemistry and Physics, 2017_

## Referee Comment (RC1) · P. Rayner (Referee) · 30 Sep 2017

article [authoryear,round]natbib times

**General Comments**

This paper lays out the potential for current and future $^{14}CO_2$ observations to improve estimates of fossil fuel emissions in Europe. It uses two types of Observing System Simulation Experiments (OSSEs) based on either the theoretical uncertainty reduction

for a well-tuned case or a more realistic case where prior uncertainties do not match differences between prior and truth. It also uses several versions of an observing network ranging from the current network to a saturated case where every grid cell in the target domain is sampled. Results are not very surprising with the current network offering useful information at the conjunction of dense networks and high emissions (and concomitant uncertainties) with the case improving as networks become more dense. results are, however, sensitive to the proper tuning of prior covariances; a salutary result the authors are right to emphasise. The paper addresses an important problem with reasonable if not state-of-the-art tools, is clearly written and within scope.

I have two concerns about the paper, one general and one specific. the authors note the dependence of their results on the resolution of their transport model ($3.75 \times 2.5°$) but I think should do more to evaluate this. It is unlikely that anyone would use this resolution for an inversion of fossil fuel emissions targeting Europe and the guidance on network density is hard to generalise. The authors can help a little here since their group has access to higher resolution models. How much do the representation and aggregation errors change with increasing model resolution. Representation error probably decreases while aggregation error increases but how much? Increased resolution makes gaps in the network inevitable, what effect will they have? this could be tested by a couple of systematic thinning experiments on the saturated network case here.

My other concern is for this saturated case. As I understand it, each grid cell is oversampled with two measurements. If this is the case and the transport Jacobians for the two measurements are the same then I think the two measurements can be combined into a single measurement by summing their information content. There should also be strong correlation between the two measurements in the same grid cell, accounting for large-scale errors in the transport model. In particular, I think that the relationship between the aggregation and representation errors for the two types of site is complex, interesting and perhaps important. It is quite possible that using both types of site

reduces the sampling inhomogeneity necessary for aggregation errors (Trampert and Snieder, 1996; Kaminski et al., 2001).

**References**

Kaminski, T., Rayner, P. J., Heimann, M., and Enting, I. G.: On Aggregation Errors in Atmospheric Transport Inversions, J. Geophys. Res., 106, 4703–4715, 2001.

Trampert, J. and Snieder, R.: Model Estimations Biased by Truncated Expansions: Possible Artifacts in Seismic Tomography, Science, 271, 1257–1260, 1996.

---

## Referee Comment (RC2) · Anonymous Referee #2 · 10 Oct 2017

The authors present an OSSE study of the capability of ICOS $^{14}CO_2$ observations to constrain European fossil fuel $CO_2$ fluxes and their trends. The study is well structured and should be published. I have a few comments which I'd like the authors to address before publication.

**Major comments**

1. **Line 112:** The assumption that $^{14}CO_2$ measurements can be accurately translated into FF $CO_2$, i.e., there are no spatial patterns introduced due to the other

terms (especially disequilibrium and nuclear plants in the European context), is a big one. Those terms will not only affect annual emission estimates, but also the ability to detect trends, as countries change their nuclear power generation capacity and switch to wood-fired domestic heating (e.g., Germany). I understand that modeling the full $^{14}CO_2$ budget is beyond the scope of the authors' framework, but it should be possible to estimate the impact, by e.g. modelling just the nuclear or disequilibrium contribution as a tracer in a transport model and looking at the change in $\Delta^{14}C$. Have the authors done that? Unless that concern is addressed, the actual numbers from the manuscript are hard to trust.

2. **Line 145:** The authors say that the inversion interpret the gradient between JFJ and other sites. I do not understand how that is implemented. Is it that JFJ is the only background site in the network, and hence the inversion implicitly interprets gradients w.r.t. JFJ (much as a global $CO_2$ inversion might interpret everything w.r.t. MLO and SPO)? Or is it that the pseudo-obs are fed in after explicitly subtracting the JFJ time series, in which case the model's observation operator looks like "site – JFJ" at each individual site? Basically, the authors say in words that they interpret the gradient w.r.t. JFJ, but I do not understand how that is implemented in practice.

3. **Line 201:** I'm having trouble deciphering the meaning of "mismatch reduction", and its bounds and limits. Instead of describing it in words after equation (4), could the authors please write down the mathematical expressions for $\varepsilon_a$ and $\varepsilon_b$? Since I did not know what those $\varepsilon$'s were, I also could not interpret maps of MR (e.g., Figure 4). In particular, I did not understand what negative vs positive MR meant.

**Minor comments**

1. **Line 151:** For NET233, each grid box is supposed to have one urban and one rural site. I'm not sure that's a good strategy. Wouldn't it be better to designate urban/rural depending on the nearest NET43 site? I mean, there could easily be grid boxes where it was more realistic to put two rural or two urban sites.

2. **Line 233:** In the ICOS protocol, are the two-week samples going to be filled continuously, or are they only going to integrate mid-afternoon (or nighttime) air? That would very much change the sensitivity of the observations to FF $CO_2$, and the impact of transport errors.

3. **Line 237:** Are the authors assuming that two week average $\Delta^{14}CO_2$ will translate into two week average FF $CO_2$? What the two week average $\Delta^{14}CO_2$ represents depends on the method of collection; an open tray will fix $CO_2$ proportional to the partial pressure of $CO_2$, while a bubbled trap will fix all the $CO_2$ in the ingested air. The former represents average FF $CO_2$ weighted by the total $CO_2$ mole fraction, while the latter represents average FF $CO_2$ over two weeks. Which one applies for the ICOS protocol?

4. **Line 253:** Did the authors model a diurnal cycle in FF $CO_2$ emissions? According to Nassar et al. (2013), the diurnal cycle can be fairly large over populated areas. Along with the selective mid-afternoon sampling used by the authors, the impact could be sizeable.

5. **Line 264:** Does "practical" refer to the operator used to generate pseudo-observations from the "true" fluxes?

6. **Line 296:** Is the covariance model global, or is this done only over Europe?

7. **Line 363:** Can the authors explain how they obtained the IER hourly inventory? I tried to download it from their website, but given that each month had to be separately downloaded, that was very inconvenient. An email to the contact person listed on the website bounced, so that was a dead end too.

8. **Line 366:** This will only get at the random error in transport, not any systematic error in transport modeling. Have the authors tried quantifying the impact of systematic errors in LMDZ, say by using $^{222}Rn$ or $SF_6$?

9. **Line 367:** I know it is usual practice in the OSSE world to perturb the measurements according to the error statistics of $R$, but I have never understood why, unless it is done in an ensemble of multiple realizations of the measurements. In an ensemble of inversions with different measurements from the same network, it makes perfect sense to produce those measurements using perturbations according to $R$, since the resulting spread in the flux estimates then gives the uncertainty due to $R$. However, for a single inversion, perturbing the measurements according to $R$ only ensures that the posterior will be different from the "true" flux, without any way to infer the significance of that difference. As in, how do the authors know that the MR's they estimate are not because in the one realization of the measurements they used, some of them just happened to be skewed in one direction? This is especially a concern for the NET17 network, since there are so few measurements, with scant opportunity to average over the perturbations.

10. **Line 374 and Figure 4:** The authors solve for monthly emissions over a year, but report a single UR/MR map of monthly emissions. Is this the RMS of UR/MR values over 12 months, or the UR/MR calculated from the RMS of the posterior errors, or...? As in, can the authors give a mathematical expression of what is being shown in Figure 4 as the "monthly" UR/MR, in terms of their control vector and/or covariance matrix?

11. **Line 381:** I think the reference to Figure 4(d) should actually be to 4(e). Likewise,

in line 384, the refences should be to 4(e) and 4(g).

12. **Line 409:** "... the posterior misfits are even larger than the prior misfits." Why does the inversion allow this? For stations within the blue regions, is this obvious from looking at the atmospheric FF $CO_2$ time series, that post-optimization the time series is further away from the pseudo-data than pre-optimization? I suspect the perturbed measurements are to blame (see earlier comment).

13. **Line 417:** The correlation is not between uncertainties, but between corrections from the prior emission.

14. **Line 455:** I'm surprised at the low UR for the NET233 network. Why are there so many white areas (low UR) still?

15. **Line 484:** In real trend detection situations, the transport will vary year by year, as will the disequilibrium and nuclear fluxes of $^{14}C$. Can the authors estimate how big an impact this will have on the trend detection?

16. **Line 499:** As far as I can tell, Basu et al. (2016) did not estimate UR.

17. **Line 531:** The authors seem to suggest that the boundary condition – a bane of most regional inversions – does not affect their flux and uncertainty estimates. Is this because everything is referenced to JFJ?

18. **Eqs. C-3 and C-4:** I believe there are errors in these two formulae. If $\vec{p}$ is obtained by minimizing the cost function $J$

$$J = \frac{1}{2} \left( \vec{y} - \tilde{y} \right)^T R^{-1} \left( \vec{y} - \tilde{y} \right) \tag{1}$$

where $R = cov(\vec{y})$ ($R$ in their case contains the posterior error estimates on fluxes), then the optimal estimate of $\vec{p}$ and the corresponding covariance are

$$\vec{p} = \left(X^T R^{-1} X\right)^{-1} X^T R^{-1} \vec{y}$$
$$cov(\vec{p}) = \left(X^T R^{-1} X\right)^{-1} X^T R^{-1} X \left(X^T R^{-1} X\right)^{-1}$$

19. **Table 2:** In columns 2 and 3, I believe rows 3 and 4 have been flipped.

20. **Figure 2(a):** Am I supposed to see 56 colors in the world map? I don't. I think the problem is that the country and state boundaries overlap with the region boundaries. I would suggest, at least in the world map, only showing the region boundaries from the control vector and eliminating the country and state boundaries.

**References**

Basu, S., Miller, J. B., and Lehman, S.: Separation of biospheric and fossil fuel fluxes of $CO_2$ by atmospheric inversion of $CO_2$ and $^{14}CO_2$ measurements: Observation System Simulations, Atmos. Chem. Phys., 16, 5665–5683, doi:10.5194/acp-16-5665-2016, http://www.atmos-chem-phys.net/16/5665/2016/, 2016.

Nassar, R., Napier-Linton, L., Gurney, K. R., Andres, R. J., Oda, T., Vogel, F. R., and Deng, F.: Improving the temporal and spatial distribution of $CO_2$ emissions from global fossil fuel emission data sets, J. Geophys. Res. Atmos., 118, 917–933, doi:10.1029/2012JD018196, http://dx.doi.org/10.1029/2012JD018196, 2013.

---

## Author Comment (AC2) · 28 Nov 2017

The authors present an OSSE study of the capability of ICOS $^{14}CO_2$ observations to constrain European fossil fuel $CO_2$ fluxes and their trends. The study is well structured and should be published. I have a few comments which I'd like the authors to address before publication.

**Response:**

   We would like to thank the reviewer for the valuable comments and suggestions for improving our manuscript. Following the reviewer's comments, we will carefully revise our manuscript. Please find below the point-to-point responses (in black) to all referee comments (in blue). All the pages and line numbers correspond to the original versions of text.

**Major comments**

1. Line 112: The assumption that $^{14}CO_2$ measurements can be accurately translated into FF $CO_2$, i.e., there are no spatial patterns introduced due to the other terms (especially disequilibrium and nuclear plants in the European context), is a big one. Those terms will not only affect annual emission estimates, but also the ability to detect trends, as countries change their nuclear power generation capacity and switch to wood-fired domestic heating (e.g., Germany). I understand that modeling the full $^{14}CO_2$ budget is beyond the scope of the authors' framework, but it should be possible to estimate the impact, by e.g. modelling just the nuclear or disequilibrium contribution as a tracer in a transport model and looking at the change in $\Delta^{14}C$. Have the authors done that? Unless that concern is addressed, the actual numbers from the manuscript are hard to trust.

**Response:**

   Indeed, the modeling of the full $^{14}CO_2$ budget is beyond the scope of this study. It would be necessary to study the impact of uncertainties in other $^{14}CO_2$ fluxes on the $^{14}CO_2$ gradients, to precisely quantify the corresponding errors when converting the gradients of atmospheric $^{14}CO_2$ measurements into $FFCO_2$ gradients. Graven and Gruber (2011) estimated the sources of $^{14}C$ from nuclear power generation and spent fuel reprocessing and used the global TM3 transport model at $1.8°\times1.8°$ resolution, which is slightly higher than LMDZv4 used in our study, to simulate their continental-scale influences on $\Delta^{14}C$. Their results showed that nuclear enrichment may cause an impact of -0.9 [-0.6, -1.4] ppm in $FFCO_2$ for Orleans, France (48.8°N, 2.5°E) and an impact of -0.7 [-0.4, -1.3] ppm for Heidelberg if nuclear $^{14}C$ enrichment was not accounted for. These two sites are representative of European continental sites that are close to nuclear power plants. Turnbull et al. (2009) estimated the impact in large-scale $FFCO_2$ signals caused by ignoring other $^{14}C$ fluxes including cosmogenic production, $^{14}C$ disequilibrium between atmosphere and biosphere and between atmosphere and ocean, and $^{14}C$ source from nuclear power plant (the estimate of $^{14}C$ sources from nuclear power plants is not as accurate as Graven and Gruber, 2011), using the same transport model LMDZv4 as in this study. Turnbull et al. (2009) showed that impact caused by other $^{14}CO_2$ sources in translating atmospheric measurements of $^{14}CO_2$ into $FFCO_2$ is mainly from terrestrial biosphere, whereas the contributions from ocean $CO_2$ exchange and cosmogenic production of

[14]C contribute are weak. According to Turnbull et al. (2009), neglecting the influences from biosphere leads to an error typically between 0.2 and 0.8 ppm. Miller et al. (2012) estimated the impact of biospheric disequilibrium $^{14}$C fluxes in North America and get a similar value as Turnbull et al. (2009), ranging from less than 0.2 ppm to 1.4 ppm.

We work with OSSEs so that what matters in our system is the correct representation of the uncertainties in the different components of the model, not a correct representation of these components. In our OSSE framework, by ignoring the fluxes of $^{14}CO_2$ other than the dilution of $^{14}$C in $CO_2$ by fossil fuel emissions, we implicitly assume that the uncertainties in these fluxes has a weak impact, not that these fluxes themselves are ignored. The results mentioned above show that the uncertainties in the signals of these $^{14}CO_2$ fluxes may cause some impact on the interpretation of $FFCO_2$ using atmospheric $^{14}CO_2$ samples indeed, but also that this impact is below 1 ppm and thus smaller than the components of observation errors, e.g. measurement error (1 ppm), representation error (0.17-2.56 ppm) and transport error (0.52-4.15 ppm) in the paper. In this context, we assume that the influence of the uncertainties in $^{14}CO_2$ fluxes other than the dilution of $^{14}$C in $CO_2$ by fossil fuel emissions on the inversion of fossil fuel emission should be relatively weak. Our estimate of URs and MRs could be slightly over-estimated due to the ignorance of other sources of uncertainties from other $^{14}CO_2$ fluxes. But this study aims at understanding how the inversion system behaves when dealing with uncertainties in the $FFCO_2$ emissions versus observation errors, about how the variation of the UR as a function of regions, of level of emissions, of the density of networks, rather than about providing absolute values of UR that would apply when conducting real-data applications. Further investigations accounting for uncertainties in other $^{14}CO_2$ fluxes will be needed to refine those numbers.

We will better explain the assumptions underlying the conversion of $^{14}CO_2$ into $FFCO_2$ gradients with a 1 ppm uncertainty given that we work with an OSSE framework (i.e. it does not mean that the nuclear plant, cosmogenic and biosphere fluxes themselves have been ignored). We will add some discussions in Sect. 4.2 to recall the fact that the uncertainties associated with the signals from $^{14}CO_2$ fluxes other than the dilution of $^{14}$C in $CO_2$ by fossil fuel emissions are expected to be relatively small compared to the uncertainties in the atmospheric $^{14}CO_2$ caused by fossil fuel emissions and by other types of observation errors. We will also add some discussions in Sect. 4.2 to emphasize that this study aims at providing some understanding of the system behavior rather than a precise quantification of UR that would apply when conducting real data applications. Lastly, we will add in the manuscript that future studies will be needed to quantify the impact from uncertainties in the nuclear plant emissions, cosmogenic production, biogenic fluxes, etc., on the inversion of $FFCO_2$ emissions.

References:

Graven, H. D. and Gruber, N.: Continental-scale enrichment of atmospheric $^{14}CO_2$ from the nuclear power industry: potential impact on the estimation of fossil fuel-derived $CO_2$, Atmos. Chem. Phys., 11(23), 12339–12349, doi:10.5194/acp-11-12339-2011, 2011.

Turnbull, J., Rayner, P., Miller, J., Naegler, T., Ciais, P. and Cozic, A.: On the use of $^{14}CO_2$ as a tracer for fossil fuel $CO_2$: Quantifying uncertainties using an atmospheric transport model, J. Geophys. Res., 114(D22), doi:10.1029/2009jd012308, 2009.

Miller, J. B., Lehman, S. J., Montzka, S. A., Sweeney, C., Miller, B. R., Karion, A., Wolak, C., Dlugokencky, E. J., Southon, J., Turnbull, J. C. and Tans, P. P.: Linking emissions of fossil fuel $CO_2$ and other anthropogenic trace gases using atmospheric $^{14}CO_2$, J. Geophys. Res., 117(D8), doi:10.1029/2011jd017048, 2012.

2. Line 145: The authors say that the inversion interpret the gradient between JFJ and other sites. I do not understand how that is implemented. Is it that JFJ is the only background site in the network, and hence the inversion implicitly interprets gradients w.r.t. JFJ (much as a global CO2 inversion might interpret everything w.r.t. MLO and SPO)? Or is it that the pseudo-obs are fed in after explicitly subtracting the JFJ time series, in which case the model's observation operator looks like "site − JFJ" at each individual site? Basically, the authors say in words that they interpret the gradient w.r.t. JFJ, but I do not understand how that is implemented in practice.

**Response:**

Our implementation corresponds to the second option explained by the reviewer, i.e. the pseudo-observation are differences between the data at other sites and at JFJ for a given time, and the observation operator relates the fluxes to "site minus JFJ" gradients. To clarify this, we will revise the sentence in lines 160 and 249 to explicitly explain that the "observations" $\mathbf{y}_o$ are $FFCO_2$ gradients to JFJ.

3. Line 201: I'm having trouble deciphering the meaning of "mismatch reduction", and its bounds and limits. Instead of describing it in words after equation (4), could the authors please write down the mathematical expressions for $\varepsilon_a$ and $\varepsilon_b$? Since I did not know what those $\varepsilon$'s were, I also could not interpret maps of MR (e.g., Figure 4). In particular, I did not understand what negative vs positive MR meant.

**Response:**

The "misfits" (i.e. the "mismatch" in the reviewer's comment) that are considered for the "misfit reduction" are the differences between the prior or posterior and "true" estimates of the emission budgets. For a given region $i$ and month $m$ in the control vector $\mathbf{x}$, the prior misfit $\mathbf{x}^b_{i,m}-\mathbf{x}^t_{i,m}$ is denoted $\boldsymbol{\varepsilon}^b_{i,m}$ and the posterior misfit $x^a_{i,m}-x^t_{i,m}$ is denoted $\boldsymbol{\varepsilon}^a_{i,m}$. We compute a misfit reduction for each region-month emission budget as the relative difference between the prior and posterior misfits:

$$MR_{i,m}=1-\boldsymbol{\varepsilon}^a_{i,m}/\boldsymbol{\varepsilon}^b_{i,m}$$

We only have one practical realization for $\mathbf{x}^b$, $\mathbf{y}_o$ and $\mathbf{x}^a$ and thus a single realization of the misfits for each month and region. However, we want to have a statistical assessment of the performance of the inversion system based on the misfits that could be compared to the scores of uncertainty reduction. Therefore, we also consider the typical MR at the 1-month scale for a given region $i$ which is the relative difference between the quadratic mean of the monthly prior misfits and the quadratic mean of the monthly posterior misfits.

$$MR_i = 1 - \frac{\sqrt{\frac{1}{12}\sum_{m=1}^{12}\left(\varepsilon_{i,m}^a\right)^2}}{\sqrt{\frac{1}{12}\sum_{m=1}^{12}\left(\varepsilon_{i,m}^b\right)^2}}$$

In all cases MR values could theoretically range between –infinity and 1. When, on average, the posterior emission estimates are closer to the synthetic truth than the prior estimates, the MR is positive (the inversion reduces the misfits). Conversely, when the posterior emission estimates are further from the synthetic truth than the prior estimates, the MR is negative (the inversion increases the misfits). MR is null when the posterior misfits are as large as the prior misfits, i.e. the inversion do not decrease or increase the misfits. We will revise the sentences in line 209-213 and add two equations for the computation of UR and MR at monthly scale.

**Minor comments**

1. Line 151: For NET233, each grid box is supposed to have one urban and one rural site. I'm not sure that's a good strategy. Wouldn't it be better to designate urban/rural depending on the nearest NET43 site? I mean, there could easily be grid boxes where it was more realistic to put two rural or two urban sites.
**Response:**

The design of NET233 was not intended to be optimal. We wanted to conduct a test with many sites whose distribution would be homogeneous and cover all the control regions. In this case, the variations of UR from one region to the other one are a direct consequence of variations in the emission uncertainties, and not as a consequence of the variations in the network density. Putting most of the sites near the areas with the highest emissions could have larger URs than spreading them homogeneously across Europe, but will not distinguish the role of network density and the role of emissions uncertainties themselves in the variations of the UR from one region to the other one.

Even though the distribution of the emissions is highly heterogeneous in Europe, the grid cells of the transport model used in this study have a 3.75 °×2.5 °resolution (i.e. they cover areas that are much larger than megacities like London or Paris), so that we can assume that nearly all pixels have both rural and urban locations.

We will clarify the rational for the NET233 network in Sect. 2.1.

2. Line 233: In the ICOS protocol, are the two-week samples going to be filled continuously, or are they only going to integrate mid-afternoon (or nighttime) air? That would very much change the sensitivity of the observations to FF $CO_2$, and the impact of transport errors.
**Response:**

The present protocol for almost all of the ICOS $^{14}CO_2$ sites is to fill continuously two-week samples. And some sites fill continuously daily/weekly atmospheric $^{14}CO_2$. However, the option of intermittent filling of air samples is feasible in practice and has been used (Levin et al., 2008; Turnbull et al., 2016). On the other hand, the state-of-the-art transport models used for atmospheric inversion studies have still

difficulties in simulating the vertical mixing during night-time and in the morning. So we prefer to consider this option of two-week afternoon sampling in the OSSE. We keep in mind the fact that using samples filled continuously over two weeks instead of during the afternoon only would definitely change the transport condition and sensitivity of the observation to $FFCO_2$, and thus may give different values of UR and MR. But we assume that it should change the result in a quantitative but not qualitative way.

We will revise the sentence in lines 233-235 to mention that intermittent filling of air samples is practically feasible so that our definition of the observations in this study can be seen as a compromise between the current requirement of state-of-the-art inversion systems and current measurement practices.

References:

Turnbull, J. C., Keller, E. D., Norris, M. W. and Wiltshire, R. M.: Independent evaluation of point source fossil fuel $CO_2$ emissions to better than 10%, Proc. Natl. Acad. Sci. U. S. A., 113(37), 10287–10291, doi:10.1073/pnas.1602824113, 2016.

Levin, I., Hammer, S., Kromer, B. and Meinhardt, F.: Radiocarbon observations in atmospheric $CO_2$: determining fossil fuel $CO_2$ over Europe using Jungfraujoch observations as background, Sci. Total Environ., 391(2–3), 211–6, doi:10.1016/j.scitotenv.2007.10.019, 2008.

3. Line 237: Are the authors assuming that two week average $\Delta^{14}CO_2$ will translate into two week average FF CO2? What the two week average $\Delta^{14}CO_2$ represents depends on the method of collection; an open tray will fix $CO_2$ proportional to the partial pressure of $CO_2$, while a bubbled trap will fix all the $CO_2$ in the ingested air. The former represents average FF $CO_2$ weighted by the total $CO_2$ mole fraction, while the latter represents average FF $CO_2$ over two weeks. Which one applies for the ICOS protocol?

**Response:**

The sampling technology (https://www.icos-cal.eu/crl/radiocarbon_samples) that ICOS utilize for two-week integrated samples follows the method developed in the 1970s (Levin et al., 1980). The sampling system is equipped with a small aquarium pump, which actively collects about 15 $m^3$ of air during the two week period. $CO_2$ is collected by chemical absorption in $CO_2$-free NaOH solution in the so-called Raschig-tube samplers. In this context, the $FFCO_2$ represents the average $FFCO_2$ over two weeks (the second way that the reviewer mentioned).

We will clarify the fact that our definition of $FFCO_2$ corresponds to the average afternoon $FFCO_2$ over the course of two weeks here.

References:

Levin, I., Munnich, K. O., Weiss, W.: The effect of anthropogenic $CO_2$ and $^{14}C$ sources on the distribution of 14C in the atmosphere, Radiocarbon, 22, 379–391, 1980.

4. Line 253: Did the authors model a diurnal cycle in FF CO2 emissions? According to Nassar et al. (2013), the diurnal cycle can be fairly large over populated areas. Along with the selective mid-afternoon sampling used by the authors, the impact could be sizeable.

**Response:**

We fully account for the diurnal cycle of emissions in our computations. The synthetic truth of the OSSEs and the synthetic $FFCO_2$ observations are modelled using the emission map IER-EDG, which is an hourly emission products and has a clear diurnal cycle in the emissions. However, the emission map used to compute the observation operator is PKU-$CO_2$ version 1 (Wang et al., 2013), which is an annual product and which does not have temporal profiles, so that there is no diurnal/seasonal cycle in $\mathbf{H}_{distr}^{PKU}$. The mismatch between the temporal profiles of synthetic true emission and $\mathbf{H}_{distr}^{PKU}$ contribute to the so-called aggregation error (Wang et al., 2017). As shown in Table S3, the typical aggregation error is only 0.17-0.30 ppm, indicating that such difference only has a small impact on the simulated $FFCO_2$ signals. Furthermore, the inversion results indicate that the inversion can significantly reduce the uncertainties and misfits of the estimate of monthly emission budgets for large regions, even if using flat temporal profiles for the emissions while the truth is modeled with diurnal, weekly and seasonal temporal profiles for the emissions. One explanation is that the two-week mean afternoon samplings integrate the signal from both daytime and nighttime emissions across Europe due to the atmospheric transport.

References:

Wang, R., Tao, S., Ciais, P., Shen, H. Z., Huang, Y., Chen, H., Shen, G. F., Wang, B., Li, W., Zhang, Y. Y., Lu, Y., Zhu, D., Chen, Y. C., Liu, X. P., Wang, W. T., Wang, X. L., Liu, W. X., Li, B. G. and Piao, S. L.: High-resolution mapping of combustion processes and implications for $CO_2$ emissions, Atmos. Chem. Phys., 13(10), 5189–5203, doi:10.5194/acp-13-5189-2013, 2013.

Wang, Y., Broquet, G., Ciais, P., Chevallier, F., Vogel, F., Kadygrov, N., Wu, L., Yin, Y., Wang, R. and Tao, S.: Estimation of observation errors for large-scale atmospheric inversion of $CO_2$ emissions from fossil fuel combustion, Tellus B: Chemical and Physical Meteorology, 69(1), 1325723, doi:10.1080/16000889.2017.1325723, 2017.

**5. Line 264: Does "practical" refer to the operator used to generate pseudo-observations from the "true" fluxes?**

**Response:**

No, $\mathbf{H}^{prac}$ means this operator is a practical representation of the flux distribution and atmospheric transport that is used in the inversion system, while $\mathbf{H}^{OSSE}$ is the theoretical "true" operator used to generate pseudo-observations for OSSEs. We explained in line 264 the $\mathbf{H}^{prac} = \mathbf{H}_{samp}^{coloc} \mathbf{H}_{transp}^{LMDZ} \mathbf{H}_{distr}^{PKU}$ and we also wrote in line 364 "The synthetic observations are generated using $\mathbf{x}^{IER-EDG}$ and the operator $\mathbf{H}^{OSSE} = \mathbf{H}_{samp}^{coloc} \mathbf{H}_{transp}^{LMDZ} \mathbf{H}_{distr}^{IER-EDG}$, which relies on the same $\mathbf{H}_{samp}^{coloc}$ and $\mathbf{H}_{transp}$ operators as the $\mathbf{H}^{prac}$ observation operator used in the inversion system." So the difference between the practical operator $\mathbf{H}^{prac}$ and $\mathbf{H}^{OSSE}$ is their last sub-operators $\mathbf{H}_{distr}^{PKU}$ and $\mathbf{H}_{distr}^{IER-EDG}$, respectively. We will clarify such a use of the "practical" term to name $\mathbf{H}^{prac}$ in Sect. 2.2.2.

**6. Line 296: Is the covariance model global, or is this done only over Europe?**

**Response:**

This covariance model (which ignores the spatial correlations) is applied for the estimate of the prior uncertainty variance and temporal correlations for each of the

control regions over the globe. We will revise the sentence in lines 295-296 to emphasize the fact that it is applied to all regions over the globe, not only to regions in Europe.

**Response:**
We downloaded the "global fossil fuel emissions" under the "product" tab month by month. These files are hourly emissions for each month and for three emission heights. We summed the emissions at different emission heights together (our simulation of $FFCO_2$ assumes that all the emissions are emitted at surface). In this case, we obtained twelve files of global hourly emission fields for one year and each file corresponds to one month.

**Response:**
We prefer to consider that errors can have long spatial and temporal scales rather than a "systematic" component, since what the people usually call biases or systematic errors vary in time and can be difficult to predict and diagnose. The inversion system integrate such temporal and spatial consistency of the errors through the temporal and spatial correlations in **R**.

The assessment of the model errors over long temporal scales, e.g. by using real tracer data was out of the scope of this study and of that of Wang et al. (2017). It has been the topic of numerous model inter-comparison studies in the past including LMDZ such as Peylin et al. (2011) and Patra et al. (2011). Locatelli (2015) also studied in details the transport errors by LMDZ. These studies show that the skills of LMDZ are in line with state-of-the-art transport models and there is no evidence that LMDZ has a systematic error compared to other transport models. Furthermore, our aim here is not to assign transport errors for the specific transport model we use, but rather for a typical global transport model in order to produce general results.

The structure of the transport error is hard to assess for a specific transport models. Some studies have made some attempts to characterize it, but have come to different conclusions. For example, Lin and Gerbig (2005) determined the correlation timescale to be 2-3 hours for U-/V- winds. Broquet et al. (2011) analyzed the distribution of differences between the simulated and measured atmospheric mole fractions of $^{222}Rn$ (Radon). Temporal auto-correlations with 3 to 6-hour timescales are found in Broquet et al. (2011) based on such an analysis. Lauvaux et al. (2009) estimated the potential of the temporal correlations for transport error based on an ensemble of perturbations of the simulation of atmospheric transport. They found there are negative correlation between the atmospheric $CO_2$ mole fractions in the day

and the night and the correlations (of the errors) with a lag time of 1 day are close to 0 for midafternoon data (Fig. 4a in Lauvaux et al., 2009). Miller et al. (2015) investigated the magnitude of temporally covarying atmospheric transport errors and found that removing temporal covariances in the transport would underestimate the transport error at the monthly scale, but their study did not try to use any function to describe the temporal auto-correlations so that their results are difficult to generalize.

Most of these studies found temporal error correlations that are generally smaller than 1 day. If considering daily to monthly concentration averages, such a correlation should increase (if the transport error is a combination of error components at high and low frequencies). However, these various studies prove that the correlations in the transport error are challenging to estimate and the majority of existing inversion studies do not account for such potential correlations in atmospheric transport modeling (Peylin et al., 2013; Chevallier et al., 2010; Kadygrov et al., 2015; Peters et al., 2007; Gurney et al., 2008; Niwa et al., 2012). So, we keep the traditional assumption. We will add a discussion about the temporal auto-correlation of the transport error in Sect. 2.2.2, when introducing the configuration of transport error.

References:
Broquet, G., Chevallier, F., Rayner, P., Aulagnier, C., Pison, I., Ramonet, M., Schmidt, M., Vermeulen, A. T. and Ciais, P.: A European summertime $CO_2$ biogenic flux inversion at mesoscale from continuous in situ mixing ratio measurements, J. Geophys. Res., 116(D23), doi:10.1029/2011jd016202, 2011.

Chevallier, F., Ciais, P., Conway, T. J., Aalto, T., Anderson, B. E., Bousquet, P., Brunke, E. G., Ciattaglia, L., Esaki, Y., Fröhlich, M., Gomez, A., Gomez-Pelaez, A. J., Haszpra, L., Krummel, P. B., Langenfelds, R. L., Leuenberger, M., Machida, T., Maignan, F., Matsueda, H., Morguí, J. A., Mukai, H., Nakazawa, T., Peylin, P., Ramonet, M., Rivier, L., Sawa, Y., Schmidt, M., Steele, L. P., Vay, S. A., Vermeulen, A. T., Wofsy, S. and Worthy, D.: $CO_2$ surface fluxes at grid point scale estimated from a global 21 year reanalysis of atmospheric measurements, J. Geophys. Res., 115(D21), doi:10.1029/2010jd013887, 2010.

Gurney, K. R., Baker, D., Rayner, P. and Denning, S.: Interannual variations in continental-scale net carbon exchange and sensitivity to observing networks estimated from atmospheric $CO_2$ inversions for the period 1980 to 2005, Global Biogeochem. Cycles, 22(3), GB3025, doi:10.1029/2007GB003082, 2008.

Kadygrov, N., Broquet, G., Chevallier, F., Rivier, L., Gerbig, C. and Ciais, P.: On the potential of the ICOS atmospheric $CO_2$ measurement network for estimating the biogenic $CO_2$ budget of Europe, Atmos. Chem. Phys., 15(22), 12765–12787, doi:10.5194/acp-15-12765-2015, 2015.

Lauvaux, T., Pannekoucke, O., Sarrat, C., Chevallier, F., Ciais, P., Noilhan, J. and Rayner, P. J.: Structure of the transport uncertainty in mesoscale inversions of $CO_2$ sources and sinks using ensemble model simulations, Biogeosciences, 6(6), 1089–1102, 2009.

Locatelli, R., Bousquet, P., Hourdin, F., Saunois, M., Cozic, A., Couvreux, F., Grandpeix, J.-Y., Lefebvre, M.-P., Rio, C., Bergamaschi, P., Chambers, S. D., Karstens, U., Kazan, V., van der Laan, S., Meijer, H. A. J., Moncrieff, J., Ramonet, M., Scheeren, H. A., Schlosser, C., Schmidt, M., Vermeulen, A. and Williams, A. G.: Atmospheric transport and chemistry of trace gases in LMDz5B: evaluation and implications for inverse modelling, Geosci. Model Dev., 8(2), 129–150, doi:10.5194/gmd-8-129-2015, 2015.

Lin, J. C. and Gerbig, C.: Accounting for the effect of transport errors on tracer inversions, Geophys. Res. Lett., 32(1), L01802, doi:10.1029/2004GL021127, 2005.

Miller, S. M., Hayek, M. N., Andrews, A. E., Fung, I. and Liu, J.: Biases in atmospheric $CO_2$ estimates from correlated meteorology modeling errors, Atmos. Chem. Phys., 15(5), 2903–2914, 2015.

Niwa, Y., Machida, T., Sawa, Y., Matsueda, H., Schuck, T. J., Brenninkmeijer, C. A. M., Imasu, R. and Satoh, M.: Imposing strong constraints on tropical terrestrial $CO_2$ fluxes using passenger aircraft based measurements, J. Geophys. Res., 117(D11), D11303, doi:10.1029/2012JD017474, 2012.

Peters, W., Jacobson, A. R., Sweeney, C., Andrews, A. E., Conway, T. J., Masarie, K., Miller, J. B., Bruhwiler, L. M., Petron, G., Hirsch, A. I., Worthy, D. E., van der Werf, G. R., Randerson, J. T.,

Wennberg, P. O., Krol, M. C. and Tans, P. P.: An atmospheric perspective on North American carbon dioxide exchange: CarbonTracker, Proc. Natl. Acad. Sci. U. S. A., 104(48), 18925–30, doi:10.1073/pnas.0708986104, 2007.

Peylin, P., Houweling, S., Krol, M. C., Karstens, U., Rödenbeck, C., Geels, C., Vermeulen, A., Badawy, B., Aulagnier, C., Pregger, T., Delage, F., Pieterse, G., Ciais, P. and Heimann, M.: Importance of fossil fuel emission uncertainties over Europe for $CO_2$ modeling: model intercomparison, Atmos. Chem. Phys., 11(13), 6607–6622, doi:10.5194/acp-11-6607-2011, 2011.

Peylin, P., Law, R. M., Gurney, K. R., Chevallier, F., Jacobson, A. R. and co-authors 2013. Global atmospheric carbon budget: results from an ensemble of atmospheric $CO_2$ inversions. *Biogeosciences* **10**, 6699–6720.

Wang, R., Tao, S., Ciais, P., Shen, H. Z., Huang, Y., Chen, H., Shen, G. F., Wang, B., Li, W., Zhang, Y. Y., Lu, Y., Zhu, D., Chen, Y. C., Liu, X. P., Wang, W. T., Wang, X. L., Liu, W. X., Li, B. G. and Piao, S. L.: High-resolution mapping of combustion processes and implications for $CO_2$ emissions, Atmospheric Chemistry and Physics, 13(10), 5189–5203, doi:10.5194/acp-13-5189-2013, 2013.

9. Line 367: I know it is usual practice in the OSSE world to perturb the measurements according to the error statistics of R, but I have never understood why, unless it is done in an ensemble of multiple realizations of the measurements. In an ensemble of inversions with different measurements from the same network, it makes perfect sense to produce those measurements using perturbations according to R, since the resulting spread in the flux estimates then gives the uncertainty due to R. However, for a single inversion, perturbing the measurements according to R only ensures that the posterior will be different from the "true" flux, without any way to infer the significance of that difference. As in, how do the authors know that the MR's they estimate are not because in the one realization of the measurements they used, some of them just happened to be skewed in one direction? This is especially a concern for the NET17 network, since there are so few measurements, with scant opportunity to average over the perturbations.

**Response:**

We carefully analyzed and discussed the UR and MR results keeping in mind and explicitly recalling the problems of working with a single realization of $\mathbf{y}_o$ and $\mathbf{x}^b$. The risk of under- or over-estimating the posterior error in the emission budgets when using one realization only for the prior and measurement errors is the reason why we compute the "statistics of misfits" at the monthly scale and the "MR for monthly emissions" (see our answer to major question 3). By assuming that the monthly misfits all follow the same distribution, we roughly consider that we have an ensemble of twelve realizations of monthly inversions for which the average-based MR give a reliable indicator or the error reduction that should not be significantly skewed by sampling errors.

Of note is also that even when considering NET17 and a 2-week sampling strategy, we have 416 data over the year and thus 416 realizations of perturbations to individual observations. Finally, the scores of UR should definitely be considered as the reference indicators of the inversion behavior, and the MR are mainly analyzed to provide confidence in these UR, as explained in Sect. 2.2.1.

The perturbation of the observation according to **R** even when a single realization of the observation and prior errors is considered is a common practice in inversion because, we believe, this is what still makes most sense if having to assess the

uncertainty reduction using a single inversion. In other words, we do not have a simple idea of perturbation that would make the single realization more adapted for assessing the typical impact of the errors.

We will revise the paragraphs in Sect. 2.2.1 to clarify the use of one realization of observations.

References:

Basu, S., Miller, J. B. and Lehman, S.: Separation of biospheric and fossil fuel fluxes of $CO_2$ by atmospheric inversion of CO2 and $^{14}CO_2$ measurements: Observation System Simulations, Atmos. Chem. Phys., 16(9), 5665–5683, doi:10.5194/acp-16-5665-2016, 2016.

10. Line 374 and Figure 4: The authors solve for monthly emissions over a year, but report a single UR/MR map of monthly emissions. Is this the RMS of UR/MR values over 12 months, or the UR/MR calculated from the RMS of the posterior errors, or...? As in, can the authors give a mathematical expression of what is being shown in Figure 4 as the "monthly" UR/MR, in terms of their control vector and/or covariance matrix?

**Response:**

This is related to the major comment #3. We computed the quadratic mean of the twelve values of monthly uncertainties and misfits to derive the so-called "average monthly URs and MRs". As stated in the response to the major comment #3, we will add two equations to show that we compute the UR and MR "at monthly scale" based on the quadratic mean of twelve monthly values across one year.

11. Line 381: I think the reference to Figure 4(d) should actually be to 4(e). Likewise, in line 384, the refences should be to 4(e) and 4(g).

**Response:**

The reviewer is right. In line 381, the reference to figure is Fig. 4e and on line 384, the reference to figure is Fig. 4e and 4g. We are sorry about these mistakes. We went through the paper again carefully and feel that there are not such mistakes anymore.

12. Line 409: "... the posterior misfits are even larger than the prior misfits." Why does the inversion allow this? For stations within the blue regions, is this obvious from looking at the atmospheric FF CO2 time series, that post-optimization the time series is further away from the pseudo-data than pre-optimization? I suspect the perturbed measurements are to blame (see earlier comment).

**Response:**

The primary reason for getting posterior misfits in the emissions that are larger than the prior misfits in the emissions is connected to the fact that the prior uncertainty matrix **B** of the inversion system does not match the statistics of the actual errors in the prior estimate of emissions:

1) As demonstrated by Wang et al. (2017), the signature of the errors in the prior estimates in the $FFCO_2$ emissions has a smaller amplitude than the observation errors and the ability to filter this information for a proper correction of the emissions

strongly relies on the knowledge of the prior uncertainty covariance. If **B** misses the amplitude, temporal and spatial correlations of the actual errors, the system can easily translate observation errors into its corrections to the emissions. While decreasing the differences between the simulated $FFCO_2$ concentration time series and the atmospheric data, it would increase the misfits to the true emissions.

2) Some of the region-months are poorly constrained by the observations due to the network distribution and to the meteorological conditions (even when using NET233), and the corrections to emissions from a poorly constrained region-months is imposed by the extrapolation of the corrections to other region-months following the patterns of **B**. If those patterns are wrong (typically, if spatial correlations in **B** for the errors of two neighbor regions are negative while the spatial correlations between the actual errors of these regions are positive), the system could apply corrections with a wrong sign or amplitude to the poorly observed region-months, which can easily lead to an average increase of the errors for such regions. The problem is similar when the network can constrain the sum of the budgets for several region-month but not these region month individually (due to being too coarse). In this case, if the structure of **B** is wrong, the repartition of the constraints from the observations between these different region-months could be erroneous.

The fact that the aggregation error has a non-Gaussian and biased distribution while the inversion system believe it is Gaussian and unbiased can also feed the generation of negative MR as well as problems of sampling errors when computing scores of MR as suggested by the reviewer. However we believe that the impact of these additional factors is relatively small because when the setup of **B** matches well (even though not perfectly) with that of the actual prior errors, we have very few negative MRs.

We will rewrite Sect. 4.3, lines 541-555 to present the inconsistencies between UR and MR and discuss the possible causes of the negative scores of MR.

References:
Wang, Y., Broquet, G., Ciais, P., Chevallier, F., Vogel, F., Kadygrov, N., Wu, L., Yin, Y., Wang, R. and Tao, S.: Estimation of observation errors for large-scale atmospheric inversion of $CO_2$ emissions from fossil fuel combustion, Tellus B: Chemical and Physical Meteorology, 69(1), 1325723, doi:10.1080/16000889.2017.1325723, 2017.

13. Line 417: The correlation is not between uncertainties, but between corrections from the prior emission.

**Response:**

We do not agree with the reviewer. Fig. 6 shows the correlations in the **B** and **A** covariance matrices. Since **B** and **A** characterize the uncertainties in the prior and posterior estimates, the correlations in both **B** and **A** are indeed between uncertainties rather than corrections.

Following Eq. (2) in the main text, the correction for **x** is actually $\mathbf{AH^TR^{-1}(y_o–Hx^b)}$, so that the covariance matrix for the corrections is $\mathbf{AH^TR^{-1}(HBH^T+R)R^{-1}HA}$, and the correlations correspond to this matrix are between corrections.

**14. Line 455: I'm surprised at the low UR for the NET233 network. Why are there so many white areas (low UR) still?**

**Response:**

    This is related to the fact that the signals from the uncertainties in the prior estimate of emissions are well below the observation errors in these regions and that the observation errors have similar error structures (temporal and spatial correlations) to the signals of uncertainties in the prior estimate of emissions, as discussed in Sect. 4.2 (line 516-520). This demonstrates that the proper account for representation and aggregation errors avoid overestimating the potential of the atmospheric inversion in OSSES when using a coarse resolution transport model. We will emphasize this by updating the discussions.

**15. Line 484: In real trend detection situations, the transport will vary year by year, as will the disequilibrium and nuclear fluxes of $^{14}$C. Can the authors estimate how big an impact this will have on the trend detection?**

**Response:**

    There is not a strong reason to think that there could be some major change in the uncertainties in the estimate of nuclear and disequilibrium fluxes of $^{14}$C from year to year, which is what really matters on the inversion results (rather than changes in the absolute value for these fluxes), as discussed in the response to the first major comments. This response also indicates that we assume that such uncertainties should have a relatively weak impact on the inversion results.

    The transport conditions (and thus **H** and the uncertainties in the transport modeling in **R**) may vary significantly from year to year which could lead to variations of the sensitivity of the observations to the emissions and variations of the scores of uncertainty reduction. We could even observe trends in the transport conditions over several years (e.g. the shallowing of the PBL between the 1990s and the 2000s by Aulagnier et al., 2009; Ramonet et al., 2010). However, the impact of the resulting changes in atmospheric transport condition on the UR and posterior uncertainties, and thus on our computation of the trend detectability is difficult to anticipate. We assume that this should not result in a behavior that is different from the one described by our computation with constant posterior uncertainties from year to year (it should change the result in a quantitative but not qualitative way). Conducting tests with varying posterior uncertainties would thus fall out of the scope of our assessment of the typical uncertainties in trends for a typical uncertainty in annual emissions, and we thus think that it is not worth investigating it.

    We will add some discussions to admit that the changing transport may have an impact on the trend detection but quantifying this impact would require detailed studies and further investigation.

References:

Aulagnier, C., Rayner, P., Ciais, P., Vautard, R., Rivier, L. and Ramonet, M.: Is the recent build-up of atmospheric $CO_2$ over Europe reproduced by models. Part 2: an overview with the atmospheric mesoscale transport model CHIMERE, Tellus B, 62, 1, 2009.

Ramonet, M., Ciais, P., Aalto, T., Aulagnier, C., Chevallier, F., Cipriano, D., Conway, T. J., Haszpra, L., Kazan, V., Meinhardt, F., Paris, J.-D., Schmidt, M., Simmonds, P., Xueref-Rémy, I. and Necki, J. N.: A recent build-up of atmospheric $CO_2$ over Europe. Part 1: observed signals and possible explanations, Tellus B, 62(1), 1–13, doi:10.1111/j.1600-0889.2009.00442.x, 2010.

16. Line 499: As far as I can tell, Basu et al. (2016) did not estimate UR.

**Response:**

Basu et al. (2016) did not estimate UR, but they analyzed the misfits between the inversion-estimated fluxes and true fluxes such as when we analyze scores of MR. The text assumed that the "error reduction" can be characterized by either UR or MR. We will revise the sentence here to avoid the confusion.

17. Line 531: The authors seem to suggest that the boundary condition – a bane of most regional inversions – does not affect their flux and uncertainty estimates. Is this because everything is referenced to JFJ?

**Response:**

Our simulation indicated that the reason why, in this study, the uncertainties in the emissions remote from Europe (i.e. what would feed the boundary conditions in a regional model) does not significantly impact the $FFCO_2$ inversion in Europe is the atmospheric diffusion of the $FFCO_2$ signal associated with these uncertainties. The high emitting regions outside Europe (USA, China) are far from the European continent. The diffused signals from these regions does not yield large gradients between the European sites which are mainly due to the emissions from European continent. Therefore, these signals should not impact the inversion of fluxes within Europe. Assimilating gradients to JFJ definitely helps the system understand that a large scale signal over Europe should not be connected to European emissions, but inversions system assimilating $FFCO_2$ data at individual site could also naturally avoid to correct for emissions between the measurements sites based on such a large scale signal.

The boundary conditions for traditional regional inversion of natural $CO_2$ fluxes assimilating total $CO_2$ concentrations which only target at natural $CO_2$ fluxes can be more critical, since there are large fluxes from Atlantic Ocean and other adjacent continents like Asian Russia which can cause significant spatial patterns between European sites. But still, some studies, such as that of Lauvaux et al. (2008) showed that the influence of the boundary conditions is not significant in their regional inversion of the natural $CO_2$ fluxes in the South West of France.

At last, we keep in mind the fact that if the inversion would account for uncertainties in the biogenic and ocean $^{14}CO_2$ fluxes, the situation could be different and could pose problems for the boundary conditions of the regional scale systems. There could be large $^{14}CO_2$ fluxes from Atlantic Ocean and other adjacent continents like Asian Russia which can cause significant spatial patterns of $^{14}CO_2$ within Europe.

This will be better discussed in Sect. 4.2.

References:

Lauvaux, T., Uliasz, M., Sarrat, C., Chevallier, F., Bousquet, P., Lac, C., Davis, K. J., Ciais, P., Denning, A. S. and Rayner, P. J.: Mesoscale inversion: first results from the CERES campaign with

synthetic data, Atmos. Chem. Phys., 8, 3459–3471, 2008.

18. Eqs. C-3 and C-4: I believe there are errors in these two formulae. If $\vec{p}$ is obtained by minimizing the cost function J

$$J = \frac{1}{2}\left(\vec{y} - \tilde{\vec{y}}\right)^T R^{-1}\left(\vec{y} - \tilde{\vec{y}}\right) \tag{1}$$

where $R = cov(\vec{y})$ (R in their case contains the posterior error estimates on fluxes), then the optimal estimate of $\vec{p}$ and the corresponding covariance are

$$\vec{p} = (X^T R^{-1} X)^{-1} X^T R^{-1} \vec{y}$$

$$cov(\vec{p}) = (X^T R^{-1} X)^{-1} X^T R^{-1} X (X^T R^{-1} X)^{-1}$$

**Response:**

The reviewer is right if cov(**Y**) (in the review's equations, it's **R**) is not diagonal or if the diagonal terms (variances of the variables in **y** vector) are not equal. In fact, the last equation proposed by the reviewer can be further simplified to:

$$cov(\vec{p}) = (X^T R^{-1} X)^{-1}$$

The equation we wrote as Eqs. C-3 and C-4 only applies when cov(**Y**) is diagonal and the terms in **y** all have equal variance, that is cov(**Y**)=$\sigma^2$**I** where $\sigma^2$ is the variance for all variables and **I** is an identity matrix. The equivalence between our equations and those proposed by the reviewer can be proved:

$$cov(\boldsymbol{p}) = (\mathbf{X^TX})^{-1}\mathbf{X^T}cov(\mathbf{Y})\mathbf{X}(\mathbf{X^TX})^{-1} = (\mathbf{X^TX})^{-1}\mathbf{X^T}\sigma^2\mathbf{IX}(\mathbf{X^TX})^{-1}$$
$$= \sigma^2(\mathbf{X^TX})^{-1}\mathbf{X^TX}(\mathbf{X^TX})^{-1} = \sigma^2(\mathbf{X^TX})^{-1} = (\mathbf{X^T}\sigma^{-2}\mathbf{IX})^{-1} = [\mathbf{X^T}cov(\mathbf{Y})^{-1}\mathbf{X}]^{-1}$$

In this sense, the equations proposed by the reviewer is more generalized. In this paper, we indeed used the cov(**Y**) in the equal variance case, so that the computation and results of the paper are correct.

In order to avoid any confusion, we will add the equations proposed by the reviewer.

19. Table 2: In columns 2 and 3, I believe rows 3 and 4 have been flipped.
**Response:**
The reviewer is right. We correct this mistake accordingly.

20. Figure 2(a): Am I supposed to see 56 colors in the world map? I don't. I think the problem is that the country and state boundaries overlap with the region boundaries. I would suggest, at least in the world map, only showing the region boundaries from the control vector and eliminating the country and state boundaries.
**Response:**
Here, we do not use 56 colors in the map actually because it is hard to find 56 colors that are visibly differentiable. In fact, we repeatedly use 12 colors for

non-adjacent regions. For example, the Northern Europe, Middle East, one region in the USA and one region in China are all red. But because they are in different continents, so that they represent four different regions. We will change Figure 2 by removing the lines country and state boundaries, as the reviewer suggested, and will also mention the fact we repeatedly use 12 colors for non-adjacent regions in the caption.

---

## Author Response (AR1)

**Response to comments on "Potential of European $^{14}CO_2$ observation network to estimate the fossil fuel $CO_2$ emissions via atmospheric inversions" by Y. Wang et al.**

We thank the three referees for their very detailed reviews. Their comments have allowed us to improve the manuscript by better emphasizing its strength. In this final response, we keep all the answers to the reviewers in previously response, but also add changes in the revised manuscript, highlighted with dark red. All the pages and line numbers correspond to the original versions of text.

General Comments
This paper lays out the potential for current and future $^{14}CO_2$ observations to improve estimates of fossil fuel emissions in Europe. It uses two types of Observing System Simulation Experiments (OSSEs) based on either the theoretical uncertainty reduction for a well-tuned case or a more realistic case where prior uncertainties do not match differences between prior and truth. It also uses several versions of an observing network ranging from the current network to a saturated case where every grid cell in the target domain is sampled. Results are not very surprising with the current network offering useful information at the conjunction of dense networks and high emissions (and concomitant uncertainties) with the case improving as networks become more dense. Results are, however, sensitive to the proper tuning of prior covariances; a salutary result the authors are right to emphasise. The paper addresses an important problem with reasonable if not state-of-the-art tools, is clearly written and within scope.

**Response:**

We would like to thank the reviewer for the valuable comments and suggestions for improving our manuscript. Following the reviewer's comments, we will carefully revise our manuscript. Most of the concerns about the observation and aggregation errors raised by the reviewer were analyzed (at least partly) in Wang et al. (2017) which is cited in our manuscript. We will better remind the conclusions from this paper in the manuscript.

Please find below the point-to-point responses (in black) to all referee comments (in blue). All the pages and line numbers correspond to the original versions of text.

References:
Wang, Y., Broquet, G., Ciais, P., Chevallier, F., Vogel, F., Kadygrov, N., Wu, L., Yin, Y., Wang, R. and Tao, S.: Estimation of observation errors for large-scale atmospheric inversion of $CO_2$ emissions from fossil fuel combustion, Tellus B: Chemical and Physical Meteorology, 69(1), 1325723, doi:10.1080/16000889.2017.1325723, 2017.

I have two concerns about the paper, one general and one specific. the authors note the dependence of their results on the resolution of their transport model (3.75×2.5°)

but I think should do more to evaluate this. It is unlikely that anyone would use this resolution for an inversion of fossil fuel emissions targeting Europe and the guidance on network density is hard to generalise.

**Response:**

In this paper, our analysis focuses on the inversion of European fossil fuel emissions. However, we have worked with a global and thus coarse resolution transport model in order to: (1) properly account for the uncertainties in emissions from other continents than Europe when inverting European emissions, and (2) because we developed a system which also allows us to study the inversion of the emissions in North America and Eastern Asia.

Sect 4.2 analyses whether the uncertainty in the emissions outside Europe has an impact on the inversion of the emissions in Europe. The results indicate that this impact is in fact weak, which was not obvious to prove before doing the study. Furthermore, studies including some of the sources of uncertainties that have been ignored here could reveal, e.g. that uncertainties in the $^{14}CO2$ fluxes from oceans and land ecosystems outside Europe have a strong impact on the inversion of the emissions in Europe. A cautious account for such uncertainties could require the use of a global inverse modeling system, or of the coupling between a European scale and global scale inverse modeling systems. At our stage of investigation in this study, we thus think that the use of a global inversion system is appropriate.

The spatial resolution of LMDZ is typical for global transport models and inversion studies (Peylin et al., 2013). For example, the Transport Model 3 (TM3, 5°×4°) used for the Jena CarbonScope (Rödenbeck et al., 2006), TM5 (3°×2° without nested version) used for CarbonTracker (Peters et al., 2007), Model of Atmospheric Transport and Chemistry (MATCH, 5.6°×2.8°) and the CSIRO Conformal-Cubic Atmospheric Model (CCAM, about 220 km) used by Rayner et al. (2008), have similar spatial resolutions as LMDZv4 used here. Using a much higher resolution transport model, e.g. 1°×1°, for global simulations is computationally expensive.

In principle, we properly accounted for the representation error and its temporal and spatial correlations by using the detailed analysis of the aggregation and representation errors from Wang et al. (2017). In particular this should prevent from overestimating the effect of the spatial sampling of $FFCO_2$ and thus performance of inversions when using dense networks. In a more general way, we think that our configuration of the observation errors support our confidence in the guidance that we derived from our relatively coarse resolution inversion system regarding the impact of the network density. In our conclusions, we were cautious regarding the dependence of the results to the transport spatial resolution.

The analysis by Wang et al. (2017) provides some insights and understanding on the dependence of the results to the resolution of the transport model. However, running atmospheric inversions using higher spatial resolution model, which was out of the scope of this study, would have been the only way to assess the dependence of the results to the spatial resolution correctly, since it depends on a complex combination between the prior and observation error covariance structures together with the atmospheric transport.

We highlight that the use of LMDZv4 aims at properly accounting for the uncertainties in FFCO$_2$ emitted over other regions outside Europe, by adding in line 112: "Although the results are presented only over Europe, we use a global inversion system and the global transport model LMDZv4 to ensure that uncertainties in FFCO$_2$ emitted over other regions of the globe are properly accounted for and to study their impact on the inversion of the FFCO$_2$ emission in Europe. LMDZv4 has a 3.75 $^\circ\times$2.5 $^\circ$ longitude$\times$latitude horizontal resolution and 19 layers in the vertical between the surface and the top of the atmosphere. This spatial resolution is comparable to that of transport models used in state-of-the-art global inversions (Peylin et al., 2013). We assess the potential…"

We also better stress the dependence of our results to the spatial resolution of the transport model but the fact this study aimed at providing some understanding of the inversion behavior and sensitivity to the network density rather than to provide a precise quantification of the uncertainty reduction that would be obtained if working with real data by:

- Adding in line 110: "… at ICOS-like stations. The study primarily aims at providing a typical quantification of the inversion performances and at understanding qualitatively how the inversion behaves depending on the level of FFCO$_2$ emissions, on the knowledge on these emissions and on the network density. …"
- Adding in line 521: "This study provides understanding of the inversion behavior and sensitivity to network density, but the precise quantification of the performance of the inversion is largely dependent on the spatial resolution of the transport model. Wang et al. (2017) showed that the representation error contributes the most to the observation errors, followed by the transport and measurement errors. …"

The authors can help a little here since their group has access to higher resolution models. How much do the representation and aggregation errors change with increasing model resolution. Representation error probably decreases while aggregation error increases but how much? Increased resolution makes gaps in the network inevitable, what effect will they have? this could be tested by a couple of systematic thinning experiments on the saturated network case here.

**Response:**

Wang et al. (2017) used the meso-scale transport model CHIMERE run with a 0.5 $^\circ$ horizontal resolution to assess the statistics of the representation and aggregation errors when working with the global inversion system that is used in our study. These statistics are summarized in Sect. 2.2.2 (Page 11, lines 327-337) and Table S3 and Table S4 of this paper.

The representation error will definitely decrease with increasing spatial resolution for the transport model. Our definition (which is also that of Wang et al., 2017) of the representation error encompasses the errors associated to the representation of the emissions using a constant value within one pixel and one time step of the transport model. Therefore, our definition of the aggregation errors limits them to the errors

associated with the fixed spatial distribution of the emissions within a region and month at the transport model spatial and temporal resolution. With such definitions, the aggregations errors increase when the spatial resolution of the transport model becomes finer. But such an increase is balanced, in the representation error, by the decrease of the component associated to the emission representation. Overall, the dominant pattern of the variations of the observation errors associated with the increase of the transport model spatial resolution should be the decrease of the representation error associated with the representation of the concentrations.

If following the specific framework and error definitions of Wang et al. 2017, a precise assessment of the change of the representation error in Europe as a function of the spatial resolution of the transport model would require series of European scale simulations with emission maps at different spatial resolutions (e.g. $1^o$, $1.5^o$, etc.) to feed the high-resolution transport model, and then require aggregating the output (concentrations) of the transport model at corresponding spatial resolutions (e.g. $1^o$, $1.5^o$, etc.). It would have been feasible but it was out of the focus of this previous paper. It would now be out of the scope of our paper to resume such computations, especially since properly assessing the impact of these changes of representation errors in the inversion results, would require conducting inversions with different transport model configurations (topographies, wind fields etc. are needed at different resolutions) as stated in the previous answer to the reviewer's comment.

Regarding the "gaps in the network", this phenomena is supposed to be well accounted for even when using the coarse resolution global transport model and when having two stations in each grid cell of this model thanks to a proper account for the observation errors. Representation errors indicate to the system that a station does not have a full coverage of its corresponding grid cell and so that it does not see the same information as the other station in the same grid cell (i.e. that there is already a gap between them even if using the coarse-resolution transport model of LMDZ). The physical separation of the stations in the grid of a higher spatial resolution system should not lead, in principle, to a strongly different behavior of the inversions, especially since the correlation length scale of the projection of the prior uncertainties in the concentration space is 700 km. In order to demonstrate it, we have conducted three additional experiments with different thinned networks: a) one with two sites located in the same grid cell of every two grid cells (113 sites in total); b) one site in each grid cell (117 sites in total); c) one site every two grid cell (57 sites in total). Fig. 1 shows the URs for INV-E inversions (the behavior of the results from INV-N inversions are similar but not shown). Fig. 1a and 1b show quite similar distributions and values of UR scores. The comparison between the Fig. 4g in the original manuscript with Fig. 1a and 1b here, and between Fig. 1a and 1b with Fig. 1c, show the decrease of UR across Europe due to using less sites. Since NET233 and the three thinned networks are uniformly distributed across Europe, this decrease of URs due to the gap in the networks are also nearly uniform, which confirms that the general behavior of the inversion does not significantly change due to generating gaps between the observed grid cells. We do not plan to include these results in the manuscript because they are not qualitatively very different from the ones we already

showed and would not really lead to new insights or conclusions on the inversion behavior.

[Figure]

**Figure 1:** Average monthly uncertainty reductions in FFCO$_2$ emissions from INV-E inversions over regions delineated by solid black lines, using three networks and 2-week sampling for the inversions. The three networks are: a) two sites located in the same grid cell of every two grid cells (113 sites in total); b) one site in each grid cell (117 sites in total); c) one site every two grid cell (57 sites in total). The dots and triangles denote the locations of the observation sites where the gradients are extracted with respect to the JFJ reference site. Dots (triangles) correspond to "urban" (or "rural") stations defined in Sect. 2.1 of the original manuscript.

We add in line 521: "followed by the transport and measurement errors. Following the definition of the observation errors in Wang et al. (2017) and in this study, the representation and the transport error are highly dependent…" We highlight the fact that assessing properly the impact of the change these errors on the inversion results would require a large amount of work (which would be worth being investigated) by adding in line 527: "… to improve the results from atmospheric inversion of FFCO₂ emissions at regional scale. A proper quantification of the change of representation and transport error as a function of spatial resolution, and of the impact of this change on the performance of the inversion system would require a series of transport models and inversions at varying spatial resolution which are out of the scope of this study but which would be worth being investigated in the future.".

We also add cautious discussions on this general topic raised by the reviewer in the discussion section by:

- Rewriting the paragraph from lines 93-107: "In this study, we study the potential of an atmospheric inversion system to quantify FFCO₂ emissions at regional scales (i.e. the size of a medium-sized country in Europe like France or Germany) over the European continent based on continental-scale networks of atmospheric CO$_2$ and $^{14}$CO$_2$ measurements. Special attention is paid to the representation and aggregation errors induced by the use of a coarse grid transport model. Wang et al. (2017) derived the statistics of these errors for the inversion system that we apply here, which is based on the Laboratoire de Météorologie Dynamique LMDZv4 global transport model (Hourdin et al., 2006) and our study strongly relies on their results. They highlighted that both the representation and aggregation errors have large magnitudes, and could thus strongly reduce the ability of the inversion to

filter the information on the uncertainties in regional FFCO$_2$ emissions. They also stressed the fact that the spatial scales of the correlations in the representation and aggregation errors are smaller than that of the projection in the atmospheric observation space of the typical uncertainties in the prior estimates of regional emissions (called "prior FFCO$_2$ errors" hereafter). More precisely, with their modelling configuration they obtained values smaller than 200 km and larger than 700 km respectively for these spatial scales. Therefore, if the observation networks are dense enough to provide information at finer spatial scale (typically with distances from a given station to the closest ones being systematically smaller than 700 km), the impact of aggregation and representation errors on the inversion of the regional budgets of FFCO$_2$ emissions could be small (Wang et al. 2017). In this study, we account for the aggregation and representation errors using their detailed and quantitative characterization and check whether using dense networks could overcome the limitations brought by coarse resolution transport models and by the uncertainties in the distribution of the emissions at high resolution when retrieving regional emission budgets. Using the error estimates from Wang et al. (2017) ensures that our inverse modelling system does not overestimate the potential of measurement networks that are dense compared to our coarse transport model resolution but whose distances between the sites are larger than the spatial scales of local atmospheric signals from the anthropogenic emissions."

- Adding in line 151: "… assumed to be one urban and one rural distant by more than 200 km in order to combine data for the structures of representation errors are different (i.e. which have a different view in terms of the scale of FFCO$_2$ emissions). Any of the transport model pixels provides such locations since having areas of nearly $10^5$ km$^2$ (Wang et al. 2017)."

- Adding in the Sect. 4.2, line 520: "… 2) the observation errors bear complex temporal and spatial correlations which are close to the prior FFCO$_2$ errors (Wang et al., 2017). Such a result illustrates the need for using a suitable observation error characterization (here based on the results from Wang et al., 2017) to prevent the stations having a full coverage of information on the emissions in the model framework shown here even when the observation network is as dense as NET233. A proper account for the observation errors and their temporal and spatial correlations avoid overestimating the potential of the atmospheric inversion in OSSES when using a coarse resolution transport model."

My other concern is for this saturated case. As I understand it, each grid cell is oversampled with two measurements. If this is the case and the transport Jacobians for the two measurements are the same then I think the two measurements can be combined into a single measurement by summing their information content. There should also be strong correlation between the two measurements in the same grid cell, accounting for large-scale errors in the transport model. In particular, I think that the

relationship between the aggregation and representation errors for the two types of site is complex, interesting and perhaps important. It is quite possible that using both types of site reduces the sampling inhomogeneity necessary for aggregation errors (Trampert and Snieder, 1996; Kaminski et al., 2001).

**Response:**

Yes, in this case, each grid cell is sampled by two measurements at each sampling time, and the transport Jacobians for the two measurements are the same. Our modeling of R (based on the statistical estimates by Wang et al., 2017) is made such that there are full correlations between the transport errors and between the aggregation errors in the two measurements within the same grid cell. However, Wang et al. (2017) showed that the spatial correlation of the representation errors is less than 100 km, while the typical distance between the stations in NET233 network (with two sites per 3.75 °×2.5 ° grid cell) is about 200 km. Therefore we ignored the spatial correlation between the representation errors in the two measurements within the same grid cell. Wang et al. (2017) also diagnosed that the spatial correlation between representation errors for urban and rural sites is even smaller than the spatial correlation between two rural or urban sites so that it is also negligible. In addition, their analysis does not reveal any correlation between representation and aggregation errors (if following their definition of these types of errors as discussed above). Our configuration of the observation error matrix in this study exactly followed these indications.

Mathematically speaking the two measurements in each grid cell could be combined into a single measurement, but this would require the derivation of a complex observation error covariance matrix for the "combined" measurements, accounting for all the components of the observation error for individual data (measurement, transport, representation and aggregation errors) with varying standard deviations (depending on the location of the stations for the computation of transport error and on the urban or rural type of the station for the representation error) and their respective temporal and spatial correlations. In this context, such a combination would not really simplify the representation and understanding of the inversion problem of the data and of their observation errors.

We rewrite the paragraph in lines 323-337 to better describe the configuration of the **R** and associated correlations in the observation errors:

"In this study, we use the estimates of the standard deviations and of the correlation functions for these different types of observation errors from Wang et al. (2017) to set up the **R** matrix. Wang et al. (2017) sampled representation and aggregation errors by using simulations with a mesoscale (with higher resolution than LMDZv4) regional transport model and by degrading the spatial and temporal resolution of the emission maps in the input of this model and in the output $FFCO_2$. Based on these samples, the standard deviation of $\varepsilon_r$ was characterized by a function of season and on whether a station is "urban" or "rural" (see Sect. 2.1). For $\varepsilon_a$, the standard deviation for spring/summer and autumn/winter were derived. The standard deviation of the transport error at a given site is set-up proportional to the temporal standard deviation of the 1-year long time-series of the high-frequency variability of

the detrended and deseasonalized simulated daily mean afternoon mixing ratios in the grid cell of the transport model, at which the sites are located. Such an estimation of transport error which relies on some results from Peylin et al. (2011) aims at representing the typical value for global transport models, not that of the specific transport model used in this study. The temporal auto-correlations in the representation and aggregation errors were characterized by Wang et al. (2017) using the sum of a long-term component and a short-term component: $r(\Delta t)=a\times e^{-\Delta t/b}+(1-a)\times e^{-\Delta t/c}$ where $\Delta t$ is the timelag (in days) and $a$, $b$, $c$ are parameters optimized by regressions against the samples of the errors. Furthermore, we do not include temporal auto-correlations in the transport error for simulated daily to 2-week mean afternoon $FFCO_2$ gradients, since previous studies of the auto-correlations of the transport errors have not evidenced that they should be significant at daily scale (Lin and Gerbig, 2005; Lauvaux, 2009; Broquet et al., 2011). This choice follows the corresponding discussion by Wang et al. (2017) and implicitly ignores that transport model errors likely bear long-term components (often referred to as "biases", Miller et al., 2015) even when being dominated by components on short timescales. The corresponding values of the standard deviation and the modelling of temporal autocorrelation of the observation errors for 2-week/daily mean afternoon $FFCO_2$ gradients are listed in Table S3 and Table S4.

A simpler account of the spatial correlations in the observation errors is derived from the diagnostics of Wang et al. (2017). We do not account for the spatial correlation in the representation error, as the scale of the spatial correlation according to Wang et al. (2017), i.e. 55-89 km, is much smaller than the size of the grid cells of the global transport model $\mathbf{H}_{transp}^{LMDZ}$ used for the inversion. When there are more than two sites are located in the same grid cell of the transport model, we consider that the aggregation errors and the transport errors are fully correlated between these sites, according to the definition by Wang et al. (2017). We do not account for spatial correlations between aggregation errors for measurements made at sites in different grid cells, because the scale of the spatial correlation is 171 km and is smaller than the size of the grid cell, according to Wang et al. (2017). Finally, we do not account for spatial correlations between transport errors or measurements made at sites in different grid cells."

We also stress in the updated manuscript the fact that using NET233 reduces the sampling inhomogeneity and can reduce the impact of aggregation errors, as shown by the references proposed by the reviewer by adding in line 141: "…of the LMDZv4 transport model (Fig. 1c). The NET233 network is denser than NET17 and NET43 in the high emitting regions, e.g. Germany, and also covers the region that is not well sampled by NET17 and NET43. However, the location of its 233 sites is not intended to be optimal since the emissions have a very heterogeneous spatial distribution. Their homogeneous spreads allow us to reduce the impact of representation and aggregation errors (Trampert and Snieder, 1996; Kaminski et al., 2001) and to assess the impact of having a dense network for all control regions."

**Response to comments on "Potential of European $^{14}CO_2$ observation network to estimate the fossil fuel $CO_2$ emissions via atmospheric inversions" by Y. Wang et al.**

We thank the three referees for their very detailed reviews. Their comments have allowed us to improve the manuscript by better emphasizing its strength. In this final response, we keep all the answers to the reviewers in previously response, but also add changes in the revised manuscript, highlighted with dark red. All the pages and line numbers correspond to the original versions of text.

The authors present an OSSE study of the capability of ICOS $^{14}CO_2$ observations to constrain European fossil fuel $CO_2$ fluxes and their trends. The study is well structured and should be published. I have a few comments which I'd like the authors to address before publication.

**Response:**

We would like to thank the reviewer for the valuable comments and suggestions for improving our manuscript. Following the reviewer's comments, we will carefully revise our manuscript. Please find below the point-to-point responses (in black) to all referee comments (in blue). All the pages and line numbers correspond to the original versions of text.

**Major comments**

1. Line 112: The assumption that $^{14}CO_2$ measurements can be accurately translated into FF $CO_2$, i.e., there are no spatial patterns introduced due to the other terms (especially disequilibrium and nuclear plants in the European context), is a big one. Those terms will not only affect annual emission estimates, but also the ability to detect trends, as countries change their nuclear power generation capacity and switch to wood-fired domestic heating (e.g., Germany). I understand that modeling the full $^{14}CO_2$ budget is beyond the scope of the authors' framework, but it should be possible to estimate the impact, by e.g. modelling just the nuclear or disequilibrium contribution as a tracer in a transport model and looking at the change in $\Delta^{14}C$. Have the authors done that? Unless that concern is addressed, the actual numbers from the manuscript are hard to trust.

**Response:**

Indeed, the modeling of the full $^{14}CO_2$ budget is beyond the scope of this study. It would be necessary to study the impact of uncertainties in other $^{14}CO_2$ fluxes on the $^{14}CO_2$ gradients, to precisely quantify the corresponding errors when converting the gradients of atmospheric $^{14}CO_2$ measurements into $FFCO_2$ gradients. Graven and Gruber (2011) estimated the sources of $^{14}C$ from nuclear power generation and spent fuel reprocessing and used the global TM3 transport model at 1.8 °×1.8 ° resolution, which is slightly higher than LMDZv4 used in our study, to simulate their

continental-scale influences on $\Delta^{14}C$. Their results showed that nuclear enrichment may cause an impact of -0.9 [-0.6, -1.4] ppm in $FFCO_2$ for Orleans, France (48.8 °N, 2.5 °E) and an impact of -0.7 [-0.4, -1.3] ppm for Heidelberg if nuclear $^{14}C$ enrichment was not accounted for. These two sites are representative of European continental sites that are close to nuclear power plants. Turnbull et al. (2009) estimated the impact in large-scale $FFCO_2$ signals caused by ignoring other $^{14}C$ fluxes including cosmogenic production, $^{14}C$ disequilibrium between atmosphere and biosphere and between atmosphere and ocean, and $^{14}C$ source from nuclear power plant (the estimate of $^{14}C$ sources from nuclear power plants is not as accurate as Graven and Gruber, 2011), using the same transport model LMDZv4 as in this study. Turnbull et al. (2009) showed that impact caused by other $^{14}CO_2$ sources in translating atmospheric measurements of $^{14}CO_2$ into $FFCO_2$ is mainly from terrestrial biosphere, whereas the contributions from ocean $CO_2$ exchange and cosmogenic production of $^{14}C$ contribute are weak. According to Turnbull et al. (2009), neglecting the influences from biosphere leads to an error typically between 0.2 and 0.8 ppm. Miller et al. (2012) estimated the impact of biospheric disequilibrium $^{14}C$ fluxes in North America and get a similar value as Turnbull et al. (2009), ranging from less than 0.2 ppm to 1.4 ppm.

We work with OSSEs so that what matters in our system is the correct representation of the uncertainties in the different components of the model, not a correct representation of these components. In our OSSE framework, by ignoring the fluxes of $^{14}CO_2$ other than the dilution of $^{14}C$ in $CO_2$ by fossil fuel emissions, we implicitly assume that the uncertainties in these fluxes has a weak impact, not that these fluxes themselves are ignored. The results mentioned above show that the uncertainties in the signals of these $^{14}CO_2$ fluxes may cause some impact on the interpretation of $FFCO_2$ using atmospheric $^{14}CO_2$ samples indeed, but also that this impact is below 1 ppm and thus smaller than the components of observation errors, e.g. measurement error (1 ppm), representation error (0.17-2.56 ppm) and transport error (0.52-4.15 ppm) in the paper. In this context, we assume that the influence of the uncertainties in $^{14}CO_2$ fluxes other than the dilution of $^{14}C$ in $CO_2$ by fossil fuel emissions on the inversion of fossil fuel emission should be relatively weak. Our estimate of URs and MRs could be slightly over-estimated due to the ignorance of other sources of uncertainties from other $^{14}CO_2$ fluxes. But this study aims at understanding how the inversion system behaves when dealing with uncertainties in the $FFCO_2$ emissions versus observation errors, about how the variation of the UR as a function of regions, of level of emissions, of the density of networks, rather than about providing absolute values of UR that would apply when conducting real-data applications. Further investigations accounting for uncertainties in other $^{14}CO_2$ fluxes will be needed to refine those numbers.

We explain the assumptions underlying the conversion of $^{14}CO_2$ into $FFCO_2$ gradients with a 1 ppm uncertainty given that we work with an OSSE framework (i.e. it does not mean that the nuclear plant, cosmogenic and biosphere fluxes themselves have been ignored) by rewriting the paragraph between lines 108-118: "Our inversion system solves for monthly $FFCO_2$ emissions in different regions of Europe over a

period of one year by assimilating synthetic observations of atmospheric gradients of $FFCO_2$ mixing ratios obtained from co-located $CO_2$ and $^{14}CO_2$ measurements at ICOS-like stations. The study primarily aims at providing a typical quantification of the inversion performances and at understanding qualitatively how the inversion behaves depending on the level of $FFCO_2$ emissions, on the knowledge on these emissions and on the network density. Furthermore, we assume here that the uncertainties in the signals from $^{14}CO_2$ fluxes other than the $FFCO_2$ emissions, such as that from terrestrial biosphere, oceans, nuclear power plants and cosmogenic production, should have a moderate impact on the order of magnitude of the inversion performances that are analysed in this study. This leads us to ignore these uncertainties and consider that the only uncertainties in the $FFCO_2$ mixing ratios data are related to the instrumental precision of $CO_2$ and $^{14}CO_2$ measurements. In practice, in the frame of this study, which focuses on the propagation of uncertainties, this is mathematically equivalent to assuming that $^{14}CO_2$ is a perfect tracer of $FFCO_2$. However, this does not imply that the signal from natural fluxes and nuclear power plants could be ignored when processing real data."

To recall the fact that the uncertainties associated with the signals from $^{14}CO_2$ fluxes other than the dilution of $^{14}C$ in $CO_2$ by fossil fuel emissions are expected to be relatively small compared to the uncertainties in the atmospheric $^{14}CO_2$ caused by fossil fuel emissions and by other types of observation errors and to stress that future studies will be needed to quantify the impact from uncertainties in the nuclear plant emissions, cosmogenic production, biogenic fluxes, etc., we modify the sentences in Sect. 4.2, between lines 502-507: "… This 1-ppm standard deviation approximately corresponds to the errors in the atmospheric measurements and ignores uncertainties in the conversion of $^{14}CO_2$ and $CO_2$ measurements into $FFCO_2$. Uncertainties in various fluxes that influence atmospheric $^{14}CO_2$, such as those from cosmogenic production, ocean, biosphere and nuclear facilities, bring errors to the conversion of $^{14}C$ measurements into $FFCO_2$ (Lehman et al., 2013; Vogel et al., 2013). Over land regions, heterotrophic respiration is expected to be one of the main contributors to the large-scale signals of atmospheric $^{14}CO_2$ (Turnbull et al., 2009). Over some areas of Europe, $^{14}C$ emissions from nuclear facilities may have even larger influences than plant and heterotrophic respiration (Graven and Gruber, 2011). The level of uncertainties in these fluxes and how much their influences on the $FFCO_2$ gradients will introduce additional errors remains to be quantified. According to the simulations by Graven and Grubber (2011), Turnbull et al. (2009) and Miller et al. (2012), one can expect that the impact of signals from the uncertainties associated in the estimate of these fluxes, on the conversion of atmospheric $^{14}CO_2$ measurements to $FFCO_2$, are typically below 1 ppm, i.e. much smaller than the observation errors that have been accounted for in this study, justifying that we have ignored these fluxes. However these signals may have complex spatial and temporal patterns leading to significant impact on the quantification of the inversion performances. Uncertainties in the trends of these fluxes could also impact that in the fossil fuel trend detection. Therefore, in future studies, especially if working with real data, the impacts from uncertainties in the $^{14}CO_2$ fluxes other than the anthropogenic fossil fuel emissions need to be

investigated and accounted for by modelling all these $^{14}CO_2$ fluxes, their atmospheric $^{14}CO_2$ signals and associated uncertainties."

References:

Graven, H. D. and Gruber, N.: Continental-scale enrichment of atmospheric $^{14}CO_2$ from the nuclear power industry: potential impact on the estimation of fossil fuel-derived $CO_2$, Atmos. Chem. Phys., 11(23), 12339–12349, doi:10.5194/acp-11-12339-2011, 2011.

Turnbull, J., Rayner, P., Miller, J., Naegler, T., Ciais, P. and Cozic, A.: On the use of $^{14}CO_2$ as a tracer for fossil fuel $CO_2$: Quantifying uncertainties using an atmospheric transport model, J. Geophys. Res., 114(D22), doi:10.1029/2009jd012308, 2009.

Miller, J. B., Lehman, S. J., Montzka, S. A., Sweeney, C., Miller, B. R., Karion, A., Wolak, C., Dlugokencky, E. J., Southon, J., Turnbull, J. C. and Tans, P. P.: Linking emissions of fossil fuel $CO_2$ and other anthropogenic trace gases using atmospheric $^{14}CO_2$, J. Geophys. Res., 117(D8), doi:10.1029/2011jd017048, 2012.

2. Line 145: The authors say that the inversion interpret the gradient between JFJ and other sites. I do not understand how that is implemented. Is it that JFJ is the only background site in the network, and hence the inversion implicitly interprets gradients w.r.t. JFJ (much as a global CO2 inversion might interpret everything w.r.t. MLO and SPO)? Or is it that the pseudo-obs are fed in after explicitly subtracting the JFJ time series, in which case the model's observation operator looks like "site – JFJ" at each individual site? Basically, the authors say in words that they interpret the gradient w.r.t. JFJ, but I do not understand how that is implemented in practice.

**Response:**

Our implementation corresponds to the second option explained by the reviewer, i.e. the pseudo-observation are differences between the data at other sites and at JFJ for a given time, and the observation operator relates the fluxes to "site minus JFJ" gradients. To clarify this, we revise the sentence in lines 159-160: "This correction is based on (i) a set of gradients of FFCO$_2$ mixing ratios between the different measurement sites and JFJ sampled during the afternoon (see Sect. 2.2.2) across Europe, called hereafter the "observations" $\mathbf{y}_o$, (ii)...", in lines 237-238: "In this study, we first consider 2-week integrated afternoon data. More precisely, we first consider 2-week averages of afternoon FFCO$_2$ gradients with respect to JFJ. In addition, we present tests with daily afternoon gradients, for which the corresponding sampling scheme would be more costly.", and in line 249: "... $\mathbf{H}_{transp}$ is the atmospheric transport model, and $\mathbf{H}_{samp}$ samples the FFCO$_2$ gradients with respect to JFJ corresponding to the observation vector from the transport model outputs (Wang et al. 2017)."

3. Line 201: I'm having trouble deciphering the meaning of "mismatch reduction", and its bounds and limits. Instead of describing it in words after equation (4), could the authors please write down the mathematical expressions for $\varepsilon_a$ and $\varepsilon_b$? Since I did not know what those $\varepsilon$'s were, I also could not interpret maps of MR (e.g., Figure 4). In particular, I did not understand what negative vs positive MR meant.

**Response:**

The "misfits" (i.e. the "mismatch" in the reviewer's comment) that are considered for the "misfit reduction" are the differences between the prior or posterior and "true"

estimates of the emission budgets. For a given region $i$ and month $m$ in the control vector $\mathbf{x}$, the prior misfit $\mathbf{x}^b_{i,m}-\mathbf{x}^t_{i,m}$ is denoted $\boldsymbol{\varepsilon}^b_{i,m}$ and the posterior misfit $x^a_{i,m}-x^t_{i,m}$ is denoted $\varepsilon^a_{i,m}$. We compute a misfit reduction for each region-month emission budget as the relative difference between the prior and posterior misfits:

$$MR_{i,m}=1-\varepsilon^a_{i,m}/\varepsilon^b_{i,m}$$

We only have one practical realization for $\mathbf{x}^b$, $\mathbf{y}_o$ and $\mathbf{x}^a$ and thus a single realization of the misfits for each month and region. However, we want to have a statistical assessment of the performance of the inversion system based on the misfits that could be compared to the scores of uncertainty reduction. Therefore, we also consider the typical MR at the 1-month scale for a given region $i$ which is the relative difference between the quadratic mean of the monthly prior misfits and the quadratic mean of the monthly posterior misfits.

$$MR_i = 1 - \frac{\sqrt{\frac{1}{12}\sum_{m=1}^{12}\left(\varepsilon^a_{i,m}\right)^2}}{\sqrt{\frac{1}{12}\sum_{m=1}^{12}\left(\varepsilon^b_{i,m}\right)^2}}$$

In all cases MR values could theoretically range between –infinity and 1. When, on average, the posterior emission estimates are closer to the synthetic truth than the prior estimates, the MR is positive (the inversion reduces the misfits). Conversely, when the posterior emission estimates are further from the synthetic truth than the prior estimates, the MR is negative (the inversion increases the misfits). MR is null when the posterior misfits are as large as the prior misfits, i.e. the inversion do not decrease or increase the misfits.

We revise the paragraphs in lines 202-210: "true values for the corresponding emission budgets. MR range from negative values (when the inversion deteriorates the precision of the estimation) to 1 (or "100%"; when the inversion provides a perfect estimate of the emissions).

We focus on uncertainties and misfits at both monthly and annual scales. However, we can have only one practical realization for $\mathbf{x}^b$, $\mathbf{y}_o$ and $\mathbf{x}^a$ following the protocol of that is presented in Sect. 2.3. Therefore, the assessment of the performance of the inversion for a given region-month using the corresponding score of MR may be over- or under-estimated due to the lack of sampling of the prior and observation errors. Consequently, at monthly scale, in order to strengthen the evaluation of the theoretical uncertainties based on these single realizations of the prior and posterior misfits, we compare, for a given region, the quadratic mean of the twelve monthly misfits (called "monthly misfits" without mention of a specific month in Sect. 3) to the quadratic mean of the standard deviations of the twelve monthly uncertainties (called "monthly uncertainties" without mention of a specific month in Sect. 3), which characterizes the average monthly uncertainties over the year. This computation implicitly assumes that the twelve monthly misfits through a year follow the same statistical distribution, and represent twelve independent realization of this distribution. In such a situation, the comparison between the averages of the prior and posterior monthly misfits give a good indications of the error reduction that should not be highly skewed by sampling errors. In the result section, for a given region $i$,

UR and MR scores derived at the "monthly" scale without mention to a specific month will correspond to the relative difference between the prior and posterior values of these average monthly uncertainties and misfits from a whole year of inversion:

$$UR_i = 1 - \frac{\sqrt{\frac{1}{12}\sum_{m=1}^{12}(\sigma_{i,m}^a)^2}}{\sqrt{\frac{1}{12}\sum_{m=1}^{12}(\sigma_{i,m}^b)^2}} \tag{5}$$

$$MR_i = 1 - \frac{\sqrt{\frac{1}{12}\sum_{m=1}^{12}(\varepsilon_{i,m}^a)^2}}{\sqrt{\frac{1}{12}\sum_{m=1}^{12}(\varepsilon_{i,m}^b)^2}} \tag{6}$$

At the annual scale, …"

**Minor comments**

1. Line 151: For NET233, each grid box is supposed to have one urban and one rural site. I'm not sure that's a good strategy. Wouldn't it be better to designate urban/rural depending on the nearest NET43 site? I mean, there could easily be grid boxes where it was more realistic to put two rural or two urban sites.

**Response:**

The design of NET233 was not intended to be optimal. We wanted to conduct a test with many sites whose distribution would be homogeneous and cover all the control regions. In this case, the variations of UR from one region to the other one are a direct consequence of variations in the emission uncertainties, and not as a consequence of the variations in the network density. Putting most of the sites near the areas with the highest emissions could have larger URs than spreading them homogeneously across Europe, but will not distinguish the role of network density and the role of emissions uncertainties themselves in the variations of the UR from one region to the other one.

Even though the distribution of the emissions is highly heterogeneous in Europe, the grid cells of the transport model used in this study have a 3.75 °×2.5 °resolution (i.e. they cover areas that are much larger than megacities like London or Paris), so that we can assume that nearly all pixels have both rural and urban locations.

We clarify the rational for the NET233 network in Sect. 2.1 by adding in line 141: "… in which two sites are placed in each European land pixel of the LMDZv4 transport model (Fig. 1c). The NET233 network is denser than NET17 and NET43 in the high emitting regions, e.g. Germany, and also covers the region that is not well sampled by NET17 and NET43. However, the location of its 233 sites is not intended to be optimal since the emissions have a very heterogeneous spatial distribution. Their homogeneous spreads allow us to reduce the impact of representation and aggregation errors (Trampert and Snieder, 1996; Kaminski et al., 2001) and to assess the impact of having a dense network for all control regions."

2. Line 233: In the ICOS protocol, are the two-week samples going to be filled continuously, or are they only going to integrate mid-afternoon (or nighttime) air?

*That would very much change the sensitivity of the observations to FF $CO_2$, and the impact of transport errors.*

**Response:**

The present protocol for almost all of the ICOS $^{14}CO_2$ sites is to fill continuously two-week samples. And some sites fill continuously daily/weekly atmospheric $^{14}CO_2$. However, the option of intermittent filling of air samples is feasible in practice and has been used (Levin et al., 2008; Turnbull et al., 2016). On the other hand, the state-of-the-art transport models used for atmospheric inversion studies have still difficulties in simulating the vertical mixing during night-time and in the morning. So we prefer to consider this option of two-week afternoon sampling in the OSSE. We keep in mind the fact that using samples filled continuously over two weeks instead of during the afternoon only would definitely change the transport condition and sensitivity of the observation to FFCO2, and thus may give different values of UR and MR. But we assume that it should change the result in a quantitative but not qualitative way.

We add a sentence in line 232 to mention that common practice of $^{14}CO_2$ sampling is continuously over the course two weeks: "Current atmospheric $^{14}CO_2$ samples in Europe are usually filled continuously over the course of two weeks (Vogel et al., 2013; Levin et al., 2013). However, state-of-art inversion systems generally make use of data during the afternoon only, …" We already mentioned in the manuscript in the same paragraph that intermittent filling of air samples is practically feasible so that our definition of the observations in this study can be seen as a compromise between the current requirement of state-of-the-art inversion systems and current measurement practices.

**Response:**

We downloaded the "global fossil fuel emissions" under the "product" tab month by month. These files are hourly emissions for each month and for three emission heights. We summed the emissions at different emission heights together (our simulation of $FFCO_2$ assumes that all the emissions are emitted at surface). In this case, we obtained twelve files of global hourly emission fields for one year and each

file corresponds to one month.

**Response:**
   We prefer to consider that errors can have long spatial and temporal scales rather than a "systematic" component, since what the people usually call biases or systematic errors vary in time and can be difficult to predict and diagnose. The inversion system integrate such temporal and spatial consistency of the errors through the temporal and spatial correlations in **R**.
   The assessment of the model errors over long temporal scales, e.g. by using real tracer data was out of the scope of this study and of that of Wang et al. (2017). It has been the topic of numerous model inter-comparison studies in the past including LMDZ such as Peylin et al. (2011) and Patra et al. (2011). Locatelli (2015) also studied in details the transport errors by LMDZ. These studies show that the skills of LMDZ are in line with state-of-the-art transport models and there is no evidence that LMDZ has a systematic error compared to other transport models. Furthermore, our aim here is not to assign transport errors for the specific transport model we use, but rather for a typical global transport model in order to produce general results.
   The structure of the transport error is hard to assess for a specific transport models. Some studies have made some attempts to characterize it, but have come to different conclusions. For example, Lin and Gerbig (2005) determined the correlation timescale to be 2-3 hours for U-/V- winds. Broquet et al. (2011) analyzed the distribution of differences between the simulated and measured atmospheric mole fractions of [222]Rn (Radon). Temporal auto-correlations with 3 to 6-hour timescales are found in Broquet et al. (2011) based on such an analysis. Lauvaux et al. (2009) estimated the potential of the temporal correlations for transport error based on an ensemble of perturbations of the simulation of atmospheric transport. They found there are negative correlation between the atmospheric $CO_2$ mole fractions in the day and the night and the correlations (of the errors) with a lag time of 1 day are close to 0 for midafternoon data (Fig. 4a in Lauvaux et al., 2009). Miller et al. (2015) investigated the magnitude of temporally covarying atmospheric transport errors and found that removing temporal covariances in the transport would underestimate the transport error at the monthly scale, but their study did not try to use any function to describe the temporal auto-correlations so that their results are difficult to generalize.
   Most of these studies found temporal error correlations that are generally smaller than 1 day. If considering daily to monthly concentration averages, such a correlation should increase (if the transport error is a combination of error components at high and low frequencies). However, these various studies prove that the correlations in the transport error are challenging to estimate and the majority of existing inversion studies do not account for such potential correlations in atmospheric transport modeling (Peylin et al., 2013; Chevallier et al., 2010; Kadygrov et al., 2015; Peters et al., 2007; Gurney et al., 2008; Niwa et al., 2012). So, we keep the traditional

assumption.

We revise the paragraphs about the configuration of transport errors, as well as representation and aggregation errors in lines 328-337: "… For $\varepsilon_a$, the standard deviation for spring/summer and autumn/winter were derived. The standard deviation of the transport error at a given site is set-up proportional to the temporal standard deviation of the 1-year long time-series of the high-frequency variability of the detrended and deseasonalized simulated daily mean afternoon mixing ratios in the grid cell of the transport model, at which the sites are located. Such an estimation of transport error which relies on some results from Peylin et al. (2011) aims at representing the typical value for global transport models, not that of the specific transport model used in this study. The temporal auto-correlations in the representation and aggregation errors were characterized by Wang et al. (2017) using the sum of a long-term component and a short-term component: $r(\varDelta t) = a \times e^{-\varDelta t/b} + (1-a) \times e^{-\varDelta t/c}$ where $\varDelta t$ is the timelag (in days) and $a$, $b$, $c$ are parameters optimized by regressions against the samples of the errors. Furthermore, we do not include temporal auto-correlations in the transport error for simulated daily to 2-week mean afternoon $FFCO_2$ gradients, since previous studies of the auto-correlations of the transport errors have not evidenced that they should be significant at daily scale (Lin and Gerbig, 2005; Lauvaux, 2009; Broquet et al., 2011). This choice follows the corresponding discussion by Wang et al. (2017) and implicitly ignores that transport model errors likely bear long-term components (often referred to as "biases", Miller et al., 2015) even when being dominated by components on short timescales. The corresponding values of the standard deviation and the modelling of temporal autocorrelation of the observation errors for 2-week/daily mean afternoon $FFCO_2$ gradients are listed in Table S3 and Table S4.

A simpler account of the spatial correlations in the observation errors is derived from the diagnostics of Wang et al. (2017). We do not account for the spatial correlation in the representation error, as the scale of the spatial correlation according to Wang et al. (2017), i.e. 55-89 km, is much smaller than the size of the grid cells of the global transport model $\mathbf{H}_{transp}^{LMDZ}$ used for the inversion. When there are more than two sites are located in the same grid cell of the transport model, we consider that the aggregation errors and the transport errors are fully correlated between these sites, according to the definition by Wang et al. (2017). We do not account for spatial correlations between aggregation errors for measurements made at sites in different grid cells, because the scale of the spatial correlation is 171 km and is smaller than the size of the grid cell, according to Wang et al. (2017). Finally, we do not account for spatial correlations between transport errors or measurements made at sites in different grid cells."

some of them just happened to be skewed in one direction? This is especially a concern for the NET17 network, since there are so few measurements, with scant opportunity to average over the perturbations.
**Response:**

    We carefully analyzed and discussed the UR and MR results keeping in mind and explicitly recalling the problems of working with a single realization of $\mathbf{y}_o$ and $\mathbf{x}^b$. The risk of under- or over-estimating the posterior error in the emission budgets when using one realization only for the prior and measurement errors is the reason why we compute the "statistics of misfits" at the monthly scale and the "MR for monthly emissions" (see our answer to major question 3). By assuming that the monthly misfits all follow the same distribution, we roughly consider that we have an ensemble of twelve realizations of monthly inversions for which the average-based MR give a reliable indicator or the error reduction that should not be significantly skewed by sampling errors.

    Of note is also that even when considering NET17 and a 2-week sampling strategy, we have 416 data over the year and thus 416 realizations of perturbations to individual observations. Finally, the scores of UR should definitely be considered as the reference indicators of the inversion behavior, and the MR are mainly analyzed to provide confidence in these UR, as explained in Sect. 2.2.1.

    The perturbation of the observation according to $\mathbf{R}$ even when a single realization of the observation and prior errors is considered is a common practice in inversion because, we believe, this is what still makes most sense if having to assess the uncertainty reduction using a single inversion. In other words, we do not have a simple idea of perturbation that would make the single realization more adapted for assessing the typical impact of the errors.

    We revise the paragraphs in Sect. 2.2.1, lines 203-204 to clarify the use of one realization of observations: "We focus on uncertainties and misfits at both monthly and annual scales. However, we can have only one practical realization for $\mathbf{x}^b$, $\mathbf{y}_o$ and $\mathbf{x}^a$ following the protocol of that is presented in Sect. 2.3. Therefore, the assessment of the performance of the inversion for a given region-month using the corresponding score of MR may be over- or under-estimated due to the lack of sampling of the prior and observation errors. Consequently, …"

this case, if the structure of **B** is wrong, the repartition of the constraints from the observations between these different region-months could be erroneous.

The fact that the aggregation error has a non-Gaussian and biased distribution while the inversion system believe it is Gaussian and unbiased can also feed the generation of negative MR as well as problems of sampling errors when computing scores of MR as suggested by the reviewer. However we believe that the impact of these additional factors is relatively small because when the setup of **B** matches well (even though not perfectly) with that of the actual prior errors, we have very few negative MRs.

We add in lines 546 to discuss the negative scores of MR: "… This degradation occurs even when using daily measurements or the network NET233. A first explanation is that the signature of the errors in the prior emission estimates in the FFCO$_2$ fields has a smaller amplitude than the observation errors and thus the ability to filter this information for a proper correction of the emissions strongly relies on the knowledge of the prior uncertainty covariance. If **B** misses the amplitude and the temporal and spatial correlations of the actual errors, the system can translate observation errors into corrections to the emissions. Furthermore, some of the region-months are poorly constrained by the observations (due to the meteorological conditions and/or to the observation network spatial distribution), and the corrections to such region months is imposed by the extrapolation of the corrections to other region-months following the uncertainty structures characterized by **B**. If those structures do not represent the actual errors correctly, the system could apply corrections with a wrong sign or amplitude to the poorly observed region-months. A similar problem occurs when the network can constrain the sum of the budgets for several region-months but not the individual budgets of these region months (due to being too coarse). If the structure of **B** is wrong, the repartition of the constraint from the observations between these different region-months can be erroneous. All these analysis reveal the difficulty to capture the signatures of uncertainties in the prior emission estimate from the assimilated prior-model data misfits in our specific inverse modeling problem and thus to derive good corrections when the prior uncertainty covariance matrix is not configured properly."

**Response:**

There is not a strong reason to think that there could be some major change in the uncertainties in the estimate of nuclear and disequilibrium fluxes of $^{14}$C from year to year, which is what really matters on the inversion results (rather than changes in the absolute value for these fluxes), as discussed in the response to the first major comments. This response also indicates that we assume that such uncertainties should have a relatively weak impact on the inversion results.

The transport conditions (and thus $\mathbf{H}$ and the uncertainties in the transport modeling in $\mathbf{R}$) may vary significantly from year to year which could lead to variations of the sensitivity of the observations to the emissions and variations of the scores of uncertainty reduction. We could even observe trends in the transport conditions over several years (e.g. the shallowing of the PBL between the 1990s and the 2000s by Aulagnier et al., 2009; Ramonet et al., 2010). However, the impact of the resulting changes in atmospheric transport condition on the UR and posterior uncertainties, and thus on our computation of the trend detectability is difficult to anticipate. We assume that this should not result in a behavior that is different from the one described by our computation with constant posterior uncertainties from year

to year (it should change the result in a quantitative but not qualitative way). Conducting tests with varying posterior uncertainties would thus fall out of the scope of our assessment of the typical uncertainties in trends for a typical uncertainty in annual emissions, and we thus think that it is not worth investigating it.

We delete the simple statement in line 479: "…fully independent , we calculate the uncertainty in relative trends …" and add a paragraph to discuss that the changing transport may have an impact on the trend detection but quantifying this impact would require detailed studies and further investigation after line 490: "Our assumption that the posterior uncertainties in annual emissions have the same amplitude from year to year should not strongly drive the results, so the results here give a good indication of the level of uncertainty in the trend detection for a typical level of uncertainty at the annual scale. However, changes of the transport from year to year or on decadal scales (Aulagnier et al. 2009; Ramonet et al., 2010) may change the level of the sensitivity of the observations to the emissions, i.e., the level of the atmospheric constraint of the inversions which leads to uncertainty reduction, and thus the level of posterior uncertainties on the same timescales. A more complex model accounting for varying levels of annual posterior uncertainties may thus be useful to refine the quantification of the uncertainty in the trends. Of note is that the level of uncertainties in the trends could be increased if the modeling framework accounts for the trends in the transport or in the sources of $^{14}CO_2$ other than the fossil fuel emissions. Such trends in the modeling errors may have to be considered for applications with real data."

$$J = \frac{1}{2}\left(\vec{y} - \tilde{\vec{y}}\right)^T R^{-1}\left(\vec{y} - \tilde{\vec{y}}\right) \qquad (1)$$

where R = cov($\vec{y}$) (R in their case contains the posterior error estimates on fluxes),

then the optimal estimate of $\vec{p}$ and the corresponding covariance are

$$\vec{p} = (X^T R^{-1} X)^{-1} X^T R^{-1} \vec{y}$$

$$\text{cov}(\vec{p}) = (X^T R^{-1} X)^{-1} X^T R^{-1} X (X^T R^{-1} X)^{-1}$$

**Response:**

The reviewer is right if cov(**Y**) (in the review's equations, it's **R**) is not diagonal or if the diagonal terms (variances of the variables in **y** vector) are not equal. In fact, the last equation proposed by the reviewer can be further simplified to:

$$\text{cov}(\vec{p}) = (X^T R^{-1} X)^{-1}$$

The equation we wrote as Eqs. C-3 and C-4 only applies when cov(**Y**) is diagonal and the terms in **y** all have equal variance, that is cov(**Y**)=$\sigma^2$**I** where $\sigma^2$ is the variance for all variables and **I** is an identity matrix. The equivalence between our equations and those proposed by the reviewer can be proved:

cov($p$)=($\mathbf{X^T X}$)$^{-1}\mathbf{X^T}$cov(**Y**)$\mathbf{X(X^T X)}$$^{-1}$=($\mathbf{X^T X}$)$^{-1}\mathbf{X^T}\sigma^2\mathbf{IX(X^T X)}$$^{-1}$

$\qquad$=$\sigma^2(\mathbf{X^T X})$$^{-1}\mathbf{X^T X(X^T X)}$$^{-1}$=$\sigma^2(\mathbf{X^T X})$$^{-1}$=($\mathbf{X^T}\sigma^{-2}\mathbf{IX}$)$^{-1}$=[$\mathbf{X^T}$cov(**Y**)$^{-1}\mathbf{X}$]$^{-1}$

In this sense, the equations proposed by the reviewer is more generalized. In this paper, we indeed used the cov(**Y**) in the equal variance case, so that the computation and results of the paper are correct.

In order to avoid any confusion, we revise the sentence in lines 478-479: "Assuming that the absolute values of the standard deviations of the uncertainties in annual emissions of different years (in Tg/year) are identical and that these uncertainties are fully independent …" and we change the equations C-3 and C4 as proposed by the reviewer:

$p$=($\mathbf{X^T}$cov$^{-1}$(**Y**)**X**)$^{-1}\mathbf{X^T}$cov$^{-1}$(**Y**)$y$ $\qquad$ (C-3)

cov($p$)=($\mathbf{X^T}$cov$^{-1}$(**Y**)**X**)$^{-1}$ $\qquad$ (C-4)

19. Table 2: In columns 2 and 3, I believe rows 3 and 4 have been flipped.
**Response:**
The reviewer is right. We correct this mistake accordingly.

20. Figure 2(a): Am I supposed to see 56 colors in the world map? I don't. I think the problem is that the country and state boundaries overlap with the region boundaries. I would suggest, at least in the world map, only showing the region boundaries from the control vector and eliminating the country and state boundaries.
**Response:**
Here, we do not use 56 colors in the map actually because it is hard to find 56

colors that are visibly differentiable. In fact, we repeatedly use 12 colors for non-adjacent regions. For example, the Northern Europe, Middle East, one region in the USA and one region in China are all red. But because they are in different continents, so that they represent four different regions. We change Figure 2 by removing the lines country and state boundaries, as the reviewer suggested, and mention the fact we repeatedly use 12 colors for non-adjacent regions in the caption:

[Figure]

[revised manuscript text omitted]
}_{i,m}^{6m}, \frac{1}{3}\sum_{m=4}^{6}\mathbf{r}_{i,m}^{6m}, \frac{1}{3}\sum_{m=7}^{9}\mathbf{r}_{i,m}^{6m}, \frac{1}{3}\sum_{m=10}^{12}\mathbf{r}_{i,m}^{6m})$$

(B-6)

And the corresponding residues:

$$\mathbf{r}_{i,m}^{3m} = \Delta\mathbf{x}_{i,m} - \varepsilon_{ann} - \frac{1}{6}\sum_{m=1}^{6}\mathbf{r}_{i,m}^{ann} - \frac{1}{3}\sum_{m=1}^{3}\mathbf{r}_{i,m}^{6m} \qquad (\text{if } m \le 3)$$

$$\mathbf{r}_{i,m}^{3m} = \Delta\mathbf{x}_{i,m} - \varepsilon_{ann} - \frac{1}{6}\sum_{m=1}^{6}\mathbf{r}_{i,m}^{ann} - \frac{1}{3}\sum_{m=4}^{6}\mathbf{r}_{i,m}^{6m} \qquad (\text{if } 4 \le m \le 6)$$

$$\mathbf{r}_{i,m}^{3m} = \Delta\mathbf{x}_{i,m} - \varepsilon_{ann} - \frac{1}{6}\sum_{m=7}^{12}\mathbf{r}_{i,m}^{ann} - \frac{1}{3}\sum_{m=7}^{9}\mathbf{r}_{i,m}^{6m} \qquad (\text{if } 7 \le m \le 9)$$

$$\mathbf{r}_{i,m}^{3m} = \Delta\mathbf{x}_{i,m} - \varepsilon_{ann} - \frac{1}{6}\sum_{m=7}^{12}\mathbf{r}_{i,m}^{ann} - \frac{1}{3}\sum_{m=10}^{12}\mathbf{r}_{i,m}^{6m} \qquad (\text{if } 10 \le m \le 12)$$

(B-7)

730    The 1-month variation $\varepsilon_{1m}$ equals the standard deviation of these residues:

$$\varepsilon_{1m} = \text{SD}(\mathbf{r}_{i,m}^{3m})$$

(B-8)

Using such a decomposition, the root mean square of the errors (RMSE) between the prior and the "true" values $\Delta\mathbf{x}_{i,j}$ satisfy the following equation:

$$\text{RMSE}_i = \frac{1}{12}\sum_{m=1}^{12}\Delta\mathbf{x}_{i,m}^2 = \varepsilon_{ann}^2 + \varepsilon_{6m}^2 + \varepsilon_{3m}^2 + \varepsilon_{1m}^2$$

(B-9)

735    Finally, for the diagonal entries of the **B** matrix corresponding to the monthly emissions of region $i$, they are equal to the RMSE$_i$, for the non-diagonal entries, the covariance between month $j$ and month $k$ for a given region is expressed as the sum of the products of the different variations multiplied by corresponding correlations (expressed by exponential decay functions) at different time scales:

$$\mathbf{B}_{i,(m,n)} = \varepsilon_{ann}^2 + \varepsilon_{6m}^2 + \varepsilon_{3m}^2 + \varepsilon_{1m}^2 \qquad (\text{if } m = n)$$

$$\mathbf{B}_{i,(m,n)} = \varepsilon_{ann}^2 + e^{-\frac{|n-m|}{6}}\varepsilon_{6m}^2 + e^{-\frac{|n-m|}{3}}\varepsilon_{3m}^2 + e^{-\frac{|n-m|}{1}}\varepsilon_{1m}^2 \qquad (\text{if } m \ne n)$$

(B-10)

**Appendix C. Calculation of trends and corresponding uncertainties**

740

Assuming the linear trend of the FFCO$_2$ emissions in an $n$-year period is to be calculated, which satisfies the function:

$$\mathbf{y} \approx \tilde{\mathbf{y}} = a\mathbf{x} + b \qquad\qquad\qquad\qquad\qquad\qquad\qquad\qquad\qquad\qquad\text{(C-1)}$$

where $\mathbf{y}$ is the vector of annual emissions for the $n$ years, $\tilde{\mathbf{y}}$ is the predicted value by the regression and $\mathbf{x}$ is the vector of corresponding years, the slope $a$ is the linear trend we are going to calculate by linear regression. We rewrite Eq. (C-1) as

745    follows:

$$\begin{bmatrix} y_1 \\ \vdots \\ y_{10} \end{bmatrix}_{\boldsymbol{y}} \approx \begin{bmatrix} y_1 \\ \vdots \\ y_{10} \end{bmatrix}_{\boldsymbol{y}} = \underbrace{\begin{bmatrix} x_1 & 1 \\ \vdots & \vdots \\ x_{10} & 1 \end{bmatrix}}_{\boldsymbol{X}} \begin{bmatrix} a \\ b \end{bmatrix}_{\boldsymbol{p}} \tag{C-2}$$

Thus the linear trend $a$ and the interception $b$ can be solved using linear algebra. With the notations used in Eq. (C-2), the result of the linear regression is:

$$\boldsymbol{p} = (\mathbf{X}^{\mathrm{T}}\mathrm{cov}^{-1}(\mathbf{Y})\mathbf{X})^{-1}\mathbf{X}^{\mathrm{T}}\mathrm{cov}^{-1}(\mathbf{Y})\boldsymbol{y}$$

750 (C-3)

the associated uncertainties in the regression parameters in vector $\boldsymbol{p}$ is thus given by the following covariance matrix:

$$\mathrm{cov}(\boldsymbol{p}) = (\mathbf{X}^{\mathrm{T}}\mathrm{cov}^{-1}(\mathbf{Y})\mathbf{X})^{-1}\mathbf{X}^{\mathrm{T}}\mathrm{cov}(\mathbf{Y})\mathbf{X}(\mathbf{X}^{\mathrm{
[revised manuscript text omitted]